# Enhanced Cyclic Coordinate Descent Methods for Elastic Net Penalized Linear Models

**Yixiao Wang**[*][†]
Department of Statistical Science
Duke University
yw676@duke.edu

**Zishan Shao**[*]
Department of Statistical Science
Duke University
zs89@duke.edu

**Ting Jiang**
Department of Electrical
& Computer Engineering
Duke University
tj147@duke.edu

**Aditya Devarakonda**
Department of Computer Science
Wake Forest University
devaraa@wfu.edu

## Abstract

We present a novel enhanced cyclic coordinate descent (ECCD) framework for solving generalized linear models with elastic net constraints that reduces training time in comparison to existing state-of-the-art methods. We redesign the CD method by performing a Taylor expansion around the current iterate to avoid nonlinear operations arising in the gradient computation. By introducing this approximation we are able to unroll the vector recurrences occurring in the CD method and reformulate the resulting computations into more efficient batched computations. We show empirically that the recurrence can be unrolled by a tunable integer parameter, $s$, such that $s > 1$ yields performance improvements without affecting convergence, whereas $s = 1$ yields the original CD method. A key advantage of ECCD is that it avoids the convergence delay and numerical instability exhibited by block coordinate descent. Finally, we implement our proposed method in C++ using Eigen to accelerate linear algebra computations. Comparison of our method against existing state-of-the-art solvers show consistent performance improvements of $3\times$ in average for regularization path variant on diverse benchmark datasets. Our implementation is available at https://github.com/Yixiao-Wang-Stats/ECCD.

## 1 Introduction

Generalized linear model (GLM) is a cornerstone of modern machine learning and statistics with applications demanding both variable selection and regularization. Among the various penalties, the elastic-net [1], which combines $\ell_1$ and $\ell_2$ regularization, has attracted significant interest because it not only promotes sparsity but also alleviates some of the limitations inherent in using either penalty alone, especially in high-dimensional settings. The coordinate descent (CD) algorithm has become a popular method for optimizing such models due to its simplicity and effectiveness, typically updating a subset of model parameters at a time [2].

Despite the widespread success of coordinate-wise updates, block generalization for training GLMs have encountered numerical stability issues especially for large block sizes. Existing approaches

---

[*]Equal contribution.
[†]Correspondence: yw676@duke.edu

39th Conference on Neural Information Processing Systems (NeurIPS 2025).

utilize second-order approximations to solve elastic-net penalized GLMs. However, utilizing BCD to train such models has resulted in deteriorating accuracy as the block size increases.

We introduce the **enhanced cyclic coordinate descent (ECCD)** method for elastic-net penalized GLMs which performs a recurrence unrolling of the single coordinate descent update. We couple the recurrence unrolling with a second-order correction in the gradient computation in order to reduce the frequency of expensive nonlinear link function evaluations. The correction term implicitly incorporates updates from one coordinate update to the next, which yields accuracy improvements over classical BCD applied to elastic-net penalized GLMs. The main contributions of this paper are:

1. Derivation of a coordinate descent update strategy that improves accuracy of block coordinate updates over the classical BCD method for elastic-net penalized GLMs.

2. Theoretical analysis to bound the approximation error and theoretically optimal choice of block size for our update strategy.

3. Experimental evaluation of single and sequential C++ implementation of our approach compared against state-of-the-art GLM solvers which shows in average $3\times$ speedup on path fits across diverse benchmark datasets.

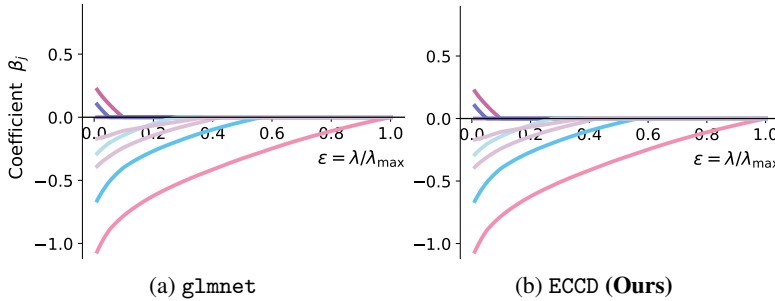

(a) `glmnet`  (b) ECCD **(Ours)**

Figure 1: Coefficient trajectories $\beta_j$ for the `diabetes` dataset over 100 values of the regularization $\lambda$ (see Table 15 in Appendix for dataset dimensions). ECCD achieves identical solutions and induced sparsity patterns to GLMnet.

## 2   Related Work

**Generalized Linear Models**   In the context of high-dimensional inference, GLMs have been extensively studied and optimized for scalable regularization-path computation. The seminal `glmnet` framework [2] introduced cyclic coordinate descent with warm starts and an inexact line search-trading off formal convergence guarantees to efficiently trace elastic-net paths for linear, logistic, and multinomial objectives. Building on nonconvex penalty theory, `ncvreg` [3] extends these ideas to MCP and SCAD penalties, while `biglasso` [4] leverages memory-mapping and sparse linear algebra to handle datasets that exceed RAM capacity. For structured sparsity, `grpnet` [5] implements block coordinate updates for group lasso, and `adelie` [6] employs an ADMM-based scheme to accelerate group-penalized regressions. To reduce per-iteration overhead, many methods integrate feature-screening techniques: "safe rules" [7] guarantee no loss of optimality by preemptively discarding inactive features, and "strong rules" [8] perform aggressive but heuristic pruning with minimal KKT checks. These screening strategies have become staples in modern regularization-path toolkits, substantially lowering the computational barrier to solving large-scale GLMs.

**$s$-Step and Block Coordinate Methods**   The $s$-step paradigm, originally developed to reduce communication in Krylov methods [9, 10, 11, 12, 13, 14]. The proposed $s$-step Krylov methods were later stabilized and implemented in distributed-memory systems for multigrid applications [15, 16, 17, 18]. More recently, these ideas have been extended to CD solvers and generalized to nonlinear convex optimization [19, 20, 21, 22, 23, 24]. Block generalizations of CD methods have also been developed with recent work establishing convergence guarantees for distributed optimization of strongly convex objectives in the deterministic and stochastic block update settings [25, 26]. Our ECCD algorithm builds directly on these prior results by generalizing the $s$-step technique to the class of GLMs. Unlike BCD applied to GLMs, ECCD couples the $s$-step technique

with a Taylor expansion of the gradient computation to ensure numerical stability for block sizes larger than is feasible through the use of BCD. We also show that our method yields speedups over existing state-of-the-art GLM solvers.

## 3 Preliminary

### 3.1 Generalized Linear Model

| Type | Distribution | Link | $\mathbb{E}[Y \mid X]$ | $F(\theta)$ | $F'(\theta)$ | $F''(\theta)$ | $d(\tau)$ |
|------|-------------|------|------------------------|-------------|--------------|---------------|-----------|
| Gaussian | $\mathcal{N}(\mu, \sigma^2)$ | Identity | $\theta$ | $\frac{1}{2}\theta^2$ | $\theta$ | $1$ | $\sigma^2$ |
| Bernoulli | $\text{Bernoulli}(p)$ | Logit | $\frac{1}{1+e^{-\theta}}$ | $\log\left(1+e^\theta\right)$ | $\frac{e^\theta}{1+e^\theta}$ | $\frac{e^\theta}{(1+e^\theta)^2}$ | $1$ |
| Poisson | $\text{Poisson}(\lambda)$ | Log | $e^\theta$ | $e^\theta$ | $e^\theta$ | $e^\theta$ | $1$ |
| Gamma | $\text{Gamma}(\alpha, \beta)$ | Inverse | $-\frac{1}{\theta}$ | $-\log(\theta)$ | $-\frac{1}{\theta}$ | $\frac{1}{\theta^2}$ | $\alpha$ |

Table 1: Common GLMs and their properties. The functions $F(\theta)$, $F'(\theta)$, and $F''(\theta)$ are the cumulant and its derivatives, and $d(\tau)$ is the dispersion parameter.

We consider a GLM with mean response: $\mathbb{E}[Y \mid X] = f^{-1}(X\boldsymbol{\beta})$, where $X \in \mathbb{R}^{n \times p}$ is the design matrix, $\boldsymbol{\beta} \in \mathbb{R}^p$ is the coefficient vector, and $f$ is an invertible link function. We assume the response variable $Y$ follows an exponential-family distribution with probability mass/density function parameterized by $\theta$ and $\tau$.

$$f_Y(y \mid \theta, \tau) = c(y, \tau) \exp\left(\frac{b(\theta)^\top T(y) - F(\theta)}{d(\tau)}\right), \tag{1}$$

where $d(\tau)$ is the dispersion parameter, $b(\theta)$ is the natural parameter, $T(y)$ is a sufficient statistic of $y$ and $F(\theta)$ is the cumulant-generating function (CGF). As demonstrated in Table 1, the specific choice of distribution determines the form of the cumulant function $F(\theta)$. Throughout, we assume that the first two derivatives of $F$, $F'(\cdot)$ and $F''(\cdot)$, exist. In the *canonical form*, where $b(\theta) = \theta$ and $T(y) = y$, one has: $\mathbb{E}[Y] = F'(\theta)$, $\text{Var}(Y) = F''(\theta)\, d(\tau)$. We solve $\boldsymbol{\beta}$ by maximizing the log-likelihood

$$\ell(\boldsymbol{\beta}) = \sum_{i=1}^n \frac{y_i \theta_i - F(\theta_i)}{d(\tau)} + C(y_i, \tau), \quad \theta_i = \boldsymbol{x}_i^\top \boldsymbol{\beta} \tag{2}$$

where $\boldsymbol{x}_i^\top$ is the $i$-th row of design matrix $X$. We define $g_j$ and $h_j$ as the first and second derivatives of the log-likelihood with respect to $\beta_j$, respectively:

$$g_j = \frac{\partial \ell}{\partial \beta_j} = \sum_{i=1}^n \frac{y_i - F'(\theta_i)}{d(\tau)} x_{ij}, \quad h_j = \frac{\partial^2 \ell}{\partial \beta_j^2} = -\sum_{i=1}^n \frac{F''(\theta_i)}{d(\tau)} x_{ij}^2. \tag{3}$$

### 3.2 Traditional Coordinate Descent for Elastic Net Regression

We begin with the classical CD method for elastic net–regularized generalized linear models, as developed in prior work [2], which accelerates updates via a first-order Taylor expansion, bypassing line search at the expense of formal convergence guarantees [2, 27]. One can write the objective as

$$\mathcal{L}(\boldsymbol{\beta}) = -\frac{1}{n}\ell(\boldsymbol{\beta}) + \lambda\left(\tfrac{1-\alpha}{2}\|\boldsymbol{\beta}\|_2^2 + \alpha\|\boldsymbol{\beta}\|_1\right), \tag{4}$$

where $-\ell(\boldsymbol{\beta})$ is the loss function, which is the negative log-likelihood for GLMs, and $\lambda \geq 0$ and $\alpha \in [0, 1]$ control the overall magnitude and the relative weighting of the $\ell_2$ (ridge-like) and $\ell_1$ (lasso-like) penalties, respectively. We solve $\hat{\boldsymbol{\beta}} = \arg\min_{\boldsymbol{\beta}} \mathcal{L}(\boldsymbol{\beta})$ to obtain the regularized estimator.

In a cyclic coordinate descent setting, we update each coefficient $\beta_j$ while fixing the others. Using a second-order Taylor expansion around $\beta_j^{(t)}$, the value of $\beta_j$ at iteration $t$, one obtains the approximate local objective:

$$\min_{\beta_j} -\frac{1}{n}\left[\ell(\boldsymbol{\beta}^{(t)}) + g_j(\beta_j - \beta_j^{(t)}) + \tfrac{1}{2}h_j(\beta_j - \beta_j^{(t)})^2\right] + \lambda\left(\tfrac{1-\alpha}{2}\beta_j^2 + \alpha|\beta_j|\right), \tag{5}$$

where $g_j$ and $h_j$ are the first- and second-order derivatives evaluated at $\boldsymbol{\beta}^{(t)}$. Setting the gradient with respect to $\beta_j$ to zero yields

$$\frac{\partial \mathcal{L}(\beta)}{\partial \beta_j} = -\frac{1}{n}g_j - \frac{1}{n}h_j\big(\beta_j - \beta_j^{(t)}\big) + \lambda(1-\alpha)\,\beta_j + \lambda\alpha\,\text{sign}(\beta_j) = 0. \tag{6}$$

Solving for $\beta_j$ gives the coordinate descent update:

$$\beta_j^{(t+1)} \leftarrow \frac{S\Big(\frac{1}{n}g_j - \frac{1}{n}h_j\,\beta_j^{(t)},\ \lambda\alpha\Big)}{-\frac{1}{n}h_j + \lambda(1-\alpha)}, \tag{7}$$

where $S(z,\gamma)$ denotes the soft-thresholding operator defined by

$$S(z,\gamma) = \begin{cases} z - \gamma & \text{if } z > 0 \text{ and } \gamma < |z|, \\ z + \gamma & \text{if } z < 0 \text{ and } \gamma < |z|, \\ 0 & \text{if } \gamma \geq |z|. \end{cases}$$

### 3.3 Efficient Screening and Warm Start Strategy

To accelerate sparse estimation in GLMs, prior works [2, 8] have introduced complementary techniques that reduce computational overhead while ensure exact solutions. In this work, we adopt a pathwise CD framework over a decreasing sequence of regularization parameters $\lambda_1 > \lambda_2 > \cdots > \lambda_K$, where $\lambda_1 = \lambda_{\max} := \max_j |X_j^\top(y - \bar{y})|/\alpha$, and $\lambda_K = \varepsilon\lambda_{\max}$, with $\varepsilon \in [10^{-4}, 10^{-2}]$. When computing a stand-alone regularized solution, these strategies can also be applied to a single $\lambda$ value.

At each $\lambda_k$, we *warm start* optimization using the solution at $\lambda_{k-1}$ and restrict update to a reduced *active set* $\mathcal{A}_k \subseteq \{1, \ldots, p\}$. The set initially includes all previously nonzero coordinates and any that violate the Karush–Kuhn–Tucker (KKT) conditions. We then iterate: update $\beta_j$ for $j \in \mathcal{A}_k$, perform a full KKT check, augment $\mathcal{A}_k$ with any newly violating indices, and repeat until convergence. By cycling only over $\mathcal{A}_k$, each iteration costs $\mathcal{O}(n|\mathcal{A}_k|)$ instead of $\mathcal{O}(np)$, yielding substantial speedups in high-dimensional settings.

To further reduce the size of the initial working set, the *sequential strong rule* [8, 7] is employed at each $\lambda_k$. In logistic regression, the rule suggests discarding feature $j$ from the working set if

$$\big|X_j^\top r^{(k-1)}\big| < 2\,\lambda_k\alpha - \lambda_{k-1}\alpha, \quad r^{(k-1)} = y - \hat{p}^{(k-1)},\ \hat{p}^{(k-1)} = \nabla F(X\beta(\lambda_{k-1}) + \beta_0). \tag{8}$$

Although this is a heuristic, performing a post-convergence KKT check guarantees recovery of any missed variables. Warm starts, strong-rule screening, and active-set cycling with KKT correction together form a robust, scalable backbone for both single-$\lambda$ and pathwise GLM solvers.

## 4 Methodology

### 4.1 Block Coordinate Descent

GLMs are well-suited for high-dimensional regimes ($p \gg n$) because the $\ell_1$ penalty encourages sparse solutions. A natural generalization of CD is BCD, in which we update a block of $s$ coordinates jointly at each iteration. However, we observe that naive BCD often diverges for moderate to large block sizes, as shown in Figure 2a (see Table 3 for quantitative results).

Each coordinate $j$ in a block remains the standard CD update (Eq. 7), but BCD differs in how it handles the intermediate value $\theta_i = \boldsymbol{x}_i^\top\boldsymbol{\beta}$. In classical CD, $\theta_i$ is recomputed before every single-coordinate update, ensuring that both the gradient $g_j$ and curvature $h_j$ reflect the most recent $\boldsymbol{\beta}$. In contrast, BCD recomputes $\theta_i$ only once per block, reusing stale values of $g_j$ and $h_j$ across $s$ updates. This delayed refresh can introduce significant numerical instability when $s$ is large. For completeness, we defer the full derivation of the BCD update rule to Appendix A.1.

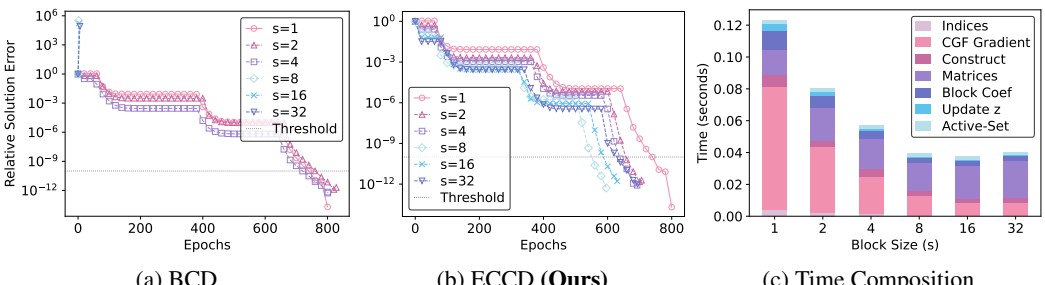

(a) BCD          (b) ECCD (**Ours**)          (c) Time Composition

Figure 2: Results on the `duke` dataset with $\alpha = 0.5$ and $\epsilon = 0.01$. (a) Convergence of BCD exhibits numerical instability as block size $s$ increases. (b) Convergence of ECCD remains stable even for $s = 32$. The convergence threshold is set to 1e-10. (c) Breakdown of ECCD's runtime components.

## 4.2 Enhanced Cyclic Coordinate Descent

To stabilize block-wise CD updates, we propose Enhanced Cyclic Coordinate Descent (ECCD). ECCD incorporates second-order approximation corrections through Taylor expansion within each block, compensating for the error that accumulates when multiple coordinates share the same input to the link function. This targeted correction restores the descent property and empirical convergence even for large block sizes, as shown in Figure 2b.

We now derive ECCD. Let $s \leq p$ be the block size, and consider the $k$-th block. For each coordinate in this block, indexed by $\ell \in \{1, 2, \ldots, s\}$, we update $\boldsymbol{\beta}$ as

$$\boldsymbol{\beta}^{(k-1)s+\ell} = \boldsymbol{\beta}^{(k-1)s} + \sum_{i=0}^{\ell-1} \Delta\beta^{(k-1)s+i} \, \boldsymbol{e}^{(k-1)s+i}, \tag{9}$$

where $\boldsymbol{e}^{(k-1)s+i}$ is the standard basis vector corresponding to the $((k-1)s+i)$-th coordinate in the descent step, and $\Delta\beta^{(k-1)s+i}$ is the scalar change at iteration $(k-1)s+i$. Equivalently, we can write $\boldsymbol{e}_{j_i}$ with $j_i \equiv ((k-1)s+i) \,(\mathrm{mod}\, p)$ to emphasize the cyclic ordering of the coordinates.

Throughout our paper, we assume the design matrix $X$ is standardized: each column is centered and scaled so $|x_j|^2 = n$. Let $\beta_0$ be the intercept. For the gradient term $F'(\beta_0^{(k-1)s+\ell} + X\boldsymbol{\beta}^{(k-1)s+\ell})$, take a first-order Taylor expansion at $\beta_0^{(k-1)s+\ell} + X\boldsymbol{\beta}^{(k-1)s}$ and denote the linearized term by $F'_q$.

$$F'_q\big(\beta_0^{(k-1)s+\ell}\mathbf{1}_n + X\boldsymbol{\beta}^{(k-1)s+\ell}\big) = F'\big(\beta_0^{(k-1)s+\ell}\mathbf{1}_n + X\boldsymbol{\beta}^{(k-1)s}\big)$$
$$+ \sum_{i=0}^{\ell-1} F''\big(\beta_0^{(k-1)s+\ell}\mathbf{1}_n + X\boldsymbol{\beta}^{(k-1)s}\big) \odot \big(X\,\boldsymbol{e}^{(k-1)s+i}\big)\Delta\beta^{(k-1)s+i}. \tag{10}$$

For numerator, we derive the coefficient update $\Delta\beta_{j_\ell}^{(k-1)s+\ell}$ for each coordinate $j_\ell \equiv ((k-1)s+\ell) \,(\mathrm{mod}\, p)$ *without* considering the $\ell_1$-penalty. Using the approximation $F'_q$, we obtain:

$$\phi_{j_\ell}^{((k-1)s+\ell)} = \frac{1}{n\,d(\tau)}\left[\mathbf{y}^\top X\,\boldsymbol{e}^{(k-1)s+\ell} - \boldsymbol{e}^{(k-1)s+\ell,\top}X^\top F'_q\big(\beta_0^{(k-1)s+\ell}\mathbf{1}_n + X\boldsymbol{\beta}^{(k-1)s+\ell}\big)\right]$$
$$- \lambda\,(1-\alpha)\,\beta_{j_\ell}^{(k-1)s+\ell}$$
$$= \frac{1}{n\,d(\tau)}\left[\mathbf{y}^\top X\,\boldsymbol{e}^{(k-1)s+\ell} - \boldsymbol{e}^{(k-1)s+\ell,\top}X^\top F'\big(\beta_0^{(k-1)s}\mathbf{1}_n + X\boldsymbol{\beta}^{(k-1)s}\big)\right.$$
$$\left.- \sum_{i=0}^{\ell-1} \boldsymbol{e}^{(k-1)s+\ell,\top}X^\top\Big(F''\big(\beta_0^{(k-1)s}\mathbf{1}_n + X\boldsymbol{\beta}^{(k-1)s}\big) \odot \big(X\,\boldsymbol{e}^{(k-1)s+i}\big)\Big)\Delta\beta^{(k-1)s+i}\right]$$
$$- \lambda\,(1-\alpha)\,\beta_{j_\ell}^{(k-1)s} \tag{11}$$

---

**Algorithm 1** ECCD for GLMs with Elastic Net Penalty (Active-Set Version)

---

**Require:** $X \in \mathbb{R}^{n \times p}$, $\mathbf{y} \in \mathbb{R}^n$, $\boldsymbol{\beta}^{(0)} \in \mathbb{R}^p$, $\beta_0^{(0)} \in \mathbb{R}$, $\lambda > 0$, $\alpha \in [0, 1]$, $s, H \in \mathbb{N}$, $\mathtt{tol} > 0$, active set $\mathcal{A} \subseteq \{1, \ldots, p\}$, $F(\cdot)$: CGF, $d(\tau)$: dispersion parameter

1: **for** outer $= 1, \ldots, H$ **do**
2:     **Intercept Update:** $\beta_0^{(t)} \leftarrow \beta_0^{(t-1)} + \frac{\mathbf{1}_n^\top (\mathbf{y} - \nabla F)}{\mathbf{1}_n^\top \nabla^2 F}$
3:     **ECCD Update:**
4:     **for** block $= 1, \ldots, \lceil |\mathcal{A}|/s \rceil$ **do**
5:         $k \leftarrow (\mathtt{iter} - 1) \cdot s \bmod |\mathcal{A}|$
6:         $\mathbb{I}_s = \left( \boldsymbol{e}_{a_k}, \boldsymbol{e}_{a_{k+1}}, \ldots, \boldsymbol{e}_{a_{k+s-1}} \right)$
7:         $A \leftarrow (X \cdot \mathbb{I}_s)^\top \nabla F$, $B \leftarrow (X \cdot \mathbb{I}_s)^\top \mathrm{diag}(\nabla^2 F)(X \cdot \mathbb{I}_s)$
8:         **for** $\ell = 1, \ldots, s$ **do**
9: 
$$\phi_{j_\ell} \leftarrow \frac{1}{nd(\tau)} \cdot \left[ \mathbf{y}^\top X \boldsymbol{e}_{j_\ell} - A_\ell - \sum_{i=0}^{\ell-1} B_{\ell i} \Delta \beta_{j_i}^{(t-1)} \right] - \lambda(1 - \alpha)\beta_{j_\ell}^{(t-1)}$$
10: 
$$\psi_{j_\ell} \leftarrow \frac{1}{nd(\tau)} B_{\ell\ell} + \lambda(1 - \alpha)$$
11: 
$$\beta_{j_\ell}^{(t)} \leftarrow S\left( \beta_{j_\ell}^{(t-1)} + \frac{\phi_{j_\ell}}{\psi_{j_\ell}}, \frac{\lambda\alpha}{\psi_{j_\ell}} \right)$$
12:         **end for**
13:     **end for**
14:     **Convergence check**: Apply Algorithm 3
15:     **if converged, break**
16: **end for**
17: **Output:** $\boldsymbol{\beta}$, $\beta_0$

---

Here we use $\beta_{j_\ell}^{(k-1)s}$ and $\beta_0^{(k-1)s}$ in the Taylor expansion for the first $\ell - 1$ updates within the same cycle, since $\beta_{j_\ell}^{(k-1)s+\ell}$ and $\beta_0^{(k-1)s+\ell}$ remain unchanged within the block.

Next, we handle the denominator by *freezing* the Hessian term across the block, as in classical BCD. Specifically, we hold the second-order derivative $F''(\cdot)$ constant for all $s$ updates, avoiding repeated evaluation of this expensive term and thereby reducing computational overhead:

$$\psi_{j_\ell}^{((k-1)s+\ell)} = \frac{1}{n\,d(\tau)}\, \boldsymbol{e}^{(k-1)s+\ell,\top}\, X^\top\, F''\big(X\boldsymbol{\beta}^{(k-1)s}\big) \odot \big( X\, \boldsymbol{e}^{(k-1)s+\ell} \big) + \lambda(1 - \alpha).$$

Hence, the interim update is

$$\Delta \beta_{j_\ell}^{(k-1)s+\ell} = \frac{\phi_{j_\ell}^{((k-1)s+\ell)}}{\psi_{j_\ell}^{((k-1)s+\ell)}} \tag{12}$$

Incorporating the $\ell_1$-penalty leads to a soft-thresholding step. Thus, the updated coefficient:

$$\beta_{j_\ell}^{(k-1)s+\ell} \;\leftarrow\; S\!\left( \beta_{j_\ell}^{(k-1)s} + \Delta \beta_{j_\ell}^{(k-1)s+\ell}, \; \frac{\lambda\,\alpha}{\psi_j^{((k-1)s+\ell)}} \right), \tag{13}$$

where $S(z, \gamma)$ is the soft-thresholding operator. This completes one ECCD iteration for the coordinate $j_\ell$. By iterating over all $\ell$ in cyclic order (modulo $p$) for $\ell \in \{1, \ldots, s\}$, and then repeating over blocks, we obtain the full Enhanced Cyclic Coordinate Descent procedure.

For clarity, we summarize the procedure in Algorithm 1, omitting the full matrix derivation. The complete matrix-form representation and additional algorithmic variants are deferred to Appendix A.2.

## 5 Theoretical Analysis

We first bound the approximation error and then show that it yields sub-linear time while preserving a $\mathcal{O}(np)$ memory footprint. All proofs, including auxiliary lemmas and technical details, can be found in Appendix A.3, A.4, A.5.

Below, we state a simplified version of Theorem 1, which characterizes the leading-order approximation error introduced by the Taylor-expanded updates.

**Theorem 1** (Taylor-Approximate Update Error Bound). *Let the true coordinate update at iteration $(k-1)s + \ell$ be given by $\Delta\beta_{j_\ell}^{true} := \tilde{\phi}_{j_\ell}/\tilde{\psi}_{j_\ell}$, and let $\widehat{\Delta\beta_{j_\ell}} := \phi_{j_\ell}/\psi_{j_\ell}$ be the corresponding approximate update computed using a first-order Taylor expansion. Then, the Taylor approximation error is bounded as:*

$$\left|\widehat{\Delta\beta_{j_\ell}} - \Delta\beta_{j_\ell}^{true}\right| \leq C_1(X, s, f, \lambda, \alpha)\|\Delta\beta\|_\infty + C_2(X, s, f, \lambda, \alpha)\|\Delta\beta\|_\infty^2. \tag{14}$$

The denominator error is of order $O(\|\Delta\beta\|_\infty^2)$; the overall update error reusing the same $\psi_{j_\ell}$ across a block of $s$ coordinates is controlled to $O(\|\Delta\beta\|_\infty)$, preventing error accumulation across iterations.

Theorems 2 and 3 demonstrate that, in comparison to the classical coordinate descent method, our enhanced scheme achieves reduced time complexity without incurring additional space overhead.

**Theorem 2** (Time Complexity Analysis). *Consider a single epoch of the enhanced cyclic coordinate descent (ECCD) algorithm, defined as one full pass over all $p$ features. Let $C$ denote the computational cost of evaluating the mean link function $\nabla F(\cdot)$ once, so that applying it to the full linear predictor $X\boldsymbol{\beta} \in \mathbb{R}^n$ requires $\mathcal{O}(nC)$ operations. Then, by choosing the block size as $s = \sqrt{C}$, the computational cost per epoch is reduced from the standard $\mathcal{O}(npC)$ to $\mathcal{O}(np\sqrt{C})$.*

This complexity gain becomes particularly significant when $C$ is large (i.e., when the cost of evaluating the link function is high; see Section D for further discussion). In the special case $s = 1$, ECCD reduces to standard coordinate descent with complexity $\mathcal{O}(npC)$. Therefore, the square-root choice achieves an asymptotical acceleration factor of $\mathcal{O}(\sqrt{C})$. Empirically, moderate block sizes (e.g. $s \in \{8, 16\}$) often provide favorable runtime/accuracy trade-offs on medium-scale datasets.

**Theorem 3** (Space Complexity Analysis). *Let $s \leq \min\{n, p\}$. The enhanced cyclic coordinate descent (ECCD) algorithm can be implemented with a space complexity of $\mathcal{O}(np)$, which is dominated by the storage of the design matrix $X \in \mathbb{R}^{n \times p}$.*

When the block size is chosen as $s = \sqrt{C}$—which is optimal from a time complexity perspective—this remains substantially smaller than $\min\{n, p\}$ for most practical settings, since $C$ denotes a constant representing the cost of evaluating the mean link function $F'(\cdot)$. As a result, the overall memory footprint of ECCD remains $\mathcal{O}(np)$, matching that of classical coordinate descent methods.

# 6 Experiments

Unless otherwise noted, all experiments were executed on a single Cray EX node using a single OS process; the only exception is BIGLASSO, which was run in a multithreaded configuration with thread counts as reported in Tables 8 and 9. Each node was equipped with an AMD EPYC 7763 ("Milan") CPU (64 physical cores, 2.45 GHz base frequency, 256 MiB L3 cache) and 512 GiB of DDR4-3200 memory. For scalability experiments on dense datasets (Table 5), we used an AMD EPYC 7352 CPU (24 physical cores, 2.30 GHz, 128 MiB L3 cache).

Our implementation is based on R v4.3.1, utilizing the following key packages: glmnet v4.1–8, Matrix v1.7–1, stringr v1.5.1, RcppEigen v0.3.4.0.2, and Rcpp v1.0.14. All experiments were performed in double precision, with machine epsilon $\epsilon_{\text{mach}} = 2.220446 \times 10^{-16}$ on our platform.

Table 2: Relative differences of the objective function on duke.

| Dataset | $\alpha$ | Relative Differences $\Delta_s$ | | | | |
|---|---|---|---|---|---|---|
| | | $s = 2$ | $s = 4$ | $s = 8$ | $s = 16$ | $s = 32$ |
| duke | 0.1 | $4.06 \times 10^{-8}$ | $4.15 \times 10^{-8}$ | $8.12 \times 10^{-8}$ | $8.90 \times 10^{-8}$ | $2.74 \times 10^{-7}$ |
| | 0.2 | $1.84 \times 10^{-8}$ | $1.93 \times 10^{-8}$ | $1.05 \times 10^{-7}$ | $2.30 \times 10^{-7}$ | $1.96 \times 10^{-7}$ |
| | 0.5 | $1.23 \times 10^{-8}$ | $1.56 \times 10^{-8}$ | $7.31 \times 10^{-7}$ | $1.18 \times 10^{-6}$ | $1.18 \times 10^{-6}$ |
| | 0.8 | $3.50 \times 10^{-9}$ | $1.19 \times 10^{-8}$ | $5.86 \times 10^{-7}$ | $2.51 \times 10^{-6}$ | $2.51 \times 10^{-6}$ |
| | 1.0 | $2.99 \times 10^{-9}$ | $2.12 \times 10^{-7}$ | $4.78 \times 10^{-7}$ | $4.78 \times 10^{-7}$ | $4.78 \times 10^{-7}$ |

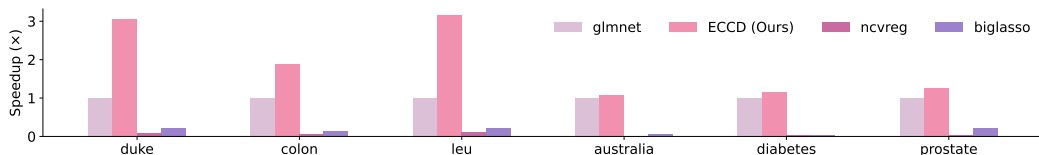

Figure 3: Comparison runtimes across six benchmark datasets ($\alpha = 0.5$). Speedups are normalized so that `glmnet` is 1.0×. ECCD ($s = 8$) consistently achieves the highest acceleration on both datasets.

## 6.1 Numerical Experiments

We evaluated ECCD against classical cyclic coordinate–descent solver for elastic-net–regularized logistic regression on standard LIBSVM benchmarks (Table 15). Unless otherwise stated, all runs were initialized at the zero vector, $\boldsymbol{\beta}^{(0)} = \mathbf{0}$. ECCD employs an active-set strategy [2], strong-rule screening [8], and periodic Karush–Kuhn–Tucker (KKT) checks to prune inactive coordinates. Performance results are averaged over 100 independent trials. Details of the synthetic-data generation protocol are provided in Appendix D.

**Convergence on ECCD**   We evaluate ECCD on the `duke` dataset using the $\lambda$-path implementation. We compare block sizes $s \in \{2, 4, 8, 16, 32\}$ against the standard coordinate update ($s = 1$), as shown in Table 2. The differential maximum relative difference $\Delta_s = \|\mathrm{obj}(s) - \mathrm{obj}(1)\|_2 / \|\mathrm{obj}(1)\|_2$ across all $\alpha$ and $s$ is $2.5 \times 10^{-6}$. For mixing ratios $\alpha \in \{0.1, 0.2\}$, $\Delta_s < 5 \times 10^{-7}$. Thus, even for large blocks, ECCD preserves solution quality up to numerical precision.

Table 3: Relative differences of the objective function on `duke`, comparing ECCD and BCD.

| | $\lambda = 0.1$ | | | $\lambda = 0.01$ | | | $\lambda = 0.001$ | |
|---|---|---|---|---|---|---|---|---|
| $s$ | **Alg** | Rel. Diff | $s$ | **Alg** | Rel. Diff | $s$ | **Alg** | Rel. Diff |
| 8 | ECCD | $1.14 \times 10^{-7}$ | 8 | ECCD | $1.19 \times 10^{-7}$ | 8 | ECCD | $1.69 \times 10^{-8}$ |
| | BCD | $7.50 \times 10^{-9}$ | | BCD | $\mathbf{1.28 \times 10^{10}}$ | | BCD | $\mathbf{5.28 \times 10^{10}}$ |
| 16 | ECCD | $5.38 \times 10^{-8}$ | 16 | ECCD | $1.41 \times 10^{-7}$ | 16 | ECCD | $9.92 \times 10^{-7}$ |
| | BCD | $1.27 \times 10^{-7}$ | | BCD | $\mathbf{5.53 \times 10^{9}}$ | | BCD | $\mathbf{1.04 \times 10^{11}}$ |
| 32 | ECCD | $8.41 \times 10^{-8}$ | 32 | ECCD | $1.26 \times 10^{-7}$ | 32 | ECCD | $4.20 \times 10^{-7}$ |
| | BCD | $\mathbf{2.53 \times 10^{9}}$ | | BCD | $\mathbf{1.03 \times 10^{10}}$ | | BCD | $\mathbf{5.79 \times 10^{10}}$ |

**Stability of Optimized ECCD with Comparison to BCD**   Table 3 shows that ECCD retains numerical stability across all block sizes and regularization strengths—the largest relative discrepancy from the $s = 1$ objective is $9.9 \times 10^{-7}$ (at $s = 16, ; \lambda = 0.001$). In contrast, BCD's errors remain small for $\lambda = 0.1$ but then explode once $\lambda \leq 0.01$ for $s \geq 8$ (reaching on the order of $10^{10}$ at $\lambda = 0.01, ; s = 8$) and diverge catastrophically at $\lambda = 0.001$ (on the order of $10^{10} \sim 10^{11}$). This pattern matches the theory: BCD's fixed-Hessian updates accumulate curvature approximation error in the denominator, which becomes ill-conditioned for small $\lambda$ and large $s$, whereas ECCD's Taylor-based correction term in the numerator controls that error and preserves stability.

## 6.2 Performance Experiments

**Block Size Selection Strategy in Practice**   While our theoretical analysis suggests that the optimal block size is $s^* = \sqrt{C}$, practical implementation requires careful estimation of the hidden constant $C$, which captures the relative cost of link function evaluations versus basic linear algebra operations. Empirically, our profiling across datasets revealed that a slightly larger block size, particularly $s = 8$, consistently delivered near-optimal speedups while preserving stability. Please refer to Appendix D for a detailed analysis on the selection strategy of block size $s$.

**Performance Experiment with Single $\lambda$**   Table 17 demonstrates that tuning the block size $s$ yields noticeable benefits, especially in high-dimensional cases ($p \gg n$). ECCD achieves up to $4.19\times$ speedups over `glmnet` on `duke`, with comparable gains on `leu` ($2.00\times$–$5.37\times$) and `colon-cancer`

Table 4: Logistic Peformance: Time (s), prediction error, and speedup under different $\alpha$.

| Filename | Method | $\alpha = 0.1$ | | | $\alpha = 0.2$ | | | $\alpha = 0.5$ | | | $\alpha = 0.8$ | | | $\alpha = 1.0$ | | |
|---|---|---|---|---|---|---|---|---|---|---|---|---|---|---|---|---|
| | | Time | Rel. Diff | Speedup | Time | Rel. Diff | Speedup | Time | Rel. Diff | Speedup | Time | Rel. Diff | Speedup | Time | Rel. Diff | Speedup |
| duke | glmnet | 6.40e-02 | – | 1.00× | 5.72e-02 | – | 1.00× | 5.30e-02 | – | 1.00× | 4.89e-02 | – | 1.00× | 4.95e-02 | – | 1.00× |
| | ncvreg | 8.29e-01 | 5.37e-01 | 0.08× | 6.88e-01 | 5.33e-01 | 0.08× | 5.80e-01 | 5.33e-01 | 0.09× | 5.55e-01 | 5.47e-01 | 0.09× | 5.45e-01 | 5.63e-01 | 0.09× |
| | biglasso | 2.37e-01 | 2.98e-01 | 0.27× | 2.29e-01 | 1.91e-01 | 0.25× | 2.26e-01 | 5.11e-02 | 0.23× | 2.28e-01 | 7.35e-03 | 0.21× | 2.59e-01 | 8.75e-06 | 0.19× |
| | Ours ($s = 8$) | 3.40e-02 | 7.91e-06 | **1.88×** | 2.46e-02 | 6.77e-06 | **2.33×** | 1.74e-02 | 5.28e-06 | **3.05×** | 1.55e-02 | 5.31e-06 | 3.16× | 2.09e-02 | 5.12e-06 | 2.37× |
| | Ours ($s = 16$) | 3.48e-02 | 8.73e-06 | 1.84× | 2.48e-02 | 8.04e-06 | 2.31× | 1.79e-02 | 7.05e-06 | 2.96× | 1.40e-02 | 7.30e-06 | **3.49×** | 1.71e-02 | 2.09e-05 | **2.89×** |
| colon | glmnet | 3.06e-02 | – | 1.00× | 2.32e-02 | – | 1.00× | 2.33e-02 | – | 1.00× | 1.92e-02 | – | 1.00× | 2.01e-02 | – | 1.00× |
| | ncvreg | 7.44e-01 | 5.05e-01 | 0.04× | 5.49e-01 | 4.96e-01 | 0.04× | 4.70e-01 | 4.82e-01 | 0.05× | 4.21e-01 | 5.06e-01 | 0.05× | 3.88e-01 | 5.14e-01 | 0.05× |
| | biglasso | 1.99e-01 | 2.56e-01 | 0.15× | 1.91e-01 | 1.58e-01 | 0.12× | 1.86e-01 | 3.86e-02 | 0.13× | 1.88e-01 | 4.77e-03 | 0.10× | 2.01e-01 | 8.14e-06 | 0.10× |
| | Ours ($s = 8$) | 3.10e-02 | 3.87e-06 | 0.99× | 2.20e-02 | 4.55e-06 | 1.05× | 1.24e-02 | 3.96e-06 | **1.88×** | 1.01e-02 | 4.32e-06 | **1.90×** | 1.43e-02 | 6.17e-06 | **1.41×** |
| | Ours ($s = 16$) | 3.18e-02 | 6.95e-06 | 0.96× | 2.12e-02 | 5.46e-06 | **1.09×** | 1.34e-02 | 3.66e-06 | 1.74× | 1.03e-02 | 6.94e-06 | 1.86× | 1.50e-02 | 3.98e-06 | 1.34× |
| leu | glmnet | 5.83e-02 | – | 1.00× | 5.17e-02 | – | 1.00× | 4.93e-02 | – | 1.00× | 4.75e-02 | – | 1.00× | 4.80e-02 | – | 1.00× |
| | ncvreg | 5.86e-01 | 5.73e-01 | 0.10× | 4.62e-01 | 5.74e-01 | 0.11× | 4.55e-01 | 5.50e-01 | 0.11× | 3.94e-01 | 5.48e-01 | 0.12× | 4.00e-01 | 5.11e-01 | 0.12× |
| | biglasso | 2.24e-01 | 3.29e-01 | 0.26× | 2.19e-01 | 2.51e-01 | 0.24× | 2.12e-01 | 1.99e-01 | 0.23× | 2.09e-01 | 1.89e-01 | 0.23× | 2.27e-01 | 1.89e-01 | 0.21× |
| | Ours ($s = 8$) | 3.33e-02 | 2.14e-05 | 1.75× | 2.45e-02 | 1.42e-05 | **2.11×** | 1.56e-02 | 6.10e-06 | **3.16×** | 1.57e-02 | 9.32e-06 | **3.03×** | 1.97e-02 | 9.13e-06 | **2.44×** |
| | Ours ($s = 16$) | 3.11e-02 | 1.10e-04 | **1.87×** | 2.49e-02 | 2.04e-05 | 2.08× | 1.56e-02 | 9.49e-06 | 3.16× | 1.58e-02 | 1.84e-05 | 3.01× | 1.98e-02 | 1.40e-05 | 2.43× |
| aus | glmnet | 8.65e-03 | – | 1.00× | 8.10e-03 | – | 1.00× | 7.67e-03 | – | 1.00× | 7.42e-03 | – | 1.00× | 7.36e-03 | – | 1.00× |
| | ncvreg | 5.43e-01 | 2.54e-01 | 0.02× | 5.42e-01 | 2.93e-01 | 0.02× | 5.49e-01 | 3.36e-01 | 0.02× | 5.40e-01 | 3.53e-01 | 0.01× | 5.35e-01 | 3.62e-01 | 0.01× |
| | biglasso | 1.60e-01 | 8.97e-02 | 0.05× | 1.67e-01 | 5.07e-02 | 0.05× | 1.65e-01 | 1.16e-02 | 0.05× | 1.65e-01 | 1.33e-03 | 0.05× | 1.59e-01 | 8.59e-05 | 0.05× |
| | Ours ($s = 8$) | 7.22e-03 | 2.16e-08 | **1.20×** | 7.52e-03 | 2.25e-08 | **1.08×** | 7.12e-03 | 2.25e-06 | **1.08×** | 7.53e-03 | 3.51e-05 | 0.99× | 7.79e-03 | 1.53e-04 | 0.95× |
| | Ours ($s = 14$) | 8.03e-03 | 2.15e-08 | 1.08× | 7.64e-03 | 2.25e-08 | 1.06× | 7.27e-03 | 2.26e-06 | 1.06× | 7.48e-03 | 3.51e-05 | 0.99× | 7.80e-03 | 1.53e-04 | 0.94× |
| diabete | glmnet | 7.54e-03 | – | 1.00× | 7.13e-03 | – | 1.00× | 7.09e-03 | – | 1.00× | 7.06e-03 | – | 1.00× | 7.49e-03 | – | 1.00× |
| | ncvreg | 2.71e-01 | 3.17e-02 | 0.03× | 2.76e-01 | 1.52e-02 | 0.03× | 2.88e-01 | 2.48e-03 | 0.03× | 2.86e-01 | 2.29e-04 | 0.03× | 3.02e-01 | 1.31e-10 | 0.02× |
| | biglasso | 1.82e-01 | 3.17e-02 | 0.04× | 1.62e-01 | 1.52e-02 | 0.04× | 1.61e-01 | 2.48e-03 | 0.04× | 1.62e-01 | 2.30e-04 | 0.04× | 1.76e-01 | 3.17e-05 | 0.04× |
| | Ours ($s = 8$) | 4.97e-03 | 2.17e-05 | **1.52×** | 5.61e-03 | 6.94e-07 | **1.27×** | 6.10e-03 | 2.75e-06 | **1.16×** | 6.06e-03 | 1.32e-05 | **1.17×** | 6.14e-03 | 2.87e-05 | **1.22×** |
| prostate | glmnet | 7.48e-02 | – | 1.00× | 7.33e-02 | – | 1.00× | 6.59e-02 | – | 1.00× | 6.53e-02 | – | 1.00× | 6.21e-02 | – | 1.00× |
| | ncvreg | 2.53e+00 | 4.40e-01 | 0.03× | 2.08e+00 | 4.33e-01 | 0.04× | 1.84e+00 | 5.06e-01 | 0.04× | 1.74e+00 | 5.09e-01 | 0.04× | 1.69e+00 | 5.15e-01 | 0.04× |
| | biglasso | 3.26e-01 | 1.96e-01 | 0.23× | 3.03e-01 | 1.03e-01 | 0.24× | 2.96e-01 | 1.98e-02 | 0.22× | 2.94e-01 | 2.19e-03 | 0.22× | 2.98e-01 | 2.09e-06 | 0.21× |
| | Ours ($s = 8$) | 9.12e-02 | 2.83e-06 | 0.82× | 7.09e-02 | 1.69e-06 | 1.03× | 5.20e-02 | 6.37e-07 | 1.27× | 4.53e-02 | 1.92e-06 | 1.44× | 4.14e-02 | 2.54e-06 | 1.50× |
| | Ours ($s = 16$) | 8.28e-02 | 3.73e-06 | 0.90× | 6.60e-02 | 2.44e-06 | **1.11×** | 5.00e-02 | 1.16e-06 | **1.32×** | 4.24e-02 | 3.15e-06 | **1.54×** | 4.02e-02 | 3.69e-06 | **1.54×** |

(up to $6.34\times$). For sample-dominated ($n \gg p$) scenarios, improvements are modest ($2.6\times$–$4.0\times$ on `diabetes-scale` and $2.8\times$-$3.4\times$ on `australian`), and ECCD is slower at larger $\lambda$ values for `phishing`, recovering to $2.19\times$ at the smallest penalty. This highlights a fundamental trade-off: block updates strongly accelerate computations in high-dimensional, strongly regularized problems, but yield smaller or even negative gains otherwise.

**Performance Experiment with the $\lambda$ Path** Table 4 demonstrates that the ECCD outperforms baseline GLM solvers-including the state-of-the-art `glmnet`-across diverse benchmark datasets with single thread under default setups of `glmnet`. ECCD consistently matches `glmnet` in predictive accuracy with relative $\ell_2$-difference of objective values below $10^{-5}$ for every dataset across the regularization path. In terms of runtime efficiency, ECCD demonstrates improvements, notably attaining up to $3.5\times$ speedup over `glmnet`. For the lower dimensional benchmarks (e.g., `australian` and `diabete`), our method shows consistent gains of $1.1$-$1.5\times$. The best speedup ($3.49\times$) is observed with block size $s = 16$ at $\alpha = 0.8$ in `duke` dataset. Notably, in the regularization-path setting, the combination of the strong rule and warm starts substantially accelerates convergence [2, 8], which makes our relative speedup appear less pronounced than in the full-model scenario.

**Scalability** We evaluate GLMNET and ECCD under identical, single-threaded settings.[3] We fix `tol = 1e-7`, precompute a length-100 $\lambda$ path in R, and pass the same sequence to both solvers with $\lambda_{\min}/\lambda_{\max} = 0.01$ when $n < p$ and $10^{-4}$ otherwise (`nlambda = 100`, `lambda.min.ratio = ifelse(nobs < nvars, 0.01, 1e-4)`). Inputs are standardized in R and in-solver standardization is disabled (`standardize = FALSE`); for GLMNET we use `type.logistic = "modified-newton"` with `intercept = TRUE`. Timing is performed on a Cray cluster across three regimes ($n < p$, $n > p$, and $n = p$). ECCD additionally reports the selected block size $s$ (Table 5).

Under matched tolerance, $\lambda$ path, and preprocessing, ECCD is consistently faster than GLMNET across all regimes (Table 5). The largest relative gains occur in the wide setting ($n > p$), where strong-rule screening plus periodic KKT checks quickly concentrate work on a small active set and block updates improve cache locality; here speedups peak around $8\times$ on mid-size problems and remain above $2\times$ even at the largest tall instance ($1\,000\,000 \times 5000$). For square problems ($n = p$), improvements are steady at roughly $3$–$4\times$ across sizes up to $50\,000 \times 50\,000$. When $p \gg n$, speedups are more modest (about $1.1$–$1.9\times$), reflecting slower active-set contraction when many features are weakly informative. In absolute terms, ECCD can reduce wall-clock time from minutes to tens of

---

[3]All runs are pinned to one thread to eliminate BLAS/OpenMP variability.

Table 5: Wall-clock time across regimes. ECCD uses the indicated block size $s$. Speedup = GLMNET time / ECCD time.

| Regime | Dataset ($n \times p$) | Time (s) | | $s$ | Speedup ($\times$) |
|---|---|---|---|---|---|
| | | GLMNET | ECCD | | |
| $n < p$ | $100 \times 1\,000\,000$ | 6.348 | 3.330 | 16 | 1.91 |
| $n < p$ | $500 \times 1\,000\,000$ | 133.606 | 113.281 | 8 | 1.18 |
| $n < p$ | $500 \times 10\,000\,000$ | 389.342 | 346.456 | 32 | 1.12 |
| $n < p$ | $10\,000 \times 50\,000$ | 127.511 | 83.490 | 4 | 1.53 |
| $n > p$ | $10\,000 \times 50$ | 0.193 | 0.093 | 4 | 2.07 |
| $n > p$ | $10\,000 \times 500$ | 3.776 | 1.080 | 4 | 3.50 |
| $n > p$ | $10\,000 \times 5000$ | 158.968 | 19.653 | 8 | 8.09 |
| $n > p$ | $1\,000\,000 \times 5000$ | 2082.622 | 1027.341 | 4 | 2.03 |
| $n = p$ | $5000 \times 5000$ | 35.061 | 8.171 | 4 | 4.29 |
| $n = p$ | $10\,000 \times 10\,000$ | 126.171 | 30.896 | 8 | 4.08 |
| $n = p$ | $20\,000 \times 20\,000$ | 661.032 | 213.412 | 4 | 3.10 |
| $n = p$ | $40\,000 \times 40\,000$ | 3369.816 | 986.692 | 4 | 3.42 |
| $n = p$ | $50\,000 \times 50\,000$ | 4956.082 | 1466.968 | 8 | 3.38 |

seconds on mid-size wide data (e.g., $10\,000 \times 5000$), while using small block sizes ($s \in \{4, 8, 16, 32\}$) that balance per-iteration cost and parallel efficiency.

**Core memory results.** Empirically, the measurements align with Theorem 3: with the active-set cap, ECCD's residual memory beyond storing $X$ is small and essentially insensitive to the block size $s$. In the square case $n=p=5000$ (Table 21), raising $s$ up to 4096 increases peak resident set size (RSS) by only $\sim 39$ MiB over baseline and then plateaus, consistent with the warm-start cap $s \approx \sqrt{|\mathcal{A}|_{\max}}$. In wide ($n=100$, $p=100{,}000$) and tall ($n=100{,}000$, $p=100$) regimes (Tables 22, 23), peak RSS is effectively flat (variation $\lesssim 40$ MiB), indicating that neither $\mathcal{O}(ns)$ nor $\mathcal{O}(s^2)$ terms dominate. By contrast, removing the cap exposes the anticipated $\mathcal{O}(s^2)$ growth: for $n=10$, $p=5000$, peak RSS increases by $\sim 129$ MiB as $s$ reaches 4096 (Table 24). Overall, the overhead scales as $ns/(np) = s/p \to 0$ in high dimensions which is negligible under the capping rule.

# 7 Conclusion

In summary, ECCD batches multiple coordinate descent updates with a tunable block size and Taylor correction, which cuts per-epoch cost without affecting convergence. Our analysis shows that setting $s^* = \sqrt{C}$ yields an $O(\sqrt{C})$ speedup, and experiments demonstrate up to $3\times$ gains on logistic and Poisson tasks. ECCD integrates warm-start and screening rules, matches leading solvers in accuracy, and naturally extends to other GLMs, structured penalties, and distributed or asynchronous settings.

# 8 Limitations

While ECCD achieves strong results, selecting an appropriate block size remains nontrivial: performance is robust across a range of $s$, but the exact optimum is hard to pinpoint.

# 9 Acknowledgment

This work was supported by the US Department of Energy, Office of Science, Advanced Scientific Computing Research program under award DE-SC-0023296. This research used resources of the National Energy Research Scientific Computing Center (NERSC), a U.S. Department of Energy Office of Science User Facility located at Lawrence Berkeley National Laboratory, operated under Contract No. DE-AC02-05CH11231 using NERSC award ASCR-ERCAP0024170. This work also used computational resources provided by the Wake Forest University (WFU) High Performance Computing Facility for testing and debugging.

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

# A  Theoretical Analysis and Matrix Derivations

## A.1  Update Rule for Block Coordinate Descent (BCD)

We set the block size equal to $s$, where $s \leq p$. Then the objective function in $t$-th iteration is given by:

$$\mathcal{L}\big(\boldsymbol{\beta}_{(block)}\big) = -\frac{1}{n}\Big[\ell\big(\boldsymbol{\beta}^{(t)}\big) + \nabla_{(block)}^{\top}\big(\boldsymbol{\beta}_{(block)} - \boldsymbol{\beta}_{(block)}^{(t)}\big)$$
$$+ \tfrac{1}{2}\big(\boldsymbol{\beta}_{(block)} - \boldsymbol{\beta}_{(block)}^{(t)}\big)^{\top} H_{(block)}\big(\boldsymbol{\beta}_{(block)} - \boldsymbol{\beta}_{(block)}^{(t)}\big)\Big]$$
$$+ \lambda\Big(\tfrac{1-\alpha}{2}\big\|\boldsymbol{\beta}_{(block)}\big\|_2^2 + \alpha\big\|\boldsymbol{\beta}_{(block)}\big\|_1\Big). \tag{15}$$

where $\nabla_{(block)}$ is the gradient of the loss function $-\frac{1}{n}\ell(\boldsymbol{\beta})$ with respect to $\boldsymbol{\beta}_{(block)}$ at $\boldsymbol{\beta}^{(t)}$. While $H_{(block)}$ is the second-order derivative (block of the Hessian matrix).

From the definitions of the first-order derivative and the second-order derivative in (3). we can express the gradient vector $\nabla_{(block)}$ and the Hessian matrix $H_{(block)}$ as follows:

$$\nabla_{(block)} = \begin{bmatrix} g_{j_1} \\ g_{j_2} \\ \vdots \\ g_{j_s} \end{bmatrix} = \sum_{i=1}^{n} \frac{(y_i - F'(\theta_i))}{d(\tau)} \begin{bmatrix} x_{i,j_1} \\ x_{i,j_2} \\ \vdots \\ x_{i,j_s} \end{bmatrix}. \tag{16}$$

$$H_{(block)} = -\sum_{i=1}^{n} \frac{F''(\theta_i)}{d(\tau)} \begin{bmatrix} x_{i,j_1}^2 & 0 & \cdots & 0 \\ 0 & x_{i,j_2}^2 & \cdots & 0 \\ \vdots & \vdots & \ddots & \vdots \\ 0 & 0 & \cdots & x_{i,j_s}^2 \end{bmatrix}. \tag{17}$$

To do block coordinate descend ,we take the derivative of objective function with respect to $\boldsymbol{\beta}_{(block)}$, we obtain:

$$\frac{\partial\mathcal{L}(\boldsymbol{\beta}_{(block)})}{\partial\boldsymbol{\beta}_{(block)}} = -\frac{1}{n}\Big[\nabla_{(block)} + H_{(block)}(\boldsymbol{\beta}_{(block)} - \boldsymbol{\beta}_{(block)}^{(t)})\Big]$$
$$+ \lambda(1-\alpha)\boldsymbol{\beta}_{(block)} + \lambda\alpha \cdot \text{sign}(\boldsymbol{\beta}_{(block)}). \tag{18}$$

Setting the gradient with respect to block $\boldsymbol{\beta}_{(block)}$ to zero and solving for the update rule:

$$\boldsymbol{\beta}_{(block)}^{(t+1)} = \Big(H_{(block)} + \lambda(1-\alpha)I_{(block)}\Big)^{-1}. \tag{19}$$

$$\Big(\frac{1}{n}\Big[\nabla_{(block)} - H_{(block)}\boldsymbol{\beta}_{(block)}^{(t)}\Big] + \lambda\alpha \cdot \text{sign}(\boldsymbol{\beta}_{(block)}^{(t+1)})\Big). \tag{20}$$

Then, for each coordinate $j$ in this block, the update rule is:

$$\beta_j^{(t+1)} = \frac{S\big(\frac{1}{n}g_j - \frac{1}{n}h_j\,\beta_j^{(t)}, \ \lambda\alpha\big)}{-\frac{1}{n}h_j + \lambda(1-\alpha)}. \tag{21}$$

The biggest difference from the coordinate descent (CD) method is that the parameter $\theta_i$ in both $g_j$ and $h_j$ is only updated once per block update, rather than at every coordinate update.

## A.2  Matrix Form Representation

In this section, we illustrate the Enhanced Cyclic Coordinate Descent updates using a matrix-based formulation. We begin by introducing notation for the coordinate-indexing scheme, the relevant design submatrices, and the first- and second-order derivative vectors:

$$\mathbb{I}_s = \big(\boldsymbol{e}_{(k-1)s}, \ \boldsymbol{e}_{(k-1)s+1}, \ \ldots, \ \boldsymbol{e}_{ks-1}\big) \in \mathbb{R}^{p\times s}, \tag{22}$$

$$X\,\mathbb{I}_s = \big(X\,\boldsymbol{e}_{(k-1)s}, \ X\,\boldsymbol{e}_{(k-1)s+1}, \ \ldots, \ X\,\boldsymbol{e}_{ks-1}\big) \in \mathbb{R}^{n\times s}, \tag{23}$$

$$\nabla F = F'\big(\beta_0 + X\,\boldsymbol{\beta}^{(k-1)s}\big) \in \mathbb{R}^{n\times 1}, \tag{24}$$

$$\nabla^2 F = F''\big(\beta_0 + X\,\boldsymbol{\beta}^{(k-1)s}\big) \in \mathbb{R}^{n\times 1}. \tag{25}$$

We first update the intercept term $\beta_0$ by subtracting the mean difference between $\mathbf{y}$ and the current predicted mean FOD. Specifically,

$$\beta_0 \;\leftarrow\; \beta_0 \;+\; \frac{\mathbf{1}_n^\top (\mathbf{y} \;-\; \nabla F)}{\mathbf{1}_n^\top \nabla^2 F}. \tag{26}$$

We next define the following quantities for the weighted linear regression associated with the current block of coordinates:

$$A = \left(X\,\mathbb{I}_s\right)^\top \nabla F \;\in\; \mathbb{R}^{s \times 1}, \quad B = \left(X\,\mathbb{I}_s\right)^\top \mathrm{diag}\!\left(\nabla^2 F\right) \left(X\,\mathbb{I}_s\right) \;\in\; \mathbb{R}^{s \times s}, \tag{27}$$

where $\mathrm{diag}(\nabla^2 F)$ is the diagonal matrix formed by the second-order derivative vector $\nabla^2 F$. For the $\ell$-th update within the current cycle ($0 \le \ell \le s$), the numerator of $\Delta\beta_{j_\ell}^{(k-1)s+\ell}$ (i.e., the update to the $((k-1)s+\ell)$-th coordinate) is given by

$$\frac{1}{n\,d(\tau)}\,\mathbf{y}^\top X\,\boldsymbol{e}_{(k-1)s+\ell} \;-\; \frac{1}{n\,d(\tau)}\,A_\ell \;-\; \frac{1}{n\,d(\tau)}\sum_{i=0}^{\ell-1} B_{\ell,i}\,\Delta\beta^{k+i} \;-\; \lambda\,(1-\alpha)\,\beta_{j_\ell}^{(k-1)s}, \tag{28}$$

and the denominator is

$$\frac{1}{n\,d(\tau)}\,B_{\ell,\ell} \;+\; \lambda\,(1-\alpha). \tag{29}$$

To incorporate the $\ell_1$-penalty, we apply a soft-thresholding step:

$$\beta_{j_\ell}^{(k-1)s+\ell} \;\leftarrow\; \frac{S\!\left(numerator,\; \lambda\,\alpha\right)}{denominator}, \tag{30}$$

where $S(\cdot,\cdot)$ is the soft-thresholding operator. This completes the matrix-based update for the $((k-1)s+\ell)$-th coordinate in the ECCD procedure. We now summarize the matrix-based updates described above into a formal algorithm (Algorithm 5) for solving GLMs with elastic net penalty at a fixed regularization level $\lambda$. In particular, the block-wise coefficient updates correspond to the matrix formulation derived in this Section.

More detailed algorithmic variants, including a full matrix-based block update procedure, are provided in Algorithm 1. In addition, we implement pathwise regularization following [8] and summarize the full procedure in Algorithm 6. All settings and subroutines follow established conventions in prior work, and are carefully adapted to fit within the proposed ECCD framework.

### A.3 Proof of Theorem 1

**Theorem 1** (Taylor-Approximate Update Error Bound). Let the true coordinate update at iteration $(k-1)s+\ell$, which corresponds to the $\ell$th coordinate in the $k$th block, be defined as $\Delta\beta_{j_\ell}^{\mathrm{true}} := \tilde{\phi}_{j_\ell}/\tilde{\psi}_{j_\ell}$. The corresponding approximate update, computed using a Taylor expansion, is given by $\widehat{\Delta\beta_{j_\ell}} := \phi_{j_\ell}/\psi_{j_\ell}$.

(A1) Let $\theta_{\max} := \|\Delta\boldsymbol{\beta}\|_\infty$ denote the maximum coordinate update within the block.

(A2) The coefficients are bounded as $|\beta_j| \le B$, for all coordinate $j$, which holds under regularization.

(A3) The residual vector $r^{(k-1)s+\ell} := F'(\mathbf{z}^{(k-1)s+\ell}) - y$ satisfies $\|r^{(k-1)s+\ell}\|_2 \le R$.

Then, the Taylor approximation error is bounded as:

$$\left|\widehat{\Delta\beta_j} - \Delta\beta_j^{\mathrm{true}}\right| \le \frac{M_3 \cdot \ell \cdot \theta_{\max} \cdot L_\infty}{\psi_{\min}^2 \cdot d(\tau)} \tag{31}$$

$$\left[\frac{R}{\sqrt{n}d(\tau)} + \lambda(1-\alpha)B + \frac{1}{2}\left(\frac{M_2}{d(\tau)} + \lambda(1-\alpha)\right)\sqrt{n}\ell \cdot \theta_{\max} \cdot L_\infty\right]. \tag{32}$$

*Proof.* For notational simplicity, we write $j$ in place of $j_\ell$, unless otherwise stated. Since the denominator terms satisfy $\psi_j, \tilde{\psi}_j \geq \psi_{\min} > 0$ (with $\psi_{\min} = \lambda(1 - \alpha)$ as a valid lower bound), we begin with:

$$\left| \widehat{\Delta \beta_j} - \Delta \beta_j^{\text{true}} \right| = \left| \frac{\phi_j}{\psi_j} - \frac{\tilde{\phi}_j}{\tilde{\psi}_j} \right| \tag{33}$$

$$\leq \frac{|\tilde{\phi}_j| \cdot |\tilde{\psi}_j - \psi_j| + |\phi_j - \tilde{\phi}_j| \cdot \psi_j}{\psi_j \cdot \tilde{\psi}_j} \tag{34}$$

$$\leq \frac{1}{\psi_{\min}^2} \left( |\tilde{\phi}_j| \cdot |\tilde{\psi}_j - \psi_j| + |\phi_j - \tilde{\phi}_j| \cdot \psi_j \right), \tag{35}$$

where

$$\phi_j = \frac{1}{nd(\tau)} X_{\cdot,j}^\top \left( F'(\mathbf{z}^{(k-1)s}) + F''(\mathbf{z}^{(k-1)s}) \circ \delta - y \right) - \lambda(1 - \alpha)\beta_j^{(k-1)s}, \tag{36}$$

$$\tilde{\phi}_j = \frac{1}{nd(\tau)} X_{\cdot,j}^\top \left( F'(\mathbf{z}^{(k-1)s+\ell}) - y \right) - \lambda(1 - \alpha)\beta_j^{(k-1)s}, \tag{37}$$

$$\psi_j = \frac{1}{nd(\tau)} X_{\cdot,j}^\top \operatorname{diag} \left( F''(\mathbf{z}^{(k-1)s}) \right) X_{\cdot,j} + \lambda(1 - \alpha), \tag{38}$$

$$\tilde{\psi}_j = \frac{1}{nd(\tau)} X_{\cdot,j}^\top \operatorname{diag} \left( F''(\mathbf{z}^{(k-1)s+\ell}) \right) X_{\cdot,j} + \lambda(1 - \alpha), \tag{39}$$

where $\mathbf{z}^{(k-1)s} = X\beta^{(k-1)s}$.

We now estimate each component. By Taylor expansion, we have: $F'(\mathbf{z}^{(k-1)s+\ell}) = F'(\mathbf{z}^{(k-1)s}) + F''(\mathbf{z}^{(k-1)s}) \circ \delta + R$, where $\forall i \in \{1, 2, \ldots, n\}$, $R_i = \frac{1}{2}f^{(3)}(\xi_i)\delta_i^2$, for some $\xi_i \in [z_i^{(k-1)s}, z_i^{(k-1)s+\ell}]$, and $\delta = \sum_{i=0}^{\ell-1} \Delta\beta^{k+i} \cdot X_{\cdot,j_i}$.

For all updated coordinates $j$, we use $L_\infty$ to denote the infinity norm bound of each column of $X$, i.e., $\|X_{\cdot,j}\|_\infty \leq L_\infty$. Then for $\mathbf{z}$ in $(k-1) + s$-th update, by using Assumption (A2), we obtain the bound:

$$\|\mathbf{z}^{(k-1)s}\|_\infty = \| \sum_{j=1}^p x_{\cdot,j} \cdot \boldsymbol{\beta}_j^{(k-1)s}\|_\infty \leq \left( \max_{i,j} |x_{ij}| \right) \cdot \|\boldsymbol{\beta}^{(k-1)s}\|_1 \tag{40}$$

$$\leq L_\infty \cdot \|\boldsymbol{\beta}^{(k-1)s}\|_1 \leq L_\infty p B. \tag{41}$$

Therefore, we assume the function $F(\cdot)$ determined by link function in the GLM is three-times differentiable, with $|F^{(3)}(z)| \leq M_3$ and $|F''(z)| \leq M_2$ for all $z$. Next, we bound the Taylor remainder:

$$\|R\|_2^2 \leq \frac{M_3^2}{4} \sum_{i=1}^n \delta_i^4 \leq \frac{M_3^2}{4} \cdot \|\delta\|_2^2 \cdot \|\delta\|_\infty^2, \quad \Rightarrow \quad \|R\|_2 \leq \frac{M_3}{2} \cdot \|\delta\|_2 \cdot \|\delta\|_\infty. \tag{42}$$

From the definition of $\delta$ and by using Assumption (A1) we have:

$$\|\delta\|_\infty \leq \sum_{i=0}^{\ell-1} |\Delta\beta^{k+i}| \cdot \|X_{\cdot,j_i}\|_\infty \leq \ell \cdot \theta_{\max} \cdot L_\infty \tag{43}$$

$$\|\delta\|_2 \leq \sum_{i=0}^{\ell-1} |\Delta\beta^{k+i}| \cdot \|X_{\cdot,j_i}\|_2 \leq \ell \cdot \theta_{\max} \cdot \sqrt{n}. \tag{44}$$

Therefore:

$$\|R\|_2 \leq \frac{M_3}{2} \cdot \ell \cdot \theta_{\max} \cdot \sqrt{n} \cdot \ell \cdot \theta_{\max} \cdot L_\infty = \frac{M_3}{2} \cdot \ell^2 \cdot \theta_{\max}^2 \cdot \sqrt{n} \cdot L_\infty. \tag{45}$$

Since the data has been standardized so that $\|X_{\cdot,j}\|_2 = \sqrt{n}$, it follows that:

$$|\phi_j - \tilde{\phi}_j| = \left| \frac{1}{nd(\tau)} X_{\cdot,j}^\top R \right| \le \frac{1}{nd(\tau)} \|X_{\cdot,j}\|_2 \cdot \|R\|_2 \le \frac{M_3}{2d(\tau)} \cdot \ell^2 \cdot \theta_{\max}^2 \cdot L_\infty. \qquad (46)$$

Next, for the denominator difference $|\tilde{\psi}_j - \psi_j|$, we again use the fact that the data has been standardized:

$$|\tilde{\psi}_j - \psi_j| = \frac{1}{nd(\tau)} \sum_{i=1}^n |F''(z_i^{(k-1)s+\ell}) - F''(z_i^{(k-1)s})| \cdot x_{ij}^2 \le \frac{M_3}{nd(\tau)} \sum_{i=1}^n |\delta_i| \cdot x_{ij}^2$$

$$\le \frac{M_3}{d(\tau)} \cdot \|\delta\|_\infty \le \frac{M_3}{d(\tau)} \cdot \ell \cdot \theta_{\max} \cdot L_\infty. \qquad (47)$$

Moreover, applying Assumptions (A2) and (A3), we obtain:

$$|\tilde{\phi}_j| = \left| \frac{1}{nd(\tau)} X_{\cdot,j}^\top (F'(\mathbf{z}^{(k-1)s+\ell}) - y) - \lambda(1-\alpha)\beta_j^{(k-1)s} \right| \le \frac{R}{\sqrt{n}d(\tau)} + \lambda(1-\alpha)B. \qquad (48)$$

And:

$$|\psi_j| \le \frac{M_2}{d(\tau)} + \lambda(1-\alpha). \qquad (49)$$

Putting all together:

$$\left| \widehat{\Delta\beta_j} - \Delta\beta_j^{\text{true}} \right| \le \frac{1}{\psi_{\min}^2} \left[ \left( \frac{R}{\sqrt{n}d(\tau)} + \lambda(1-\alpha)B \right) \cdot \frac{M_3}{d(\tau)} \cdot \ell \cdot \theta_{\max} \cdot L_\infty \right.$$

$$\left. + \left( \frac{M_2}{d(\tau)} + \lambda(1-\alpha) \right) \cdot \frac{M_3}{2d(\tau)} \cdot \ell^2 \cdot \theta_{\max}^2 \cdot \sqrt{n} \cdot L_\infty \right] \qquad (50)$$

$$= \frac{M_3 \cdot \ell \cdot \theta_{\max} \cdot L_\infty}{\psi_{\min}^2 \cdot d(\tau)} \left[ \frac{R}{\sqrt{n}d(\tau)} + \lambda(1-\alpha)B + \frac{1}{2} \left( \frac{M_2}{d(\tau)} + \lambda(1-\alpha) \right) \cdot \sqrt{n}\ell \cdot \theta_{\max} \cdot L_\infty \right]. \qquad (51)$$

This completes the proof. $\qquad\square$

### A.4 Proof of Theorem 2

**Theorem2** (Time Complexity Analysis) Consider a single epoch of the enhanced cyclic coordinate descent (ECCD) algorithm, defined as one full pass over all $p$ features. Let $C$ denote the computational cost of evaluating the mean link function $\nabla F(\cdot)$ once, so that applying it to the full linear predictor $X\beta \in \mathbb{R}^n$ requires $\mathcal{O}(nC)$ operations. Then, by choosing the block size as $s = \sqrt{C}$, the computational cost per epoch is reduced from the standard $\mathcal{O}(npC)$ to $\mathcal{O}(np\sqrt{C})$.

*Proof.* During an epoch the algorithm processes the $p$ coordinates in $\lceil p/s \rceil$ blocks of size at most $s$. For each block we (i) extract an $n \times s$ slice of $X$, (ii) update the working response $F'(X\beta + \beta_0)$, and (iii) solve a tiny weighted least-squares system of dimension $s$.

First we calculate the cost per block. By extracting the slice and forming the weighted Gram quantities requires $\mathcal{O}(ns)$ and $\mathcal{O}(ns^2)$ operations, respectively; the latter dominates when $s > 1$. Recomputing or refreshing the $\nabla F$ and $\nabla^2 F$ in link function contributes a further $\mathcal{O}(nC)$. Hence,, a single block incurs $\mathcal{O}(nC + ns^2)$ work.

Aggregating over all blocks, we multiply by the number of blocks to obtain the per-epoch cost:

$$\mathcal{O}\left( \tfrac{p}{s}[nC + ns^2] \right) = \mathcal{O}\left( \frac{npC}{s} \right) + \mathcal{O}(nps).$$

Applying the arithmetic–geometric mean inequality to the two terms, we have

$$\frac{npC}{s} + nps \ge 2np\sqrt{C}. \qquad (52)$$

This lower bound is attained when $s^* = \sqrt{C}$, yielding the minimal total cost $\mathcal{O}(np\sqrt{C})$.

**Comparison with the one-at-a-time update.** When $s = 1$ the second term collapses to $\mathcal{O}(np)$, leaving the classical $\mathcal{O}(npC)$ bound. Thus, the square-root choice reduces the asymptotic cost by a factor of $\sqrt{C}$, an advantage that widens as the link evaluation becomes more expensive.

$\square$

### A.5 Proof of Theorem 3

**Theorem** 3 (Space Complexity Analysis) Let $s \leq \min\{n, p\}$. The enhanced cyclic coordinate descent (ECCD) algorithm can be implemented with a space complexity of $\mathcal{O}(np)$, which is dominated by the storage of the design matrix $X \in \mathbb{R}^{n \times p}$.

*Proof.* The principal memory allocation is the dense design matrix $X$, whose cost is $\mathcal{O}(np)$ and, as we shall see, dominates all subsidiary structures. The response vector $\mathbf{y} \in \mathbb{R}^n$ and the current iterate $\boldsymbol{\beta} \in \mathbb{R}^p$ contribute only $\mathcal{O}(n)$ and $\mathcal{O}(p)$, respectively. At each sweep the algorithm selects an index set of size $s$ and manipulates the corresponding indicator matrix $\mathbb{I}_s \in \mathbb{R}^{p \times (s)}$. Because $\mathbb{I}_s$ is stored as an index list rather than a full column slice, its cost is merely $\mathcal{O}(s)$. Multiplying $X$ by $\mathbb{I}_s$ materialises an $n \times s$ submatrix; even if this object is formed explicitly (the pessimistic scenario), it incurs $\mathcal{O}(ns)$ additional space. Element-wise logistic weights such as $\nabla F(\cdot)$ and $\nabla^2 F(\cdot)$ are cached as length-$n$ vectors, yielding another $\mathcal{O}(n)$. Finally, the blockwise normal equations manipulate a tiny $\times s$ Gram matrix together with its right-hand side, which together add $\mathcal{O}(s^2)$.

Collecting the above, the aggregate per-epoch requirement is

$$\mathcal{O}(np) \; + \; \mathcal{O}(n) \; + \; \mathcal{O}(p) \; + \; \mathcal{O}(ns) \; + \; \mathcal{O}(s^2). \tag{53}$$

Because $s \leq \min\{n, p\}$, we have both $ns \leq np$ and $s^2 \leq np$, rendering every term other than the first negligible in the asymptotic sense. Consequently, the complexity of the working memory of the ECCD routine is $\mathcal{O}(np)$; no additional allocation beyond the data matrix alters the leading order. $\square$

## B Algorithm

### B.1 Deviance Computation and Stopping Rules

Deviance is a likelihood-based measure used to quantify the goodness-of-fit in generalized linear models. Formally, it is defined as:

$$\text{Deviance} = -2 \left[ \ell(\widehat{\boldsymbol{\beta}}) - \ell_{\text{sat}} \right],$$

where $\ell(\widehat{\boldsymbol{\beta}})$ is the log-likelihood under the fitted model, and $\ell_{\text{sat}}$ is the log-likelihood of a saturated model (which perfectly fits the data). The *null deviance* is similarly defined by replacing the fitted model with the intercept-only model. We present the specific formulas for logistic regression and Poisson regression.

**Logistic Regression.** For binary responses $y_i \in \{0, 1\}$ and predicted probabilities $p_i = \sigma(\boldsymbol{x}_i^\top \widehat{\boldsymbol{\beta}})$, the log-likelihood is:

$$\ell(\widehat{\boldsymbol{\beta}}) = \sum_{i=1}^{n} \left[ y_i \log p_i + (1 - y_i) \log(1 - p_i) \right]. \tag{54}$$

The saturated log-likelihood is:

$$\ell_{\text{sat}} = \sum_{i=1}^{n} \left[ y_i \log y_i + (1 - y_i) \log(1 - y_i) \right], \tag{55}$$

with the convention that $0 \log 0 := 0$. The log-likelihood of intercept only model uses $\bar{y}$:

$$\ell_{\text{null}} = \sum_{i=1}^{n} \left[ y_i \log \bar{y} + (1 - y_i) \log(1 - \bar{y}) \right], \quad \bar{y} = \frac{1}{n} \sum_{i=1}^{n} y_i. \tag{56}$$

**Algorithm 2** Computation of Null Deviance for Generalized Linear Models

---

**Require:** $y \in \mathbb{R}^n$, GLM family, $F(\cdot)$, $d(\tau)$
1: $\bar{y} \leftarrow \frac{1}{n} \sum_{i=1}^n y_i$; $\theta_0 \leftarrow \nabla F^{-1}(\bar{y})$
2: $\ell_{\text{null}} \leftarrow \sum_{i=1}^n [y_i \cdot \theta_0 - F(\theta_0)] / d(\tau)$
3: $\ell_{\text{sat}} \leftarrow \sum_{i=1}^n [y_i \cdot \theta_i^{\text{sat}} - F(\theta_i^{\text{sat}})] / d(\tau)$    where $F'(\theta_i^{\text{sat}}) = y_i$
4: NullDeviance $\leftarrow -2 \cdot (\ell_{\text{null}} - \ell_{\text{sat}})$
5: **return** NullDeviance

---

**Algorithm 3** Deviance-Based Block Convergence Check

---

**Require:** Current $\boldsymbol{\beta}^{(t)}$, previous $\boldsymbol{\beta}^{(t-1)}$, $F(\cdot)$, $X$, `tol`, block size`blz`, dev_0 is computed using Algorithm 2
1: $\Delta_0 \leftarrow \beta_0^{(t)} - \beta_0^{(t-1)}$; $L_0 \leftarrow (\sum_i \nabla^2 F(\beta_0 + \boldsymbol{x}_i^\top \boldsymbol{\beta}^{(t)})) \cdot \Delta_0^2$; $L_{\text{coef}} \leftarrow 0$
2: **for** each active coordinate $j$ **do**
3:     $\Delta_j \leftarrow \beta_j^{(t)} - \beta_j^{(t-1)}$; $L_j \leftarrow (\sum_i \nabla^2 F(\beta_0 + \boldsymbol{x}_i^\top \boldsymbol{\beta}^{(t)}) X_{ij}^2) \cdot \Delta_j^2$; $L_{\text{coef}} \leftarrow \max\{L_{\text{coef}}, L_j\}$
4: **end for**
5: **if** $L_0 < $ `tol` $\cdot$ dev$_0$ **or** $L_{\text{coef}} < $ `tol` $\cdot$ dev$_0$ $\cdot$ `blz` **then**
6:     **return** Converged
7: **else**
8:     **return** Not Converged
9: **end if**

---

**Poisson Regression.** For count data $y_i \in \mathbb{N}$ and $\mu_i = \exp(\boldsymbol{x}_i^\top \widehat{\boldsymbol{\beta}})$, the log-likelihood is:

$$\ell(\widehat{\boldsymbol{\beta}}) = \sum_{i=1}^n [y_i \log \mu_i - \mu_i - \log y_i!]. \tag{57}$$

The saturated log-likelihood is:

$$\ell_{\text{sat}} = \sum_{i=1}^n [y_i \log y_i - y_i - \log y_i!], \tag{58}$$

and log-likelihood of intercept only model uses overall mean $\bar{y}$.

$$\ell_{\text{null}} = \sum_{i=1}^n [y_i \log \bar{y} - \bar{y} - \log y_i!]. \tag{59}$$

The computation of the null deviance for generalized linear models is detailed in Algorithm 2, with all notation consistent with that used throughout this paper.

For the stopping rule, we follow [8] and monitor the maximum change in deviance, normalized by the null deviance. For our ECCD algorithm, we further incorporate the block size into the threshold, in order to account for the variation in deviance magnitude induced by different block sizes.

The log-likelihood is approximated via a second-order Taylor expansion, which allows us to reuse the first- and second-order quantities already computed during optimization. The detailed procedure is shown in Algorithm 3.

## B.2 Baseline: Cyclic Coordinate Descent for GLMs with Elastic Net

As a baseline, we implement the classical cyclic coordinate descent (CCD) algorithm under the generalized linear model (GLM) framework with elastic net regularization, as proposed by Friedman et al. [2]. The algorithm performs coordinate-wise updates in a cyclic fashion, equivalent to applying ECCD with a fixed block size of one, minimizing a penalized approximation to the negative log-likelihood in each step. The full procedure is summarized in Algorithm 4.

**Algorithm 4** Cyclic Coordinate Descent for GLMs with Elastic Net Penalty (Baseline)

---

**Require:** $X \in \mathbb{R}^{n \times p}$, $\mathbf{y} \in \mathbb{R}^n$, $\boldsymbol{\beta}^{(0)} \in \mathbb{R}^p$, $\beta_0^{(0)} \in \mathbb{R}$, $\lambda > 0$, $\alpha \in [0,1]$, $H \in \mathbb{N}$, $\texttt{tol} > 0$, $F(\cdot)$

1: **for** $t = 1, 2, \ldots, H$ **do**
2:     $\eta \leftarrow \beta_0^{(t-1)} + X\boldsymbol{\beta}^{(t-1)}; \nabla F \leftarrow \nabla F(\eta), \quad \nabla^2 F \leftarrow \nabla^2 F(\eta)$
3:     **(Intercept update)**:
$$\beta_0^{(t)} \leftarrow \beta_0^{(t-1)} + \frac{\mathbf{1}_n^\top (\mathbf{y} - \nabla F)}{\mathbf{1}_n^\top \nabla^2 F}$$
4:     **(Coordinate updates)**:
5:     **for** $k = 0, \ldots, p-1$ **do**
6:         $x_k \leftarrow X\boldsymbol{e}_k; A \leftarrow x_k^\top \nabla F; B \leftarrow x_k^\top \left( \nabla^2 F \odot x_k \right)$
7:         $\beta_k^{(t)} \leftarrow \dfrac{S\left( \frac{1}{n} y^\top x_k - \frac{1}{n} A + \frac{1}{n} B \beta_k^{(t-1)}, \; \lambda\alpha \right)}{\frac{1}{n} B + \lambda(1-\alpha)}$
8:     **end for**
9:     **(Convergence check)**: Apply Algorithm 3
10:    **if converged, break**
11: **end for**
12: **Output:** $\boldsymbol{\beta}^{(t)}, \beta_0^{(t)}$

---

## B.3 ECCD for GLMs: Single-Step and Pathwise Algorithms

In this section, we introduce our proposed *Enhanced Cyclic Coordinate Descent* (ECCD) algorithm for solving generalized linear models (GLMs) with elastic net regularization. We present three variants:

- **Algorithm 1** implements ECCD with a fixed active set.
- **Algorithm 5** solves the penalized GLM for a single regularization level.
- **Algorithm 6** extends ECCD to a pathwise setting with sequential strong rules for active set screening.

Our method improves computational efficiency by grouping updates in blocks and reusing gradient/Hessian terms via second-order Taylor expansion. A minor but effective optimization we adopt is to **only recompute the log-likelihood gradient, $\nabla F$,** when there is a non-zero update in $\boldsymbol{\beta}$, avoiding unnecessary recomputation in epochs with no coefficient changes.

For other standard components such as warm-start initialization, early stopping criteria, and $\lambda$-grid construction, we closely follow established practices in the paper [8] .

---
**Algorithm 5** ECCD With Single $\lambda$
---
1: **// STRONG-RULE ACTIVE SET**
2: **COMPUTE** $q_0$, $\beta_0$, $\mathcal{A}$, null deviance
3: **SET** $z \leftarrow X\boldsymbol{\beta}^{(0)}$
4: **while** Not Converged **do**
5:     **Reset** *epoch_loss*, *has_inc*
6:     **for all** Block $\mathcal{B} \subseteq \mathcal{A}$ **do**
7:         **if** *has_inc* **or** First Block **then** $\mathbf{p} \leftarrow \sigma(z + \beta_0)$
8:         **if** with_intercept **then** update $\beta_0$
9:         Update $\boldsymbol{\beta}_{\mathcal{B}}$ (Apply Algorithm 1)
10:         $\boldsymbol{\delta} \leftarrow$ change in $\boldsymbol{\beta}_{\mathcal{B}}$, $z \leftarrow z + X_{\mathcal{B}}\boldsymbol{\delta}$; Update *has_inc*, *epoch_loss*
11:     **end for**
12:     **if** First Epoch **or** Apply Algorithm 3 **then**
13:         Check the KKT condition and update $\mathcal{A}$ (see Section 3.3 and Algorithm 6 for details).
14:         **if** No Change in $\mathcal{A}$ **then**
15:             **Break**
16:         **end if**
17:     **end if**
18: **end while**
19: **RETURN** $\widehat{\boldsymbol{\beta}}$, $\widehat{\beta_0}$
---

---
**Algorithm 6** Pathwise Enhanced Cyclic Coordinate Descent (ECCD) with Sequential Strong Rule
---
**Require:** $X \in \mathbb{R}^{n \times p}$, $\mathbf{y} \in \mathbb{R}^n$, $\boldsymbol{\beta}^{(0)} \in \mathbb{R}^p$, $\beta_0^{(0)} \in \mathbb{R}$, $\{\lambda_1 > \lambda_2 > \cdots > \lambda_q\}$, $\alpha \in [0,1]$, $s, H \in \mathbb{N}$, `tol`, `ne_limit`, `rsq_max`, `sml` $> 0$, $F', F''$ for log-likelihood, null deviance $\texttt{dev}_0$.

**Ensure:** Estimated $\boldsymbol{\beta}, \beta_0$ and deviance for each $\lambda_k$

1: Set gradient: $\mathbf{r} \leftarrow \mathbf{y} - \nabla F(\beta_0^{(0)})$,    $\mathbf{X}^\top \mathbf{r} \rightarrow \texttt{grad}$
2: Compute: $\lambda_1 \leftarrow \max_j \frac{|\texttt{grad}_j|}{\alpha}$,    $\lambda_q = \epsilon\lambda_1$
3: **for** $k = 1, \ldots, q$ **do**
4:     Run Algorithm 1 with block updates on current active set $\mathcal{A}_k$, reusing $(\boldsymbol{\beta}, \beta_0)$ and $\mathcal{A}_k \leftarrow \mathcal{A}_{k-1}$ from previous $\lambda_{k-1}$ (**warm start**).
5:     **for** outer $= 1, \ldots, H$ **do**
6:         (**Strong Rule Screening**):

$$\mathcal{S}_k \leftarrow \left\{ j : \left| \mathbf{x}_j^\top (\mathbf{y} - \nabla F(\eta)) \right| \geq \alpha(2\lambda_k - \lambda_{k-1}) \right\}$$

7:         Let $\mathcal{A}_k \leftarrow \mathcal{S}_k \cup \{j : \beta_j \neq 0\}$
8:         Run Algorithm 1 on block updates with predictors in $\mathcal{A}_k$, current $\lambda = \lambda_k$
9:         (**KKT Violation Check**): For $j \notin \mathcal{A}_k$, add back if:

$$\left| \mathbf{x}_j^\top (\mathbf{y} - \nabla F(X\boldsymbol{\beta} + \beta_0)) \right| > \alpha\lambda_k$$

10:         Update $\mathcal{A}_k$ and continue outer loop if any violations
11:     **end for**
12:     Compute deviance:

$$\texttt{dev}_k \leftarrow -2\left( \ell(\boldsymbol{\beta}, \beta_0) - \ell_{\text{sat}} \right)$$

13:     Optional Early Stop if:

$$\text{(a) } |\mathcal{A}_k| > \texttt{ne\_limit}, \quad \text{(b) } R^2 > \texttt{rsq\_max}, \quad \text{(c) } \frac{\texttt{dev}_{k-1} - \texttt{dev}_k}{2 \cdot \texttt{dev}_0} < \texttt{sml}$$

14:     Warm start next $\lambda_{k+1}$ using current $(\boldsymbol{\beta}, \beta_0)$
15: **end for**
16: **return** $\boldsymbol{\beta}, \beta_0, \{\text{deviance}\}$
---

Table 6: **Runtimes on GLMNET default path** ($\lambda_{\max} \to \lambda_{\max} \times 10^{-4}$ or $10^{-2}$; 100 points). ECCD is single-thread; BIGLASSO uses the listed thread count.

| Dataset | GLMNET [s] | ECCD [s] | $s$ | BIGLASSO [s] (threads) |
|---|---|---|---|---|
| Duke | 0.117 | 0.012 | 8 | 0.215(8) |
| $10,000 \times 10,000$ | 44.204 | 31.644 | 8 | 122.114(64) |
| $100 \times 100,000$ | 0.937 | 0.640 | 4 | 1.198(16) |
| $100,000 \times 100$ | 1.862 | 0.767 | 8 | 4.372(2) |

Table 7: **Runtimes on BIGLASSO default path** ($\lambda_{\max} \to \lambda_{\max} \times 10^{-1}$; 100 points). ECCD is single-thread; BIGLASSO uses the listed thread count.

| Dataset | GLMNET [s] | ECCD [s] | $s$ | BIGLASSO [s] (threads) |
|---|---|---|---|---|
| Duke | 0.076 | 0.010 | 8 | 0.197(8) |
| $10,000 \times 10,000$ | 4.115 | 2.250 | 4 | 1.419(64) |
| $100 \times 100,000$ | 0.942 | 0.709 | 4 | 0.890(8) |
| $100,000 \times 100$ | 1.033 | 0.484 | 16 | 1.519(64) |

## C Benchmarking Experiments with Additional Baselines

While the most direct baseline to the ECCD is the GLMNet that explore a path-wise solution with a single-thread focus. There are many popular softwares that involves solving the LASSO, ElasticNet problem for regularized logistic problems, despite differences in solution type (i.e. single lambda fit), convergence criterion[28, 29], and system-level design [30, 4]. Yet we still select some of the method serves as baselines for comparison because we expect the performance to be consistent across various of domains. Below we perform intensive comparison with other baselines:

### C.1 BigLasso

BIGLASSO[31, 30, 4] is a parallel coordinate–descent solver that targets the same objective and design as GLMNET; in the single–thread limit it reproduces GLMNET's solution. Motivated by the reviewer's suggestion, we benchmarked BIGLASSO on a Cray EX cluster using $\{2, 4, 8, 16, 32, 64\}$ threads and compared against *single-thread* ECCD. We report two settings: (i) GLMNET's default path ($\lambda_{\max} \to \lambda_{\max} \times 10^{-4}$ or $10^{-2}$, 100 points), where ECCD remains faster than BIGLASSO even when the latter uses up to 64 threads (Table 6); and (ii) BIGLASSO's shorter default path ($\lambda_{\max} \to \lambda_{\max} \times 10^{-1}$, 100 points), where ECCD is still faster on three of four datasets while BIGLASSO wins on the $10k \times 10k$ case (Table 7). These results indicate that ECCD's pathwise, active-set strategy provides a strong sequential baseline; parallelizing ECCD is straightforward future work and would further widen the gap.

### C.2 SKGLM

We include a comparison against SKGLM, a Python library that provides pathwise coordinate descent primarily for linear models and exposes logistic regression as a *single-$\lambda$* solver at the time of our experiments. Because a path solver for logistic regression is not available in SKGLM, a direct path-to-path comparison with ECCD is not possible. To enable a controlled evaluation, we reuse the $\lambda$-sequence generated for ECCD and invoke SKGLM's logistic solver at those values, matching preprocessing (standardization) and error tolerances. Runs are conducted under the same BenchOpt protocol as in Section C.4; wall-clock time is reported.

Table 10 summarizes the results. ECCD attains substantially lower runtimes across all datasets. On *Duke*, ECCD completes in $0.013$ s versus $1.375$ s for SKGLM ($\approx 106\times$ faster); on *Diabetes* the gap widens to $0.002$ s versus $6.155$ s ($\approx 3078\times$). The advantage persists in large synthetic regimes: for a tall design ($n=100$, $p=100,000$), ECCD achieves $1.426$ s versus $131.340$ s ($\approx 92\times$); for a wide design ($n=100,000$, $p=100$), $1.032$ s versus $213.012$ s ($\approx 206\times$). These findings are consistent with the expected behavior of pathwise methods: ECCD leverages warm starts, strong rules, and block updates with active-set screening to amortize work along the sequence, whereas repeatedly solving single points lacks these path-specific accelerations.

Table 8: **Table A** — BigLasso sweep grouped as 1/2/4/8 and 16/32/64/128. Times in seconds.

| Dataset | GLMNet | ECCD | BIGLASSO (threads) 1/2/4/8 | BIGLASSO (threads) 16/32/64/128 |
|---|---|---|---|---|
| Duke | 1.375 | 0.012 | 0.330 / 0.265 / 0.226 / 0.215 | 0.221 / 0.228 / 0.276 / 0.293 |
| 10k×10k | 43.807 | 32.477 | 127.226 / 123.642 / 123.034 / 122.464 | 123.462 / 126.189 / 124.638 / 123.583 |
| 100×100k | 131.340 | 0.640 | 2.919 / 1.755 / 1.340 / 1.245 | 1.198 / 1.253 / 1.306 / 1.251 |
| 100k×100 | 213.012 | 0.767 | 4.501 / 4.372 / 4.415 / 4.518 | 4.822 / 4.995 / 4.928 / 5.038 |

Table 9: **Table B** — BigLasso sweep grouped as 1/2/4/8 and 16/32/64/128. Times in seconds.

| Dataset | GLMNet | ECCD | BIGLASSO (threads) 1/2/4/8 | BIGLASSO (threads) 16/32/64/128 |
|---|---|---|---|---|
| Duke | 0.076 | 0.010 | 0.230 / 0.289 / 0.203 / 0.197 | 0.206 / 0.211 / 0.215 / 0.280 |
| 10k×10k | 4.196 | 2.263 | 9.944 / 5.557 / 3.176 / 2.287 | 2.017 / 2.189 / 1.902 / 1.729 |
| 100×100k | 0.942 | 0.709 | 2.923 / 1.281 / 0.980 / 0.890 | 0.913 / 0.927 / 0.971 / 1.011 |
| 100k×100 | 1.033 | 0.484 | 2.066 / 1.744 / 1.553 / 1.519 | 1.632 / 1.778 / 1.815 / 1.659 |

Taken together, the results indicate that ECCD provides a strong sequential baseline for logistic regression paths-even against highly optimized Python implementations, while remaining competitive at individual $\lambda$ values 4. We emphasize that this comparison isolates the benefit of path-aware computations: SKGLM is competitive for single-point fits, but without a logistic path solver it incurs near-linear cost in the number of $\lambda$ values.

### C.3  ABESS

We also compare against ABESS, which solves best-subset selection with an explicit $\ell_0$ constraint-a different objective from the $\ell_1/\ell_2$-regularized problems (Lasso/Elastic Net) considered here-so direct pathwise equivalence does not hold. Nevertheless, using the correlated synthetic generator from Section D with $\rho \in \{0.1, 0.9\}$, we benchmark small ($n=20, p=2000$), moderate ($n=100, p=1000$; $n=1000, p=100$), and large ($n=100,000, p=100$) regimes. Table 11 shows that ECCD is consistently fastest across settings, including high-correlation cases; ABESS slows substantially as $n$ grows, while ECCD maintains sub-second to single-second times under identical preprocessing and tolerance. These findings reinforce that ECCD retains its advantage even in regimes where $\ell_0$-based methods are often promoted.[4]

### C.4  More Experiment on Skglm Family with BenchOPT for Comparison

Recently the BenchOpt [32] emerges as an strong baseline for generalized linear models with convex/nonconvex regularization. Thus we augmented our baselines with the *skglm* family-SKGLM, CELER, and BLITZL1-using BenchOpt. These packages are strong sparse estimators at a *single* regularization level but differ from ECCD/GLMNET in two substantive respects: (i) they are primarily engineered for *single-point* fitting rather than full regularization paths; and (ii) their model coverage differs (e.g., Elastic Net and/or GLM path solvers are not uniformly exposed). To ensure fairness and reproducibility, we therefore report results in two regimes: *single-$\lambda$* versus *sequential 100-$\lambda$ path*.

**Setups.** All experiments use BenchOpt with package *defaults* unless otherwise noted: `skglm.SparseLogisticRegression()`, `celer.LogisticRegression()`, and `blitzl1.LogRegProblem()`. Single–$\lambda$ evaluations use three levels $\lambda/\lambda_{\max} \in \{0.5, 0.1, 0.05\}$; path experiments use a 100–$\lambda$ sequence. We retain each solver's default stopping rule and enable warm starts wherever available (BLITZL1 reuses previous solutions; CELER and SKGLM expose warm starts). ECCD and GLMNET use warm starts *and* strong rules, which are known to be decisive for

---

[4]On $n=100$, $p=100,000$ with $\rho \in \{0.1, 0.9\}$, data generation process runs exceeded 12 hours without completion on our cluster.

Table 10: ECCD vs. SKGLM (logistic; times in seconds). ECCD uses block size $s=4$.

| Dataset | SKGLM [s] | ECCD [s] |
|---|---|---|
| Duke | 1.375 | 0.013 |
| Diabetes | 6.155 | 0.002 |
| $n=100$, $p=100,000$ | 131.340 | 1.426 |
| $n=100,000$, $p=100$ | 213.012 | 1.032 |

Table 11: ECCD vs. GLMNET vs. ABESS (times in seconds) on correlated synthetic data and *Duke*. ECCD uses block size $s=4$.

| Dataset | GLMNET [s] | ABESS [s] | ECCD [s] |
|---|---|---|---|
| $n=20$, $p=2000$, $\rho=0.1$ | 0.014 | 0.011 | 0.002 |
| $n=20$, $p=2000$, $\rho=0.9$ | 0.018 | 0.011 | 0.002 |
| $n=2000$, $p=20$, $\rho=0.1$ | 0.030 | 0.426 | 0.013 |
| $n=2000$, $p=20$, $\rho=0.9$ | 0.047 | 0.493 | 0.027 |
| $n=100$, $p=1000$, $\rho=0.1$ | 0.028 | 0.026 | 0.005 |
| $n=100$, $p=1000$, $\rho=0.9$ | 0.066 | 0.028 | 0.006 |
| $n=1000$, $p=100$, $\rho=0.1$ | 0.021 | 0.131 | 0.007 |
| $n=1000$, $p=100$, $\rho=0.9$ | 0.026 | 0.128 | 0.006 |
| $n=100,000$, $p=100$, $\rho=0.1$ | 2.384 | 904.368 | 1.426 |
| $n=100,000$, $p=100$, $\rho=0.9$ | 3.840 | 931.303 | 1.032 |
| Duke | 0.022 | 0.025 | 0.013 |

path efficiency. Inputs are standardized upstream of the solvers; timings are wall-clock seconds. A dash (—) denotes unsupported model/setting combinations (e.g., Elastic Net for BLITZL1/CELER). [5]

**Findings.** (i) In the *single–λ* regime, Python sparse solvers are competitive and can be fastest on individual instances; ECCD remains consistently strong across datasets (Tables 12–13). (ii) In the *100–λ* path regime, ECCD/GLMNET are consistently faster—often by one to two orders of magnitude—because the combination of warm starts and strong rules mitigates the near-linear cost growth with the number of λ values (Table 14). (iii) These trends are robust across real (Duke, Leu, Diabetes) and synthetic (tall Syn1 and wide Syn2) settings. ECCD's block updates with active-set screening yield stable wins in path mode while remaining competitive in single-point mode.

**Takeaways.** **(A)** In *single-λ* mode, SKGLM/BLITZL1 are competitive on small dense problems and sometimes match or beat GLMNET; ECCD remains in the same ballpark (often faster) across real and synthetic datasets. **(B)** In *path* mode, ECCD and GLMNET dominate across all datasets because strong rules prune inactive coordinates early, so the effective work per λ shrinks markedly. Warm starts alone (as in SKGLM/CELER/BLITZL1) reduce constants but still scale roughly with the number of λ values. **(C)** Elastic Net support is uneven in the Python sparse packages; results shown reflect what each library exposes under BenchOpt. Overall, these experiments corroborate our central claim: ECCD provides a *pathwise* solver that is both competitive at single points and substantially faster on full paths—the regime most relevant to model selection and real workflows.

# D Additional Experiment Result

**Synthetic data generation.** We generate datasets with $n$ observations and $p$ predictors by first drawing rows of the design matrix $X \in \mathbb{R}^{n \times p}$ i.i.d. from a zero-mean Gaussian with an equi-correlated covariance,

$$X_{i\cdot} \sim \mathcal{N}_p(0, \Sigma), \qquad \Sigma_{jj} = 1, \ \Sigma_{jk} = \rho \ (j \neq k),$$

which imposes a controlled, tunable correlation among features (the special case $\rho = 0$ recovers independence). We highlight this family because it explicitly produces correlated designs and is therefore a relevant stress test for the ABESS baseline. To induce sparsity, we construct $\beta^\star \in \mathbb{R}^p$ by

---

[5]Objective values (not shown) matched within $10^{-3}$ across methods in our runs, except that BLITZL1 occasionally reported a slightly different objective. We followed the BenchOpt reference code verbatim.

Table 12: **Single-λ** runtime (s), **Lasso**. Each cell lists $\lambda/\lambda_{\max} \in \{0.5, 0.1, 0.05\}$ from top to bottom.

| Dataset | SKGLM (0.5 / 0.1 / 0.05) | CELER (0.5 / 0.1 / 0.05) | BLITZL1 (0.5 / 0.1 / 0.05) | GLMNET (0.5 / 0.1 / 0.05) | **ECCD** (0.5 / 0.1 / 0.05) |
|---|---|---|---|---|---|
| Duke | $1.8 \times 10^{-3}$ | $1.6 \times 10^{-2}$ | $2.3 \times 10^{-3}$ | $5.5 \times 10^{-3}$ | $2.5 \times 10^{-3}$ |
|  | $2.3 \times 10^{-3}$ | $3.0 \times 10^{-2}$ | $5.6 \times 10^{-3}$ | $7.8 \times 10^{-3}$ | $5.0 \times 10^{-3}$ |
|  | $2.3 \times 10^{-3}$ | $3.5 \times 10^{-2}$ | $6.5 \times 10^{-3}$ | $1.4 \times 10^{-2}$ | $6.2 \times 10^{-3}$ |
| Leu | $2.2 \times 10^{-3}$ | $1.2 \times 10^{-2}$ | $2.1 \times 10^{-3}$ | $5.2 \times 10^{-3}$ | $2.1 \times 10^{-3}$ |
|  | $2.3 \times 10^{-3}$ | $2.3 \times 10^{-2}$ | $4.4 \times 10^{-3}$ | $7.3 \times 10^{-3}$ | $4.3 \times 10^{-3}$ |
|  | $2.3 \times 10^{-3}$ | $2.2 \times 10^{-2}$ | $5.7 \times 10^{-3}$ | $7.7 \times 10^{-3}$ | $4.7 \times 10^{-3}$ |
| Diabetes | $1.6 \times 10^{-3}$ | $3.6 \times 10^{-3}$ | $8.0 \times 10^{-4}$ | $1.3 \times 10^{-3}$ | $4.0 \times 10^{-5}$ |
|  | $1.6 \times 10^{-3}$ | $4.9 \times 10^{-3}$ | $6.0 \times 10^{-4}$ | $1.6 \times 10^{-3}$ | $4.7 \times 10^{-4}$ |
|  | $1.6 \times 10^{-3}$ | $5.0 \times 10^{-3}$ | $5.0 \times 10^{-4}$ | $1.8 \times 10^{-3}$ | $5.6 \times 10^{-4}$ |
| Syn1 (100×100k) | $3.8 \times 10^{-2}$ | $1.1 \times 10^{-1}$ | $1.0 \times 10^{-1}$ | $1.0 \times 10^{-1}$ | $6.2 \times 10^{-2}$ |
|  | $3.7 \times 10^{-2}$ | $3.2 \times 10^{-1}$ | $2.2 \times 10^{-1}$ | $1.2 \times 10^{-1}$ | $7.7 \times 10^{-2}$ |
|  | $3.7 \times 10^{-2}$ | $3.4 \times 10^{-1}$ | $2.7 \times 10^{-1}$ | $1.5 \times 10^{-1}$ | $8.8 \times 10^{-2}$ |
| Syn2 (100k×100) | $2.9 \times 10^{-1}$ | $6.8 \times 10^{-1}$ | $1.8 \times 10^{-1}$ | $1.1 \times 10^{-1}$ | $6.2 \times 10^{-2}$ |
|  | $9.0 \times 10^{-1}$ | $5.4 \times 10^{-1}$ | $2.8 \times 10^{-1}$ | $1.1 \times 10^{-1}$ | $6.4 \times 10^{-2}$ |
|  | $2.5 \times 10^{-1}$ | $5.6 \times 10^{-1}$ | $2.9 \times 10^{-1}$ | $1.3 \times 10^{-1}$ | $8.4 \times 10^{-2}$ |

Table 13: **Single-λ** runtime (s), **Elastic Net** ($\alpha = 0.5$). Each cell lists $\lambda/\lambda_{\max} \in \{0.5, 0.1, 0.05\}$ from top to bottom. "—" = unsupported.

| Dataset | SKGLM (0.5 / 0.1 / 0.05) | CELER (0.5 / 0.1 / 0.05) | BLITZL1 (0.5 / 0.1 / 0.05) | GLMNET (0.5 / 0.1 / 0.05) | **ECCD** (0.5 / 0.1 / 0.05) |
|---|---|---|---|---|---|
| Duke | $2.3 \times 10^{-3}$ | — | — | $5.1 \times 10^{-3}$ | $2.4 \times 10^{-3}$ |
|  | $2.6 \times 10^{-3}$ |  |  | $8.0 \times 10^{-3}$ | $4.5 \times 10^{-3}$ |
|  | $2.4 \times 10^{-3}$ |  |  | $1.1 \times 10^{-2}$ | $6.7 \times 10^{-3}$ |
| Leu | $2.2 \times 10^{-3}$ | — | — | $4.8 \times 10^{-3}$ | $2.2 \times 10^{-3}$ |
|  | $2.2 \times 10^{-3}$ |  |  | $7.6 \times 10^{-3}$ | $4.0 \times 10^{-3}$ |
|  | $2.3 \times 10^{-3}$ |  |  | $9.6 \times 10^{-3}$ | $5.2 \times 10^{-3}$ |
| Diabetes | $1.6 \times 10^{-3}$ | — | — | $1.1 \times 10^{-3}$ | $6.0 \times 10^{-5}$ |
|  | $1.6 \times 10^{-3}$ |  |  | $1.5 \times 10^{-3}$ | $4.7 \times 10^{-4}$ |
|  | $1.5 \times 10^{-3}$ |  |  | $1.3 \times 10^{-3}$ | $5.7 \times 10^{-4}$ |
| Syn1 (100×100k) | $6.0 \times 10^{-2}$ | — | — | $1.0 \times 10^{-1}$ | $6.1 \times 10^{-2}$ |
|  | $3.9 \times 10^{-2}$ |  |  | $1.3 \times 10^{-1}$ | $7.6 \times 10^{-2}$ |
|  | $4.0 \times 10^{-2}$ |  |  | $1.5 \times 10^{-1}$ | $1.0 \times 10^{-1}$ |
| Syn2 (100k×100) | $2.6 \times 10^{-1}$ | — | — | $1.0 \times 10^{-1}$ | $5.6 \times 10^{-2}$ |
|  | $3.5 \times 10^{-1}$ |  |  | $1.1 \times 10^{-1}$ | $6.5 \times 10^{-2}$ |
|  | $2.4 \times 10^{-1}$ |  |  | $1.2 \times 10^{-1}$ | $8.3 \times 10^{-2}$ |

selecting $s$ indices uniformly at random and assigning them values drawn uniformly from a fixed interval (e.g., $[1, 2]$), leaving all other entries zero. Given $X$ and $\beta^\star$, we form the linear predictor $\eta = X\beta^\star$, convert to success probabilities via the logistic link

$$p_i = \frac{1}{1 + e^{-\eta_i}}, \quad i = 1, \ldots, n,$$

and finally sample the binary response as $y_i \sim \text{Bernoulli}(p_i)$.

**Experiment Dataset Summary.** We evaluate ECCD and GLMNet on a diverse set of eight publicly available benchmarks drawn from applications in genomics, credit modeling, biomedical studies, and web security. These include a couple of small-sample, high-feature gene-expression sets, an RNA-seq dataset with an extreme feature-to-sample ratio, several moderate-scale classification tasks (e.g. credit approval and medical diagnostics), and a larger web phishing collection. By covering a broad range of dimensionalities and aspect ratios, this suite stresses both numerical stability and computational performance under the varied conditions practitioners commonly face.

Table 14: **Sequential 100-$\lambda$ path runtime (seconds)** via BenchOpt. ECCD entries include the block size $s$ used. A dash (—) indicates unsupported.

| Method | Model | Duke | | Leu | | Diabetes | | Syn1 | | Syn2 | |
|--------|-------|------|------|------|------|----------|------|------|------|------|------|
| | | Lasso | ENet | Lasso | ENet | Lasso | ENet | Lasso | ENet | Lasso | ENet |
| SKGLM | Lasso/ENet | 2.8e-1 | 2.9e-1 | 5.5 | 7.2 | 1.4e-1 | 1.4e-1 | 6.2 | 4.1 | 5.3e1 | 5.8e1 |
| CELER | Lasso | 4.1 | — | 2.4 | — | 3.0e-1 | — | 2.2e1 | — | 5.6e1 | — |
| BLITZL1 | Lasso | 4.5e-1 | — | 2.3e-1 | — | 4.1e-2 | — | 2.1 | — | 1.5e1 | — |
| GLMNET | Lasso/ENet | 2.1e-2 | 4.9e-2 | 2.0e-2 | 5.9e-2 | 6.0e-3 | 6.0e-3 | 4.7e-1 | 4.8e-1 | 1.7 | 2.0 |
| **ECCD** | Lasso/ENet | 1.3e-2 (s=32) | 2.2e-2 (s=8) | 1.0e-2 (s=16) | 1.8e-2 (s=16) | 4.0e-3 (s=8) | 4.0e-3 (s=8) | 2.7e-1 (s=32) | 2.9e-1 (s=32) | 1.6 (s=4) | 1.6 (s=4) |

Table 15: Summary of datasets used for numerical tests.

| Filename | # Observations (n) | # Features (p) |
|----------|--------------------|-----------------|
| duke | 44 | 7,129 |
| leukemia | 38 | 7,129 |
| colon-cancer | 63 | 2,000 |
| diabete-scale | 768 | 8 |
| australian | 690 | 14 |
| airway | 8 | 63679 |
| prostate | 102 | 6033 |
| phishing | 11,055 | 68 |

**Implementation details and baselines.** To ensure a fair comparison with the state-of-the-art, we align our path-tracking experiments exactly to GLMNet's settings. In particular, we use GLM-Net's default convergence threshold (`thresh` $= 10^{-7}$), intercept fitting, and grid of 100 $\lambda$s with $\lambda_{\min}/\lambda_{\max} = 10^{-2}$. All input features are pre-standardized by us, and we disable GLMNet's internal standardization to avoid double-scaling. For the single lambda case, we simply set the lambda parameter euqals given lambda input, and set the path parameter equals false.

While we also benchmark biglasso (v1.6.1, GPL-3 license) and ncvreg (v3.15.0, GPL-3 license), GLMNet remains the de facto standard for elastic-net penalized GLMs, so we continue to report GLMNet's solutions as our canonical baseline. GLMNet (v4.1-8, GPL-2.0 license) remains the most widely used coordinate-descent engine for elastic-net–penalized generalized linear models and enforces a stricter stopping rule than these alternatives. We ran all methods with their latest CRAN versions, always matching sparsity parameters $\alpha$ and, where possible, using their fastest safe defaults.

**Estimating the per-operation cost $C$.** To understand the relative cost of estimating the expensive first and second order derivative of the CGF (e.g. sigmoid) versus a simple arithmetic update, we measure the time per call and form the ratio

$$C = \frac{\text{time}(\nabla F(x))}{\text{time}(x + b)}.$$

We perform one complementary experiments at the vector level (to mimic typical linear-algebra kernels) and report the result on our HPC node.

All performance experiments were carried out on a Cray compute node (Intel Xeon 6248R @ 3.0 GHz, 192 GB RAM) running SUSE Linux Enterprise Server 15 SP5 with the vendor-provided multi-threaded LibSCI BLAS/LAPACK (libsci_gnu_123_mp, LAPACK 3.10.1). We used R 4.3.1 (built under glibc) with C++ extensions compiled via Rcpp 1.0.10 and RcppEigen 0.3.3 under `-O3 -march=native -ffast-math`. Vectorized kernels leveraged Eigen's dense operations (on dimensions up to 100), while scalar tests invoked plain C++ loops for $\exp$ and addition.

Using a microbenchmark harness in R, we call two routines in each iteration: (1) a sigmoid on a length–100 Eigen vector, (2) a pure elementwise addition of a constant. We fix $n = 10^5$ iterations and repeat each measurement 100 times. This setup approximates the hotspot cost in block-update kernels on a high-performance system.

Table 16: ECCD convergence on *Duke* under adaptive vs. fixed block sizes $s$. Entries are relative objective differences (smaller is better).

| Block size $s$ | Rel. Obj. Diff (Adaptive) | Rel. Obj. Diff (Fixed) |
|---|---|---|
| 1 | $1.38 \times 10^{-4}$ | $1.38 \times 10^{-4}$ |
| 16 | $8.025 \times 10^{-5}$ | $9.810 \times 10^{-5}$ |
| 64 | $8.674 \times 10^{-5}$ | $1.598 \times 10^{-4}$ |
| 256 | $8.674 \times 10^{-5}$ | $1.240 \times 10^{-4}$ |
| 2048 | $8.674 \times 10^{-5}$ | $1.218 \times 10^{-4}$ |
| 7129 | $8.674 \times 10^{-5}$ | $1.470 \times 10^{-4}$ |

Across both setups we observe that computing the sigmoid is roughly $13\times \sim 17\times$ more expensive than a simple add, depending on argument range and vector-width effects.

**Block Size Selection Strategy in Practice**     While our theoretical analysis suggests that the optimal block size is $s^* = \sqrt{C}$, practical implementation requires careful estimation of the hidden constant $C$, which captures the relative cost of link function evaluations versus basic linear algebra operations. To empirically estimate $C$, we benchmarked the cluster environment under two settings: (i) fully optimized Eigen-based matrix operations, and (ii) vanilla implementations involving repeated sigmoid evaluations and summations. Across both settings, we consistently observed that, in logistic regression, sigmoid function evaluations were in average $13\times$ to $17\times$ more expensive than basic matrix-vector multiplications. This implies a theoretical optimal block size of $s^* = \sqrt{C} \approx 4$. However, in practice, additional considerations arise. Notably, ECCD avoids redundant sigmoid function updates when coefficient increments are negligible, which slightly decouples actual update frequency from worst-case complexity bounds. Moreover, secondary factors—including the need to explicitly compute the second derivative of the sigmoid function, hardware-level vectorization effects, and dynamic fluctuations in the active set size—can further shift the empirical optimal block size. Our profiling across datasets revealed that a slightly larger block size, particularly $s = 8$, consistently delivered near-optimal speedups while preserving stability across various scaled tasks. Therefore, although identifying the precise optimum can be challenging, $s=8$ emerges as a robust default that achieves near-optimal speedups and beats most baselines across varied scales.

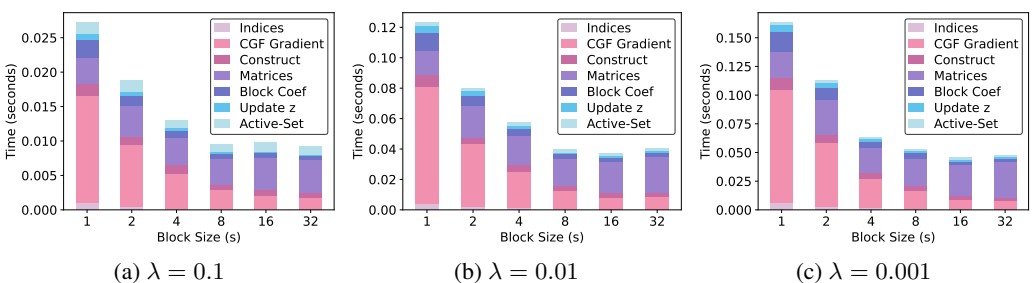

     (a) $\lambda = 0.1$            (b) $\lambda = 0.01$            (c) $\lambda = 0.001$

Figure 4: ECCD Time Decomposition Plot (`Duke`)

**Convergence of ECCD with Different Blocksize Selection.**     To illustrate the convergence behavior of ECCD under different blocksize selection, we extended ECCD to block sizes up to $s = p = 7129$ on the Duke dataset (the maximum possible blocksize for this dataset); results in Table 16 show stable convergence even at $s = p$, underscoring the robustness of our second-order Taylor correction and its advantage over naïve block coordinate descent (cf. main-paper Table 3). Column 2 reports relative objective differences under our *adaptive* active-set scheme: updates are restricted to currently nonzero coefficients, KKT violations trigger working-set expansion, and when $s > |\mathcal{A}|$ we cap the working block by setting $s \leftarrow \lfloor \sqrt{|\mathcal{A}|} \rfloor$ to avoid the $s^2$ memory/compute spike. Column 3 gives the same metric with *fixed* $s$ (no resizing), providing a direct comparison. Across all $s$, the adaptive variant matches or improves the objective and remains numerically stable through $s = 7129$. Consistent with our theory, the practically optimal $s$ is a small constant determined by link-function cost, so our original range $s \in [2, 32]$ is both relevant and justified; the large-$s$ study is included for completeness.

**Empirical Cost Breakdown** Figure 4 reports the per-epoch wall-clock composition of ECCD on the Duke dataset for three regularization strengths. For the baseline update schedule ($s = 1$), evaluation of the cumulant-generating-function (CGF) gradient (pink) and matrix-update operations (purple) together consume over 80% of total runtime; coefficient-batch updates (dark blue) and active-set maintenance (teal) also contribute nontrivially. As the block size $s$ grows, we observe:

- A roughly $1/s$ reduction in CGF-gradient cost, confirming that ECCD's block-amortization defers expensive derivative evaluations until after each batch of $s$ coordinate updates.
- A relative increase in the share of matrix-construction and coefficient-batch costs (dark purple and dark blue), since these are still performed once per block.
- Matrix-update overhead and active-set maintenance become nearly negligible at larger $s$, validating that their amortization is effective.

Across all three $\lambda$ settings, the absolute time spent on each component drops by approximately 75% when moving from $s = 1$ to $s = 32$, aligning tightly with our theoretical $O(np/s + p)$ complexity for sigmoid-evaluation–dominated workloads. ECCD thus shifts the bottleneck away from per-coordinate derivative calls toward block-level matrix operations, unlocking substantial runtime savings.

**Memory complexity.** Theorem 3 provides a detailed space–complexity analysis of ECCD. Here we extend our experiments to verify this claim, explicitly accounting for the additional memory required by ECCD beyond the storage cost of the design matrix. Concretely, ECCD allocates $\mathcal{O}(ns)$ memory for the working block $X_s$ and $\mathcal{O}(s^2)$ for its Gram submatrix, where the ideal block size $s$ is a small constant that is approximately independent of $n$ and $p$. For comparison, GLMNET corresponds to the special case $s = 1$ from the standpoint of asymptotic memory. Consequently, in high-dimensional regimes with $p \gg s$, the relative overhead of ECCD scales as $ns/(np) = s/p \to 0$ and is therefore negligible. This behavior is consistent with our empirical observations.

In all experiments we report resident set size (RSS) and track peak usage during a regularization path. To better isolate ECCD's overhead, we distinguish (i) the baseline RSS at process start, (ii) the peak RSS during optimization, and (iii) the end RSS. Because absolute baselines can vary slightly across runs and environments, the $s=1$ row in each table serves as the reference point within that experiment.

With the active-step capping mechanism, the extra storage in ECCD is dominated by $s$ vectors of length $n$ (the working block $X_s$) and an $s \times s$ Gram submatrix. In the extreme where $s$ approaches the size of the active set, the $\mathcal{O}(s^2)$ term can dominate. To prevent this, we cap the working block by dynamically setting $s \leftarrow \lfloor \sqrt{|\mathcal{A}|} \rfloor$ once the active set reaches size $|\mathcal{A}|$. This ensures that further increases in $|\mathcal{A}|$ do not cause quadratic growth in memory, keeping ECCD's overhead negligible relative to the design-matrix storage. Without the active-set cap, the $\mathcal{O}(s^2)$ term becomes dominant for large $s$, and peak RSS rises accordingly (Table 24). This confirms the necessity and effectiveness of the capping rule in practice.

Using the solution at the previous $\lambda$ as a warm start, peak RSS grows only until $s$ reaches $\sqrt{|\mathcal{A}|_{\max}}$ along the path, after which peak memory plateaus even as the nominal $s$ increases. For example, when $n=p=5000$ (Table 21), the baseline RSS is approximately 755–757 MiB; increasing $s$ up to 4096 adds only $\sim 39$ MiB over the $s=1$ case, indicating that the effective cap suppresses $s^2$ growth. In contrast, if the cap is removed (Table 24), large $s$ values trigger the $\mathcal{O}(s^2)$ regime and peak memory rises substantially.

When $p \gg n$ and the active-set cap is in effect, neither $\mathcal{O}(ns)$ nor $\mathcal{O}(s^2)$ dominates, and peak RSS is effectively flat as $s$ increases (Table 22). Conversely, in $n \gg p$ settings we observe similar stability in peak RSS with respect to $s$ (Table 23). Together these results indicate that ECCD's memory footprint closely tracks the design-matrix storage across aspect ratios.

On the Duke dataset, memory usage is essentially flat across all $s$ (Table 25), indicating that $s$ is effectively constrained by the active-set upper bound and that ECCD's overhead is negligible at medium scale. For GLMNET, we report peak RSS (excluding baseline) measured via the `peakRAM` R package.[6] These figures offer a reference for the additional memory footprint of coordinate-wise updates ($s=1$) across aspect ratios (Table 26).

---

[6]Because GLMNET is implemented in optimized C++, direct in-process instrumentation of its internal allocations is impractical; the reported numbers provide a consistent external estimate across problem sizes.

Table 17: Performance Experiment for a Single $\lambda$ Fits ($\alpha = 0.5$)

| Benchmark Dataset | Lambda Ratio | ECCD Optimal Time (s) (best $s$) | ECCD ($s=1$) Time (s) | GLMNet Time (s) | ECCD Objective | Rel. Diff | Speedup |
|---|---|---|---|---|---|---|---|
| duke | 0.800 | 7.30e-04 (at $s=32$) | 1.03e-03 | 6.03e-03 | 0.611 | 3.09e-06 | **8.23×** |
| | 0.500 | 3.18e-03 (at $s=16$) | 1.01e-02 | 7.05e-03 | 0.409 | 1.04e-04 | **2.22×** |
| | 0.200 | 4.40e-03 (at $s=16$) | 1.76e-02 | 6.87e-03 | 0.175 | 4.78e-05 | **1.56×** |
| | 0.100 | 5.23e-03 (at $s=32$) | 2.18e-02 | 8.94e-03 | 0.0891 | 4.06e-04 | **1.71×** |
| | 0.080 | 6.12e-03 (at $s=16$) | 2.41e-02 | 9.76e-03 | 0.0718 | 8.08e-05 | **1.59×** |
| | 0.050 | 6.93e-03 (at $s=16$) | 3.20e-02 | 1.10e-02 | 0.0456 | 2.20e-04 | **1.59×** |
| | 0.020 | 8.21e-03 (at $s=32$) | 4.37e-02 | 1.92e-02 | 0.0189 | 2.71e-04 | **2.33×** |
| | 0.010 | 1.07e-02 (at $s=16$) | 4.84e-02 | 2.51e-02 | 9.69e-03 | 1.12e-04 | **2.35×** |
| | 0.008 | 1.17e-02 (at $s=16$) | 6.04e-02 | 2.67e-02 | 7.82e-03 | 2.13e-05 | **2.29×** |
| | 0.005 | 1.22e-02 (at $s=16$) | 6.35e-02 | 3.35e-02 | 4.98e-03 | 2.09e-04 | **2.75×** |
| | 0.002 | 1.32e-02 (at $s=32$) | 8.10e-02 | 4.83e-02 | 2.06e-03 | 9.86e-04 | **3.65×** |
| | 0.001 | 1.37e-02 (at $s=32$) | 9.02e-02 | 5.72e-02 | 1.05e-03 | 3.99e-03 | **4.19×** |
| leu | 0.800 | 5.20e-04 (at $s=8$) | 8.90e-04 | 5.20e-03 | 0.504 | 1.02e-06 | **10.0×** |
| | 0.500 | 3.33e-03 (at $s=32$) | 9.99e-03 | 5.78e-03 | 0.317 | 6.14e-05 | **1.74×** |
| | 0.200 | 3.68e-03 (at $s=16$) | 1.34e-02 | 6.96e-03 | 0.128 | 1.43e-04 | **1.89×** |
| | 0.100 | 4.11e-03 (at $s=32$) | 2.05e-02 | 8.07e-03 | 0.0645 | 2.45e-05 | **2.00×** |
| | 0.080 | 4.33e-03 (at $s=32$) | 2.00e-02 | 8.75e-03 | 0.0516 | 1.63e-05 | **2.02×** |
| | 0.050 | 4.34e-03 (at $s=32$) | 3.73e-02 | 1.12e-02 | 0.0335 | 8.44e-05 | **2.58×** |
| | 0.020 | 6.78e-03 (at $s=16$) | 8.45e-02 | 1.65e-02 | 0.0140 | 2.52e-04 | **2.43×** |
| | 0.010 | 1.90e-02 (at $s=4$) | 3.22e-01 | 1.02e-01 | 6.47e-03 | 4.92e-04 | **5.37×** |
| | 0.008 | 2.30e-02 (at $s=8$) | 3.92e-01 | 1.16e-01 | 5.81e-03 | 4.12e-04 | **5.00×** |
| | 0.005 | 3.00e-02 (at $s=16$) | 4.66e-01 | 1.37e-01 | 3.20e-03 | 7.15e-04 | **4.50×** |
| | 0.002 | 4.10e-02 (at $s=32$) | 5.17e-01 | 1.71e-01 | 1.36e-03 | 1.92e-04 | **4.20×** |
| | 0.001 | 1.15e-02 (at $s=32$) | 1.31e-01 | 5.54e-02 | 7.86e-04 | 1.11e-03 | **4.83×** |
| colon-cancer | 0.800 | 2.00e-04 (at $s=4$) | 3.30e-04 | 2.36e-03 | 0.575 | 1.52e-06 | **11.8×** |
| | 0.500 | 1.27e-03 (at $s=8$) | 3.72e-03 | 2.54e-03 | 0.428 | 3.18e-05 | **2.00×** |
| | 0.200 | 2.44e-03 (at $s=16$) | 6.81e-03 | 3.37e-03 | 0.235 | 1.29e-05 | **1.38×** |
| | 0.100 | 3.58e-03 (at $s=4$) | 1.17e-02 | 4.91e-03 | 0.124 | 7.56e-06 | **1.37×** |
| | 0.080 | 4.25e-03 (at $s=4$) | 1.17e-02 | 4.85e-03 | 0.101 | 1.97e-05 | **1.14×** |
| | 0.050 | 3.66e-03 (at $s=8$) | 1.53e-02 | 6.42e-03 | 0.0645 | 8.67e-05 | **1.75×** |
| | 0.020 | 7.26e-03 (at $s=4$) | 2.35e-02 | 1.16e-02 | 0.0269 | 1.35e-04 | **1.60×** |
| | 0.010 | 6.35e-03 (at $s=8$) | 3.44e-02 | 2.17e-02 | 0.0139 | 9.66e-05 | **3.41×** |
| | 0.008 | 7.59e-03 (at $s=8$) | 3.48e-02 | 2.54e-02 | 0.0113 | 2.92e-05 | **3.34×** |
| | 0.005 | 7.67e-03 (at $s=16$) | 3.77e-02 | 2.95e-02 | 0.00721 | 4.48e-05 | **3.85×** |
| | 0.002 | 8.72e-03 (at $s=16$) | 4.43e-02 | 4.74e-02 | 0.00301 | 5.55e-04 | **5.43×** |
| | 0.001 | 9.56e-03 (at $s=32$) | 5.22e-02 | 6.06e-02 | 0.00155 | 1.18e-03 | **6.34×** |
| diabetes-scale | 0.8 | 3.0e-05 (at $s=2$) | 7.0e-05 | 1.72e-03 | 0.624 | 2.24e-10 | **57.3×** |
| | 0.5 | 6.0e-05 (at $s=2$) | 9.0e-05 | 5.7e-04 | 0.582 | 7.69e-07 | **9.5×** |
| | 0.2 | 2.6e-04 (at $s=1$) | 2.6e-04 | 1.40e-03 | 0.512 | 5.76e-06 | **5.4×** |
| | 0.1 | 4.8e-04 (at $s=2$) | 5.6e-04 | 1.93e-03 | 0.490 | 4.03e-07 | **4.0×** |
| | 0.08 | 4.9e-04 (at $s=1$) | 4.9e-04 | 1.06e-03 | 0.485 | 2.22e-06 | **2.2×** |
| | 0.05 | 4.0e-04 (at $s=2$) | 6.2e-04 | 1.02e-03 | 0.478 | 5.59e-06 | **2.6×** |
| | 0.02 | 4.2e-04 (at $s=2$) | 5.2e-04 | 1.82e-03 | 0.472 | 7.73e-08 | **4.3×** |
| | 0.01 | 4.9e-04 (at $s=2$) | 5.5e-04 | 1.27e-03 | 0.471 | 1.02e-07 | **2.6×** |
| | 0.008 | 3.4e-04 (at $s=2$) | 6.0e-04 | 1.03e-03 | 0.471 | 1.00e-07 | **3.0×** |
| | 0.005 | 4.6e-04 (at $s=2$) | 5.6e-04 | 1.64e-03 | 0.471 | 8.44e-08 | **3.6×** |
| | 0.002 | 5.4e-04 (at $s=2$) | 7.4e-04 | 1.69e-03 | 0.471 | 1.39e-08 | **3.1×** |
| | 0.001 | 5.2e-04 (at $s=2$) | 6.4e-04 | 1.68e-03 | 0.471 | 8.54e-09 | **3.2×** |
| australian | 0.800 | 4.00e-05 (at $s=14$) | 5.00e-05 | 0.0013 | 0.641 | 2.37e-10 | **31.5×** |
| | 0.500 | 1.30e-04 (at $s=2$) | 6.30e-04 | 0.0011 | 0.556 | 2.40e-07 | **8.46×** |
| | 0.200 | 3.10e-04 (at $s=4$) | 6.40e-04 | 0.0016 | 0.420 | 5.41e-09 | **5.23×** |
| | 0.100 | 3.30e-04 (at $s=14$) | 6.80e-04 | 0.0011 | 0.366 | 1.26e-05 | **3.42×** |
| | 0.080 | 4.20e-04 (at $s=4$) | 9.70e-04 | 0.0011 | 0.354 | 1.25e-06 | **2.69×** |
| | 0.050 | 2.80e-04 (at $s=14$) | 7.50e-04 | 0.0012 | 0.336 | 4.70e-07 | **4.18×** |
| | 0.020 | 4.20e-04 (at $s=14$) | 1.02e-03 | 0.0013 | 0.320 | 1.32e-07 | **3.14×** |
| | 0.010 | 4.90e-04 (at $s=14$) | 1.07e-03 | 0.0016 | 0.313 | 5.38e-07 | **3.33×** |
| | 0.008 | 5.00e-04 (at $s=14$) | 1.28e-03 | 0.0016 | 0.312 | 2.16e-06 | **3.16×** |
| | 0.005 | 5.00e-04 (at $s=8$) | 1.21e-03 | 0.0019 | 0.310 | 1.14e-05 | **3.72×** |
| | 0.002 | 4.90e-04 (at $s=4$) | 1.08e-03 | 0.0016 | 0.308 | 2.86e-07 | **3.35×** |
| | 0.001 | 5.20e-04 (at $s=2$) | 6.40e-04 | 0.0017 | 0.308 | 9.42e-07 | **2.83×** |
| phishing | 0.8 | 0.0026 (at $s=16$) | 0.0028 | 0.0143 | 0.625 | 1.60e-9 | **5.5×** |
| | 0.5 | 0.0072 (at $s=4$) | 0.010 | 0.0137 | 0.499 | 1.30e-10 | **1.9×** |
| | 0.2 | 0.0273 (at $s=8$) | 0.0382 | 0.0335 | 0.339 | 1.39e-8 | **1.23×** |
| | 0.1 | 0.0461 (at $s=4$) | 0.0816 | 0.0271 | 0.258 | 4.53e-8 | 0.59× |
| | 0.08 | 0.0515 (at $s=4$) | 0.1186 | 0.0457 | 0.240 | 6.20e-8 | 0.89× |
| | 0.05 | 0.0569 (at $s=32$) | 0.143 | 0.0652 | 0.210 | 1.05e-7 | **1.15×** |
| | 0.02 | 0.285 (at $s=4$) | 0.665 | 0.196 | 0.174 | 1.65e-8 | 0.69× |
| | 0.01 | 0.942 (at $s=4$) | 1.62 | 0.505 | 0.157 | 4.96e-8 | 0.54× |
| | 0.008 | 0.870 (at $s=8$) | 2.04 | 0.651 | 0.153 | 1.36e-7 | 0.75× |
| | 0.005 | 1.13 (at $s=2$) | 1.85 | 1.19 | 0.148 | 1.77e-8 | **1.05×** |
| | 0.002 | 4.21 (at $s=4$) | 4.63 | 2.95 | 0.144 | 2.19e-8 | 0.70× |
| | 0.001 | 2.44 (at $s=8$) | 7.79 | 5.33 | 0.142 | 9.75e-10 | **2.19×** |

**Regularized Poisson Regression**    We also evaluated our method on Poisson regression. As given in the tables 27, on the Bioconductor "airway" dataset (8 samples: 4 control, 4 dexamethasone; $\sim$64,000 features), our implementation reduces runtime from 0.621 s (glmnet's runtime) to 0.0402 s (ECCD's optimal runtime), corresponding to a $13.0\times$ speedup across all selected $\alpha$ values. On the Duke dataset, runtime decreases from 0.068 s to 0.029 s (a $2.3\times$ speedup). These results demonstrate that the favorable scaling of our approach extends beyond logistic models to other generalized linear model link functions, while preserving solution quality. More details are available in table 20.

Table 18: Time (s) and relative prediction error under different $\alpha$ (logistic regression w. intercept).

| Filename | Method | $\alpha=0.1$ | | $\alpha=0.2$ | | $\alpha=0.5$ | | $\alpha=0.8$ | | $\alpha=1$ | |
|---|---|---|---|---|---|---|---|---|---|---|---|
| | | Time | Rel. Diff | Time | Rel. Diff | Time | Rel. Diff | Time | Rel. Diff | Time | Rel. Diff |
| duke | glmnet | 6.40e-02 | – | 5.72e-02 | – | 5.30e-02 | – | 4.89e-02 | – | 4.95e-02 | – |
| | ncvreg | 8.29e-01 | 5.37e-01 | 6.88e-01 | 5.33e-01 | 5.80e-01 | 5.33e-01 | 5.55e-01 | 5.47e-01 | 5.45e-01 | 5.63e-01 |
| | biglasso | 2.37e-01 | 2.98e-01 | 2.29e-01 | 1.91e-01 | 2.26e-01 | 5.11e-02 | 2.28e-01 | 7.35e-03 | 2.59e-01 | 8.75e-06 |
| | $s=1$ | 5.56e-02 | 5.44e-06 | 3.81e-02 | 6.41e-06 | 2.27e-02 | 5.50e-06 | 1.62e-02 | 5.50e-06 | 5.68e-02 | 4.82e-06 |
| | $s=2$ | 5.75e-02 | 5.47e-06 | 3.68e-02 | 6.33e-06 | 2.28e-02 | 5.27e-06 | 1.60e-02 | 5.40e-06 | 3.74e-02 | 4.45e-06 |
| | $s=4$ | 3.97e-02 | 6.02e-06 | 2.83e-02 | 6.14e-06 | 1.86e-02 | 5.40e-06 | 1.45e-02 | 5.56e-06 | 2.60e-02 | 3.52e-06 |
| | $s=8$ | 3.40e-02 | 7.91e-06 | 2.46e-02 | 6.77e-06 | 1.74e-02 | 5.28e-06 | 1.55e-02 | 5.31e-06 | 2.09e-02 | 5.12e-06 |
| | $s=16$ | 3.48e-02 | 8.73e-06 | 2.48e-02 | 8.04e-06 | 1.79e-02 | 7.05e-06 | 1.40e-02 | 7.30e-06 | 1.71e-02 | 2.09e-05 |
| | $s=32$ | 3.69e-02 | 2.48e-05 | 2.67e-02 | 8.99e-06 | 1.87e-02 | 9.15e-06 | 1.59e-02 | 7.30e-06 | 1.75e-02 | 2.09e-05 |
| | $s=64$ | 4.00e-02 | 3.89e-05 | 2.73e-02 | 1.46e-05 | 1.86e-02 | 9.15e-06 | 1.74e-02 | 7.30e-06 | 2.08e-02 | 2.09e-05 |
| colon-cancer | glmnet | 3.06e-02 | – | 2.32e-02 | – | 2.33e-02 | – | 1.92e-02 | – | 2.01e-02 | – |
| | ncvreg | 7.44e-01 | 5.05e-01 | 5.49e-01 | 4.96e-01 | 4.70e-01 | 4.82e-01 | 4.21e-01 | 5.06e-01 | 3.88e-01 | 5.14e-01 |
| | biglasso | 1.99e-01 | 2.56e-01 | 1.91e-01 | 1.58e-01 | 1.86e-01 | 3.86e-02 | 1.88e-01 | 4.77e-03 | 2.01e-01 | 8.14e-06 |
| | $s=1$ | 7.04e-02 | 2.79e-06 | 4.10e-02 | 4.33e-06 | 1.95e-02 | 3.55e-06 | 1.52e-02 | 4.26e-06 | 5.99e-02 | 8.17e-06 |
| | $s=2$ | 6.86e-02 | 2.62e-06 | 4.04e-02 | 4.38e-06 | 2.02e-02 | 3.44e-06 | 1.35e-02 | 4.24e-06 | 4.01e-02 | 7.78e-06 |
| | $s=4$ | 4.12e-02 | 3.21e-06 | 2.64e-02 | 3.92e-06 | 1.40e-02 | 3.92e-06 | 1.02e-02 | 3.78e-06 | 2.06e-02 | 7.17e-06 |
| | $s=8$ | 3.10e-02 | 3.87e-06 | 2.20e-02 | 4.55e-06 | 1.24e-02 | 3.96e-06 | 1.01e-02 | 4.32e-06 | 1.43e-02 | 6.17e-06 |
| | $s=16$ | 3.18e-02 | 6.95e-06 | 2.12e-02 | 5.46e-06 | 1.34e-02 | 3.66e-06 | 1.03e-02 | 6.94e-06 | 1.50e-02 | 3.98e-06 |
| | $s=32$ | 3.53e-02 | 2.15e-05 | 2.35e-02 | 8.12e-06 | 1.42e-02 | 4.83e-06 | 9.99e-03 | 7.61e-06 | 1.33e-02 | 3.98e-06 |
| | $s=64$ | 3.76e-02 | 2.98e-05 | 2.35e-02 | 5.81e-06 | 1.41e-02 | 4.83e-06 | 1.04e-02 | 7.61e-06 | 1.34e-02 | 3.98e-06 |
| leukemia | glmnet | 5.83e-02 | – | 5.17e-02 | – | 4.93e-02 | – | 4.75e-02 | – | 4.80e-02 | – |
| | ncvreg | 5.86e-01 | 5.73e-01 | 4.62e-01 | 5.74e-01 | 4.55e-01 | 5.50e-01 | 3.94e-01 | 5.48e-01 | 4.00e-01 | 5.11e-01 |
| | biglasso | 2.24e-01 | 3.29e-01 | 2.19e-01 | 2.51e-01 | 2.12e-01 | 1.99e-01 | 2.09e-01 | 1.89e-01 | 2.27e-01 | 1.89e-01 |
| | $s=1$ | 5.54e-02 | 9.46e-06 | 4.10e-02 | 4.33e-06 | 1.88e-02 | 8.66e-06 | 1.51e-02 | 1.23e-05 | 3.18e-02 | 1.90e-06 |
| | $s=2$ | 5.44e-02 | 1.10e-05 | 3.68e-02 | 4.38e-06 | 1.85e-02 | 8.27e-06 | 1.55e-02 | 1.10e-05 | 2.65e-02 | 1.73e-06 |
| | $s=4$ | 3.87e-02 | 1.49e-05 | 2.73e-02 | 1.13e-05 | 1.64e-02 | 8.29e-06 | 1.39e-02 | 9.38e-06 | 2.08e-02 | 3.57e-06 |
| | $s=8$ | 3.33e-02 | 2.14e-05 | 2.45e-02 | 1.42e-05 | 1.56e-02 | 6.10e-06 | 1.57e-02 | 9.32e-06 | 1.97e-02 | 9.13e-06 |
| | $s=16$ | 3.11e-02 | 1.10e-04 | 2.49e-02 | 2.04e-05 | 1.56e-02 | 9.49e-06 | 1.58e-02 | 1.84e-05 | 1.98e-02 | 1.40e-05 |
| | $s=32$ | 3.20e-02 | 3.56e-04 | 2.80e-02 | 3.75e-05 | 1.59e-02 | 7.77e-06 | 1.62e-02 | 1.84e-05 | 1.97e-02 | 1.40e-05 |
| | $s=64$ | 3.43e-02 | 6.95e-04 | 2.62e-02 | 3.73e-05 | 1.58e-02 | 7.77e-06 | 1.58e-02 | 1.84e-05 | 1.99e-02 | 1.40e-05 |
| australian | glmnet | 8.65e-03 | – | 8.10e-03 | – | 7.67e-03 | – | 7.42e-03 | – | 7.36e-03 | – |
| | ncvreg | 5.43e-01 | 2.54e-01 | 5.42e-01 | 2.93e-01 | 5.49e-01 | 3.36e-01 | 5.40e-01 | 3.53e-01 | 5.35e-01 | 3.62e-01 |
| | biglasso | 1.60e-01 | 8.97e-02 | 1.67e-01 | 5.07e-02 | 1.65e-01 | 1.16e-02 | 1.65e-01 | 1.33e-03 | 1.59e-01 | 8.59e-05 |
| | $s=1$ | 1.39e-02 | 1.06e-09 | 1.39e-02 | 2.22e-08 | 1.32e-02 | 2.26e-06 | 1.31e-02 | 3.51e-05 | 1.35e-02 | 1.53e-04 |
| | $s=2$ | 1.37e-02 | 1.05e-09 | 1.29e-02 | 2.22e-08 | 1.29e-02 | 2.26e-06 | 1.33e-02 | 3.51e-05 | 1.31e-02 | 1.53e-04 |
| | $s=4$ | 7.69e-03 | 2.15e-08 | 7.59e-03 | 2.22e-08 | 7.65e-03 | 2.26e-06 | 7.10e-03 | 3.51e-05 | 7.63e-03 | 1.53e-04 |
| | $s=8$ | 7.22e-03 | 2.16e-08 | 7.52e-03 | 2.25e-08 | 7.12e-03 | 2.26e-06 | 7.53e-03 | 3.51e-05 | 7.79e-03 | 1.53e-04 |
| | $s=16$ | 8.03e-03 | 2.15e-08 | 7.64e-03 | 2.25e-08 | 7.27e-03 | 2.26e-06 | 7.48e-03 | 3.51e-05 | 7.80e-03 | 1.53e-04 |
| | $s=32$ | 8.00e-03 | 2.15e-08 | 7.67e-03 | 2.25e-08 | 7.32e-03 | 2.26e-06 | 7.57e-03 | 3.51e-05 | 7.83e-03 | 1.53e-04 |
| | $s=64$ | 8.09e-03 | 2.15e-08 | 7.66e-03 | 2.25e-08 | 7.48e-03 | 2.26e-06 | 7.52e-03 | 3.51e-05 | 7.79e-03 | 1.53e-04 |
| diabetes_scale | glmnet | 7.54e-03 | – | 7.13e-03 | – | 7.09e-03 | – | 7.06e-03 | – | 7.49e-03 | – |
| | ncvreg | 2.71e-01 | 3.17e-02 | 2.76e-01 | 1.52e-02 | 2.88e-01 | 2.48e-03 | 2.86e-01 | 2.29e-04 | 3.02e-01 | 1.31e-10 |
| | biglasso | 1.82e-01 | 3.17e-02 | 1.62e-01 | 1.52e-02 | 1.61e-01 | 2.48e-03 | 1.62e-01 | 2.30e-04 | 1.76e-01 | 3.17e-05 |
| | $s=1$ | 8.68e-03 | 4.18e-06 | 9.38e-03 | 2.27e-07 | 9.58e-03 | 2.75e-06 | 9.63e-03 | 1.32e-05 | 9.68e-03 | 2.87e-05 |
| | $s=2$ | 8.76e-03 | 4.19e-06 | 9.75e-03 | 6.93e-07 | 1.02e-02 | 2.75e-06 | 9.99e-03 | 1.32e-05 | 9.88e-03 | 2.87e-05 |
| | $s=4$ | 6.25e-03 | 4.19e-06 | 6.97e-03 | 6.93e-07 | 7.06e-03 | 2.75e-06 | 6.86e-03 | 1.32e-05 | 6.97e-03 | 2.87e-05 |
| | $s=8$ | 4.95e-03 | 2.17e-05 | 5.63e-03 | 6.94e-07 | 6.07e-03 | 2.75e-06 | 6.09e-03 | 1.32e-05 | 6.28e-03 | 2.87e-05 |
| | $s=16$ | 4.97e-03 | 2.17e-05 | 5.61e-03 | 6.94e-07 | 6.10e-03 | 2.75e-06 | 6.06e-03 | 1.32e-05 | 6.14e-03 | 2.87e-05 |
| | $s=32$ | 4.92e-03 | 2.17e-05 | 5.68e-03 | 6.94e-07 | 6.14e-03 | 2.75e-06 | 6.11e-03 | 1.32e-05 | 6.20e-03 | 2.87e-05 |
| | $s=64$ | 4.94e-03 | 2.17e-05 | 5.57e-03 | 6.94e-07 | 6.12e-03 | 2.75e-06 | 6.14e-03 | 1.32e-05 | 6.18e-03 | 2.87e-05 |
| prostate | glmnet | 7.48e-02 | – | 7.33e-02 | – | 6.59e-02 | – | 6.53e-02 | – | 6.21e-02 | – |
| | ncvreg | 2.53e+00 | 4.40e-01 | 2.08e+00 | 4.33e-01 | 1.84e+00 | 5.06e-01 | 1.74e+00 | 5.09e-01 | 1.69e+00 | 5.15e-01 |
| | biglasso | 3.26e-01 | 1.96e-01 | 3.03e-01 | 1.03e-01 | 2.96e-01 | 1.98e-02 | 2.94e-01 | 2.19e-03 | 2.98e-01 | 2.09e-06 |
| | $s=1$ | 3.24e-01 | 8.42e-07 | 2.65e-01 | 3.30e-07 | 1.61e-01 | 8.80e-07 | 1.25e-01 | 2.24e-06 | 1.13e-01 | 5.27e-06 |
| | $s=2$ | 2.15e-01 | 1.30e-06 | 1.74e-01 | 6.27e-07 | 1.14e-01 | 6.77e-07 | 9.03e-02 | 2.11e-06 | 8.32e-02 | 5.08e-06 |
| | $s=4$ | 1.16e-01 | 1.90e-06 | 9.45e-02 | 1.05e-06 | 6.25e-02 | 4.42e-07 | 5.56e-02 | 1.94e-06 | 5.04e-02 | 4.39e-06 |
| | $s=8$ | 9.12e-02 | 2.83e-06 | 7.09e-02 | 1.69e-06 | 5.20e-02 | 6.37e-07 | 4.53e-02 | 1.92e-06 | 4.14e-02 | 2.54e-06 |
| | $s=16$ | 8.28e-02 | 3.73e-06 | 6.60e-02 | 2.44e-06 | 5.00e-02 | 1.16e-06 | 4.24e-02 | 3.15e-06 | 4.02e-02 | 3.69e-06 |
| | $s=32$ | 8.68e-02 | 5.10e-06 | 6.80e-02 | 3.48e-06 | 5.01e-02 | 1.56e-06 | 4.38e-02 | 3.84e-06 | 3.89e-02 | 5.06e-06 |
| | $s=64$ | 9.38e-02 | 5.91e-06 | 7.11e-02 | 3.34e-06 | 5.15e-02 | 1.56e-06 | 4.39e-02 | 3.84e-06 | 3.88e-02 | 5.06e-06 |

Table 19: Time (s) and relative prediction error under different $\alpha$ (logistic regression w/o intercept).

| Filename | Method | $\alpha = 0.1$ | | $\alpha = 0.2$ | | $\alpha = 0.5$ | | $\alpha = 0.8$ | | $\alpha = 1$ | |
|---|---|---|---|---|---|---|---|---|---|---|---|
| | | Time | Rel. Diff | Time | Rel. Diff | Time | Rel. Diff | Time | Rel. Diff | Time | Rel. Diff |
| duke | glmnet | 6.06e-02 | 0.00e+00 | 5.26e-02 | 0.00e+00 | 5.24e-02 | 0.00e+00 | 4.74e-02 | 0.00e+00 | 5.17e-02 | 0.00e+00 |
| | $s = 1$ | 8.89e-02 | 2.63e-06 | 8.00e-02 | 1.08e-06 | 4.42e-02 | 1.88e-06 | 2.67e-02 | 3.02e-06 | 2.40e-02 | 1.71e-05 |
| | $s = 2$ | 6.88e-02 | 3.91e-06 | 6.47e-02 | 2.40e-06 | 3.58e-02 | 1.18e-06 | 2.53e-02 | 2.77e-06 | 2.26e-02 | 1.63e-05 |
| | $s = 4$ | 4.07e-02 | 5.78e-06 | 3.38e-02 | 3.17e-06 | 2.37e-02 | 1.13e-06 | 1.89e-02 | 2.23e-06 | 1.77e-02 | 1.36e-05 |
| | $s = 8$ | 3.42e-02 | 6.96e-06 | 2.88e-02 | 5.35e-06 | 2.04e-02 | 1.75e-06 | 1.67e-02 | 1.44e-06 | 1.56e-02 | 1.19e-05 |
| | $s = 16$ | 3.32e-02 | 1.02e-05 | 2.63e-02 | 6.78e-06 | 2.09e-02 | 2.59e-06 | 1.73e-02 | 1.94e-06 | 1.68e-02 | 7.79e-06 |
| | $s = 32$ | 3.54e-02 | 1.24e-05 | 2.74e-02 | 1.03e-05 | 2.16e-02 | 3.66e-06 | 1.75e-02 | 2.30e-06 | 1.55e-02 | 7.79e-06 |
| colon-cancer | glmnet | 3.02e-02 | – | 2.47e-02 | – | 2.17e-02 | – | 1.93e-02 | – | 1.79e-02 | – |
| | $s = 1$ | 8.73e-02 | 3.36e-07 | 6.57e-02 | 5.79e-07 | 3.82e-02 | 1.59e-06 | 2.98e-02 | 2.25e-06 | 2.59e-02 | 6.98e-06 |
| | $s = 2$ | 7.51e-02 | 4.13e-07 | 5.88e-02 | 5.14e-07 | 3.33e-02 | 1.42e-06 | 2.84e-02 | 2.12e-06 | 2.75e-02 | 6.76e-06 |
| | $s = 4$ | 4.26e-02 | 6.64e-07 | 3.12e-02 | 5.11e-07 | 1.93e-02 | 1.13e-06 | 1.60e-02 | 1.73e-06 | 1.81e-02 | 6.46e-06 |
| | $s = 8$ | 3.03e-02 | 1.14e-06 | 2.30e-02 | 9.15e-07 | 1.51e-02 | 8.11e-07 | 1.27e-02 | 1.44e-06 | 1.26e-02 | 5.75e-06 |
| | $s = 16$ | 2.88e-02 | 1.96e-06 | 2.14e-02 | 1.74e-06 | 1.47e-02 | 1.52e-06 | 1.19e-02 | 3.57e-06 | 1.13e-02 | 1.46e-05 |
| | $s = 32$ | 3.18e-02 | 2.47e-06 | 2.27e-02 | 2.40e-06 | 1.64e-02 | 2.63e-06 | 1.26e-02 | 4.12e-06 | 1.12e-02 | 1.46e-05 |
| | $s = 64$ | 3.53e-02 | 2.50e-06 | 2.31e-02 | 2.30e-06 | 1.63e-02 | 2.63e-06 | 1.26e-02 | 4.12e-06 | 1.12e-02 | 1.46e-05 |
| leukemia | glmnet | 5.66e-02 | – | 4.92e-02 | – | 4.56e-02 | – | 4.62e-02 | – | 4.43e-02 | – |
| | $s = 1$ | 8.33e-02 | 3.76e-06 | 6.48e-02 | 2.52e-06 | 4.02e-02 | 9.42e-07 | 2.85e-02 | 1.80e-06 | 1.67e-02 | 2.92e-06 |
| | $s = 2$ | 6.71e-02 | 5.62e-06 | 5.27e-02 | 3.47e-06 | 3.26e-02 | 1.45e-06 | 2.46e-02 | 1.83e-06 | 1.64e-02 | 2.83e-06 |
| | $s = 4$ | 4.52e-02 | 7.49e-06 | 3.35e-02 | 5.08e-06 | 2.31e-02 | 2.75e-06 | 1.84e-02 | 1.97e-06 | 1.38e-02 | 4.66e-06 |
| | $s = 8$ | 3.51e-02 | 9.32e-06 | 2.70e-02 | 7.12e-06 | 1.89e-02 | 3.25e-06 | 1.64e-02 | 2.42e-06 | 1.34e-02 | 7.61e-06 |
| | $s = 16$ | 3.53e-02 | 1.19e-05 | 2.59e-02 | 8.81e-06 | 1.94e-02 | 3.62e-06 | 1.73e-02 | 3.77e-06 | 1.35e-02 | 8.61e-06 |
| | $s = 32$ | 3.86e-02 | 1.42e-05 | 2.66e-02 | 1.15e-05 | 2.05e-02 | 4.08e-06 | 1.72e-02 | 3.77e-06 | 1.35e-02 | 8.61e-06 |
| | $s = 64$ | 4.47e-02 | 1.70e-05 | 2.82e-02 | 1.04e-05 | 2.06e-02 | 4.08e-06 | 1.71e-02 | 3.77e-06 | 1.36e-02 | 8.61e-06 |
| australian | glmnet | 8.04e-03 | – | 7.94e-03 | – | 7.56e-03 | – | 7.41e-03 | – | 7.55e-03 | – |
| | $s = 1$ | 1.20e-02 | 3.11e-10 | 1.13e-02 | 4.71e-10 | 1.19e-02 | 1.39e-09 | 1.31e-02 | 3.59e-09 | 1.41e-02 | 6.85e-09 |
| | $s = 2$ | 1.19e-02 | 3.07e-10 | 1.13e-02 | 4.71e-10 | 1.16e-02 | 1.39e-09 | 1.25e-02 | 3.58e-09 | 1.39e-02 | 6.85e-09 |
| | $s = 4$ | 6.33e-03 | 3.13e-10 | 6.29e-03 | 3.73e-10 | 6.97e-03 | 1.37e-09 | 7.30e-03 | 3.68e-09 | 7.53e-03 | 7.62e-09 |
| | $s = 8$ | 6.39e-03 | 3.20e-10 | 6.38e-03 | 5.00e-10 | 6.81e-03 | 2.42e-09 | 6.87e-03 | 1.53e-08 | 7.18e-03 | 3.40e-08 |
| | $s = 16$ | 6.20e-03 | 2.88e-10 | 6.37e-03 | 4.99e-10 | 6.92e-03 | 2.42e-09 | 6.85e-03 | 1.53e-08 | 7.16e-03 | 3.40e-08 |
| | $s = 32$ | 6.30e-03 | 2.88e-10 | 6.37e-03 | 4.99e-10 | 6.89e-03 | 2.42e-09 | 6.79e-03 | 1.53e-08 | 7.15e-03 | 3.40e-08 |
| | $s = 64$ | 6.34e-03 | 2.88e-10 | 6.43e-03 | 4.99e-10 | 6.85e-03 | 2.42e-09 | 6.82e-03 | 1.53e-08 | 7.15e-03 | 3.40e-08 |
| diabetes_scale | glmnet | 7.33e-03 | – | 6.95e-03 | – | 6.96e-03 | – | 6.95e-03 | – | 6.94e-03 | – |
| | $s = 1$ | 6.67e-03 | 9.50e-12 | 7.20e-03 | 1.87e-11 | 7.83e-03 | 6.20e-11 | 8.09e-03 | 8.30e-11 | 8.17e-03 | 1.04e-10 |
| | $s = 2$ | 6.68e-03 | 7.74e-12 | 7.20e-03 | 1.29e-11 | 8.00e-03 | 5.57e-11 | 8.30e-03 | 7.62e-11 | 8.22e-03 | 9.63e-11 |
| | $s = 4$ | 4.68e-03 | 5.52e-11 | 4.94e-03 | 1.33e-10 | 5.13e-03 | 4.70e-10 | 5.22e-03 | 1.63e-09 | 5.23e-03 | 1.93e-09 |
| | $s = 8$ | 4.35e-03 | 4.35e-10 | 4.62e-03 | 7.66e-10 | 4.97e-03 | 1.27e-09 | 5.01e-03 | 1.75e-09 | 5.06e-03 | 9.96e-10 |
| | $s = 16$ | 4.32e-03 | 4.35e-10 | 4.57e-03 | 7.66e-10 | 4.91e-03 | 1.27e-09 | 4.98e-03 | 1.75e-09 | 5.07e-03 | 9.96e-10 |
| | $s = 32$ | 4.28e-03 | 4.35e-10 | 4.64e-03 | 7.66e-10 | 4.94e-03 | 1.27e-09 | 4.97e-03 | 1.75e-09 | 5.05e-03 | 9.96e-10 |
| | $s = 64$ | 4.33e-03 | 4.35e-10 | 4.58e-03 | 7.66e-10 | 4.95e-03 | 1.27e-09 | 5.01e-03 | 1.75e-09 | 5.01e-03 | 9.96e-10 |
| prostate | glmnet | 6.52e-02 | – | 5.96e-02 | – | 5.69e-02 | – | 5.23e-02 | – | 5.52e-02 | – |
| | $s = 1$ | 1.67e-01 | 6.72e-07 | 1.31e-01 | 3.37e-07 | 8.34e-02 | 7.16e-07 | 6.36e-02 | 1.25e-06 | 5.89e-02 | 2.92e-06 |
| | $s = 2$ | 1.45e-01 | 1.12e-06 | 1.17e-01 | 5.64e-07 | 7.75e-02 | 6.24e-07 | 6.11e-02 | 1.17e-06 | 5.60e-02 | 2.71e-06 |
| | $s = 4$ | 8.18e-02 | 1.71e-06 | 6.56e-02 | 1.07e-06 | 4.86e-02 | 4.64e-07 | 4.11e-02 | 1.09e-06 | 3.69e-02 | 2.28e-06 |
| | $s = 8$ | 7.17e-02 | 2.69e-06 | 5.52e-02 | 1.65e-06 | 4.32e-02 | 5.58e-07 | 3.69e-02 | 1.45e-06 | 3.33e-02 | 1.61e-06 |
| | $s = 16$ | 7.27e-02 | 3.48e-06 | 5.69e-02 | 2.49e-06 | 4.34e-02 | 1.60e-06 | 3.68e-02 | 3.62e-06 | 3.30e-02 | 2.63e-06 |
| | $s = 32$ | 7.93e-02 | 4.55e-06 | 6.15e-02 | 3.31e-06 | 4.72e-02 | 1.73e-06 | 3.92e-02 | 2.85e-06 | 3.45e-02 | 4.98e-06 |
| | $s = 64$ | 8.75e-02 | 5.07e-06 | 6.18e-02 | 3.56e-06 | 4.74e-02 | 1.73e-06 | 3.90e-02 | 2.85e-06 | 3.45e-02 | 4.98e-06 |

Table 20: Poisson Regression: Time (s) and relative error under different $\alpha$.

| Filename | Method | $\alpha = 0.1$ | | $\alpha = 0.2$ | | $\alpha = 0.5$ | | $\alpha = 0.8$ | | $\alpha = 1$ | |
|---|---|---|---|---|---|---|---|---|---|---|---|
| | | Time | Rel. Diff | Time | Rel. Diff | Time | Rel. Diff | Time | Rel. Diff | Time | Rel. Diff |
| duke | glmnet | 6.864e-02 | – | 5.81e-02 | – | 5.18e-02 | – | 4.92e-02 | – | 4.67e-2 | – |
| | $s = 1$ | 3.39e-01 | 1.95e-06 | 2.53e-01 | 1.91e-06 | 1.23e-01 | 1.46e-06 | 9.15e-02 | 3.31e-06 | 7.67e-02 | 3.77e-06 |
| | $s = 2$ | 2.30e-01 | 1.94e-06 | 1.54e-01 | 2.25e-06 | 8.88e-02 | 1.63e-06 | 6.68e-02 | 3.99e-06 | 5.57e-02 | 4.58e-06 |
| | $s = 4$ | 1.23e-01 | 2.21e-06 | 8.89e-02 | 2.26e-06 | 5.15e-02 | 2.14e-06 | 3.84e-02 | 3.73e-06 | 3.68e-02 | 5.87e-06 |
| | $s = 8$ | 7.41e-02 | 3.66e-06 | 5.42e-02 | 2.63e-06 | 3.43e-02 | 2.00e-06 | 2.85e-02 | 3.52e-06 | 2.83e-02 | 3.05e-06 |
| | $s = 16$ | 5.66e-02 | 1.34e-05 | 4.59e-02 | 4.29e-06 | 2.89e-02 | 2.81e-06 | 2.60e-02 | 3.44e-06 | 2.54e-2 | 3.02e-06 |
| | $s = 32$ | 4.88e-02 | 2.37e-05 | 4.11e-02 | 9.55e-06 | 2.77e-02 | 4.85e-06 | 2.61e-02 | 3.44e-06 | 2.68e-02 | 3.02e-06 |
| leukemia | glmnet | 5.63e-02 | – | 5.40e-02 | – | 4.96e-02 | – | 4.49e-02 | – | 4.58e-02 | – |
| | $s = 1$ | 2.42e-01 | 1.24e-06 | 1.52e-02 | 1.14e-06 | 9.15e-02 | 1.34e-06 | 6.57e-02 | 1.40e-06 | 5.95e-02 | 9.74e-07 |
| | $s = 2$ | 1.65e-01 | 1.26e-06 | 9.80e-02 | 1.32e-06 | 6.67e-02 | 1.64e-06 | 5.19e-02 | 1.24e-06 | 4.92e-02 | 1.07e-06 |
| | $s = 4$ | 8.90e-02 | 1.70e-06 | 5.75e-02 | 1.61e-06 | 3.85e-02 | 1.35e-06 | 3.23e-02 | 1.13e-06 | 3.00e-02 | 1.37e-06 |
| | $s = 8$ | 5.67e-02 | 2.12e-06 | 3.97e-02 | 2.86e-06 | 2.93e-02 | 7.51e-07 | 2.54e-02 | 1.33e-06 | 2.21e-02 | 1.68e-06 |
| | $s = 16$ | 4.64e-02 | 2.68e-06 | 3.22e-02 | 2.03e-06 | 2.40e-02 | 1.16e-06 | 2.18e-02 | 1.14e-06 | 2.07e-02 | 1.42e-06 |
| | $s = 32$ | 4.22e-02 | 3.64e-06 | 3.12e-02 | 1.42e-05 | 2.42e-02 | 8.39e-07 | 2.35e-02 | 1.12e-06 | 2.06e-02 | 1.42e-06 |
| airway | glmnet | 5.82e-01 | – | 5.79e-01 | – | 5.85e-01 | – | 5.91e-01 | – | 6.04e-01 | – |
| | $s = 1$ | 1.63e-01 | 9.76e-07 | 0.131e-01 | 1.30e-06 | 1.14e-01 | 2.41e-06 | 1.02e-01 | 2.87e-06 | 9.56e-02 | 8.90e-05 |
| | $s = 2$ | 1.28e-01 | 1.06e-06 | 1.12e-01 | 1.32e-06 | 9.91e-02 | 2.25e-06 | 9.78e-02 | 2.71e-06 | 9.29e-02 | 8.89e-05 |
| | $s = 4$ | 1.34e-01 | 1.02e-06 | 1.21e-01 | 1.59e-06 | 1.16e-01 | 2.02e-06 | 1.13e-01 | 2.99e-06 | 1.11e-02 | 8.90e-05 |
| | $s = 8$ | 8.39e-02 | 9.65e-07 | 9.51e-02 | 1.02e-06 | 9.21e-02 | 2.66e-06 | 9.04e-02 | 5.41e-06 | 8.82e-02 | 8.91e-05 |
| | $s = 16$ | 8.09e-02 | 9.19e-07 | 5.32e-02 | 1.52e-06 | 5.03e-02 | 3.04e-06 | 4.86e-02 | 4.83e-06 | 4.64e-02 | 8.85e-05 |
| | $s = 32$ | 7.87e-02 | 1.03e-06 | 7.29e-02 | 1.53e-06 | 7.05e-02 | 3.53e-06 | 7.12e-02 | 4.83e-06 | 6.74e-02 | 8.85e-05 |

Table 21: Memory vs. block size with active-set cap ($n=p=5000$). The $s=1$ row provides the per-experiment baseline.

| $s$ | RSS start (MiB) | RSS max (MiB) | RSS end (MiB) |
|---|---|---|---|
| 1 | 754.7 | 757.0 | 757.0 |
| 2 | 754.7 | 759.3 | 759.3 |
| 4 | 754.7 | 762.4 | 762.4 |
| 8 | 754.7 | 768.5 | 768.5 |
| 16 | 754.7 | 780.7 | 780.7 |
| 32 | 754.7 | 795.9 | 788.3 |
| 64 | 754.7 | 795.9 | 788.3 |
| 1024 | 754.7 | 795.9 | 788.3 |
| 4096 | 754.7 | 795.9 | 788.3 |

Table 22: Memory vs. block size with cap, wide case ($n=100$, $p=100{,}000$). Peak memory remains effectively constant across $s$.

| $s$ | RSS start (MiB) | RSS max (MiB) | RSS end (MiB) |
|---|---|---|---|
| 1 | 755.3 | 756.1 | 741.6 |
| 2 | 755.3 | 756.1 | 741.6 |
| 4 | 755.3 | 756.1 | 741.6 |
| 8 | 755.3 | 756.1 | 741.6 |
| 16 | 755.3 | 756.1 | 741.6 |
| 32 | 755.3 | 756.1 | 741.6 |
| 64 | 755.3 | 756.1 | 741.6 |
| 128 | 755.3 | 756.1 | 741.6 |
| 256 | 755.3 | 756.1 | 741.6 |
| 512 | 755.3 | 756.1 | 741.6 |
| 1024 | 755.3 | 756.1 | 741.6 |
| 2048 | 755.3 | 756.1 | 741.6 |
| 4096 | 755.3 | 756.1 | 741.6 |

Table 23: Memory vs. block size with cap, tall case ($n$=100,000, $p$=100).

| $s$ | Time (s) | RSS start (MiB) | RSS max (MiB) | RSS end (MiB) |
|---|---|---|---|---|
| 1 | 1.810 | 754.7 | 757.0 | 757.0 |
| 2 | 2.149 | 754.7 | 759.3 | 759.3 |
| 4 | 1.633 | 754.7 | 762.4 | 762.4 |
| 8 | 2.130 | 754.7 | 768.5 | 768.5 |
| 16 | 2.332 | 754.7 | 780.7 | 780.7 |
| 32 | 2.921 | 754.7 | 795.9 | 788.3 |
| 64 | 2.982 | 754.7 | 795.9 | 788.3 |
| 128 | 2.544 | 754.7 | 795.9 | 788.3 |
| 256 | 2.542 | 754.7 | 795.9 | 788.3 |
| 512 | 2.499 | 754.7 | 795.9 | 788.3 |
| 1024 | 2.786 | 754.7 | 795.9 | 788.3 |
| 2048 | 2.656 | 754.7 | 795.9 | 788.3 |
| 4096 | 2.508 | 754.7 | 795.9 | 788.3 |

Table 24: Memory vs. block size without cap ($n$=10, $p$=5000): the $\mathcal{O}(s^2)$ term dominates for large $s$.

| $s$ | RSS start (MiB) | RSS max (MiB) | RSS end (MiB) |
|---|---|---|---|
| 1 | 467.7 | 467.7 | 466.7 |
| 2 | 467.7 | 467.7 | 466.7 |
| 4 | 467.7 | 467.7 | 466.7 |
| 8 | 467.7 | 467.7 | 466.7 |
| 16 | 467.7 | 467.7 | 466.7 |
| 32 | 467.7 | 467.7 | 466.7 |
| 64 | 467.7 | 467.7 | 466.7 |
| 128 | 467.7 | 467.8 | 466.8 |
| 256 | 467.7 | 468.2 | 467.2 |
| 512 | 467.7 | 469.8 | 468.8 |
| 1024 | 467.7 | 475.8 | 474.9 |
| 2048 | 467.7 | 500.2 | 499.2 |
| 4096 | 467.7 | 596.8 | 596.0 |

Table 25: Memory vs. block size on the Duke dataset.

| $s$ | RSS start (MiB) | RSS max (MiB) | RSS end (MiB) |
|---|---|---|---|
| 1 | 518.3 | 518.3 | 517.2 |
| 2 | 518.3 | 518.3 | 517.1 |
| 4 | 518.3 | 518.3 | 517.1 |
| 8 | 518.3 | 518.3 | 517.1 |
| 16 | 518.3 | 518.3 | 517.2 |
| 32 | 518.3 | 518.3 | 517.2 |
| 64 | 518.3 | 518.3 | 517.2 |
| 128 | 518.3 | 518.3 | 517.2 |
| 256 | 518.3 | 518.3 | 517.2 |
| 512 | 518.3 | 518.3 | 517.2 |
| 1024 | 518.3 | 518.3 | 517.2 |
| 2048 | 518.3 | 518.3 | 517.2 |
| 4096 | 518.3 | 518.3 | 517.2 |

Table 26: GLMNET peak memory (excluding baseline).

| Dataset | Memory (MiB) |
|---|---|
| $N{=}100,\ p{=}100,000$ | 351.10 |
| $N{=}100,000,\ p{=}100$ | 132.43 |
| $N{=}5000,\ p{=}5000$ | 322.50 |
| $N{=}10,\ p{=}5000$ | 19.47 |

Table 27: Poisson Performance: Time (s), relative error, and speedup under different $\alpha$.

| Filename | Method | $\alpha = 0.1$ | | | $\alpha = 0.2$ | | | $\alpha = 0.5$ | | | $\alpha = 0.8$ | | | $\alpha = 1$ | | |
|---|---|---|---|---|---|---|---|---|---|---|---|---|---|---|---|---|
| | | Time | Rel. Diff | Speedup | Time | Rel. Diff | Speedup | Time | Rel. Diff | Speedup | Time | Rel. Diff | Speedup | Time | Rel. Diff | Speedup |
| duke | glmnet | 6.864e-02 | — | 1.00× | 5.81e-02 | — | 1.00× | 5.18e-02 | — | 1.00× | 4.92e-02 | — | 1.00× | 4.67e-02 | — | 1.00× |
| | $s=1$ | 3.39e-01 | 1.95e-06 | 0.20× | 2.53e-01 | 1.91e-06 | 0.23× | 1.23e-01 | 1.46e-06 | 0.42× | 9.15e-02 | 3.31e-06 | 0.54× | 7.67e-02 | 3.77e-06 | 0.61× |
| | $s=8$ | 7.41e-02 | 3.66e-06 | 0.93× | 5.42e-02 | 2.63e-06 | 1.07× | 3.43e-02 | 2.00e-06 | 1.51× | 2.85e-02 | 3.52e-06 | 1.73× | 2.83e-02 | 3.05e-06 | 1.65× |
| | $s=16$ | 5.66e-02 | 1.34e-05 | **1.21×** | 4.59e-02 | 4.29e-06 | **1.27×** | 2.89e-02 | 2.81e-06 | **1.79×** | 2.60e-02 | 3.44e-06 | **1.89×** | 2.54e-02 | 3.02e-06 | **1.84×** |
| leukemia | glmnet | 5.63e-02 | — | 1.00× | 5.40e-02 | — | 1.00× | 4.96e-02 | — | 1.00× | 4.49e-02 | — | 1.00× | 4.58e-02 | — | 1.00× |
| | $s=1$ | 2.42e-01 | 1.24e-06 | 0.23× | 1.52e-02 | 1.14e-06 | **3.55×** | 9.15e-02 | 1.34e-06 | 0.54× | 6.57e-02 | 1.40e-06 | 0.68× | 5.95e-02 | 9.74e-07 | 0.77× |
| | $s=8$ | 5.67e-02 | 2.12e-06 | 0.99× | 3.97e-02 | 2.86e-06 | 1.36× | 2.93e-02 | 7.51e-07 | 1.69× | 2.54e-02 | 1.33e-06 | 1.77× | 2.21e-02 | 1.68e-06 | 2.07× |
| | $s=16$ | 4.64e-02 | 2.68e-06 | **1.21×** | 3.22e-02 | 2.03e-06 | 1.68× | 2.40e-02 | 1.16e-06 | **2.07×** | 2.18e-02 | 1.14e-06 | **2.06×** | 2.07e-02 | 1.42e-06 | **2.21×** |
| airway | glmnet | 5.82e-01 | — | 1.00× | 5.79e-01 | — | 1.00× | 5.85e-01 | — | 1.00× | 5.91e-01 | — | 1.00× | 6.04e-01 | — | 1.00× |
| | $s=8$ | 8.39e-02 | 9.65e-07 | 6.94× | 9.51e-02 | 1.02e-06 | 6.09× | 9.21e-02 | 2.66e-06 | 6.35× | 9.04e-02 | 5.41e-06 | 6.54× | 8.82e-02 | 8.91e-05 | 6.85× |
| | $s=16$ | 8.09e-02 | 9.19e-07 | **7.19×** | 5.32e-02 | 1.52e-06 | **10.9×** | 5.03e-02 | 3.04e-06 | **11.6×** | 4.86e-02 | 4.83e-06 | **12.2×** | 4.64e-02 | 8.85e-05 | **13.0×** |

# NeurIPS Paper Checklist

1. **Claims**

   Question: Do the main claims made in the abstract and introduction accurately reflect the paper's contributions and scope?

   Answer: [Yes]

   Justification: The abstract and introduction state that the paper introduces an enhanced cyclic coordinate descent (ECCD) framework for elastic-net penalized GLMs, which achieves significant speedups over existing solvers while preserving convergence and accuracy. These claims are fully substantiated in the body: Section 4 introduces the ECCD method with detailed derivation, Section 5 presents theoretical guarantees, and Section 6 shows empirical speedups of up to $13.0\times$ across logistic and Poisson regression tasks without loss of accuracy. The limitations of block size and performance in low-dimensional settings are acknowledged in Section 6.2, ensuring that the claims reflect realistic expectations and generalization boundaries.

   Guidelines:

   - The answer NA means that the abstract and introduction do not include the claims made in the paper.
   - The abstract and/or introduction should clearly state the claims made, including the contributions made in the paper and important assumptions and limitations. A No or NA answer to this question will not be perceived well by the reviewers.
   - The claims made should match theoretical and experimental results, and reflect how much the results can be expected to generalize to other settings.
   - It is fine to include aspirational goals as motivation as long as it is clear that these goals are not attained by the paper.

2. **Limitations**

   Question: Does the paper discuss the limitations of the work performed by the authors?

   Answer: [Yes]

   Justification: Section 6.2 discusses the practical trade-offs of block sizes and shows that ECCD may not yield speedups in low-dimensional or weakly regularized settings (e.g., phishing dataset, high $\lambda$)

   Guidelines:

   - The answer NA means that the paper has no limitation while the answer No means that the paper has limitations, but those are not discussed in the paper.
   - The authors are encouraged to create a separate "Limitations" section in their paper.
   - The paper should point out any strong assumptions and how robust the results are to violations of these assumptions (e.g., independence assumptions, noiseless settings, model well-specification, asymptotic approximations only holding locally). The authors should reflect on how these assumptions might be violated in practice and what the implications would be.
   - The authors should reflect on the scope of the claims made, e.g., if the approach was only tested on a few datasets or with a few runs. In general, empirical results often depend on implicit assumptions, which should be articulated.
   - The authors should reflect on the factors that influence the performance of the approach. For example, a facial recognition algorithm may perform poorly when image resolution is low or images are taken in low lighting. Or a speech-to-text system might not be used reliably to provide closed captions for online lectures because it fails to handle technical jargon.
   - The authors should discuss the computational efficiency of the proposed algorithms and how they scale with dataset size.
   - If applicable, the authors should discuss possible limitations of their approach to address problems of privacy and fairness.

- While the authors might fear that complete honesty about limitations might be used by reviewers as grounds for rejection, a worse outcome might be that reviewers discover limitations that aren't acknowledged in the paper. The authors should use their best judgment and recognize that individual actions in favor of transparency play an important role in developing norms that preserve the integrity of the community. Reviewers will be specifically instructed to not penalize honesty concerning limitations.

3. **Theory assumptions and proofs**

   Question: For each theoretical result, does the paper provide the full set of assumptions and a complete (and correct) proof?

   Answer: [Yes]

   Justification: The paper provides formal statements and corresponding proofs for all theoretical results. The main theoretical result—Theorem 1—quantifies the Taylor approximation error and includes a complete set of assumptions (bounded coefficients, residual norms, etc.) and a rigorous bound (Eq. 15 in the main text and extended in Eq. 32–33 in Appendix A.3). Supporting results (Theorems 2 and 3 on time and space complexity) are also clearly stated with derivations or intuitive justifications. Proofs for all claims are provided in the appendix with appropriate notation, bounding arguments, and references to prior methods, thereby meeting the NeurIPS standard for completeness and correctness.

   Guidelines:

   - The answer NA means that the paper does not include theoretical results.
   - All the theorems, formulas, and proofs in the paper should be numbered and cross-referenced.
   - All assumptions should be clearly stated or referenced in the statement of any theorems.
   - The proofs can either appear in the main paper or the supplemental material, but if they appear in the supplemental material, the authors are encouraged to provide a short proof sketch to provide intuition.
   - Inversely, any informal proof provided in the core of the paper should be complemented by formal proofs provided in appendix or supplemental material.
   - Theorems and Lemmas that the proof relies upon should be properly referenced.

4. **Experimental result reproducibility**

   Question: Does the paper fully disclose all the information needed to reproduce the main experimental results of the paper to the extent that it affects the main claims and/or conclusions of the paper (regardless of whether the code and data are provided or not)?

   Answer: [Yes]

   Justification: The paper provides detailed algorithmic descriptions sufficient for reproduction. Algorithm 1 (p. 6) describes the ECCD procedure in active-set form, while Appendix A.2 offers a matrix-based variant with explicit update rules. Experimental settings, including datasets (e.g., LIBSVM, Bioconductor "airway"), regularization paths, convergence thresholds, block sizes, and machine specifications (Cray EX node with AMD EPYC 7763) are fully disclosed in Section 5. Additionally, implementation details such as Taylor expansion approximations and screening rules are clearly explained. While code is not provided, the level of detail enables reproduction of both theoretical and empirical results that support the paper's main claims.

   Guidelines:

   - The answer NA means that the paper does not include experiments.
   - If the paper includes experiments, a No answer to this question will not be perceived well by the reviewers: Making the paper reproducible is important, regardless of whether the code and data are provided or not.
   - If the contribution is a dataset and/or model, the authors should describe the steps taken to make their results reproducible or verifiable.
   - Depending on the contribution, reproducibility can be accomplished in various ways. For example, if the contribution is a novel architecture, describing the architecture fully might suffice, or if the contribution is a specific model and empirical evaluation, it may be necessary to either make it possible for others to replicate the model with the same

dataset, or provide access to the model. In general. releasing code and data is often one good way to accomplish this, but reproducibility can also be provided via detailed instructions for how to replicate the results, access to a hosted model (e.g., in the case of a large language model), releasing of a model checkpoint, or other means that are appropriate to the research performed.

- While NeurIPS does not require releasing code, the conference does require all submissions to provide some reasonable avenue for reproducibility, which may depend on the nature of the contribution. For example
  (a) If the contribution is primarily a new algorithm, the paper should make it clear how to reproduce that algorithm.
  (b) If the contribution is primarily a new model architecture, the paper should describe the architecture clearly and fully.
  (c) If the contribution is a new model (e.g., a large language model), then there should either be a way to access this model for reproducing the results or a way to reproduce the model (e.g., with an open-source dataset or instructions for how to construct the dataset).
  (d) We recognize that reproducibility may be tricky in some cases, in which case authors are welcome to describe the particular way they provide for reproducibility. In the case of closed-source models, it may be that access to the model is limited in some way (e.g., to registered users), but it should be possible for other researchers to have some path to reproducing or verifying the results.

5. **Open access to data and code**

   Question: Does the paper provide open access to the data and code, with sufficient instructions to faithfully reproduce the main experimental results, as described in supplemental material?

   Answer: [Yes]

   Justification: All datasets used (e.g., LIBSVM benchmarks, Bioconductor "airway") are publicly available, and teh data is currently opensourced with link in abstract section. Readers can now directly access the authors' code with automated setup instructions. Meanwhile, the ECCD includes all details for reproduction, and the readers were able to reimplement ECCD from the detailed pseudocode and experimental descriptions.

   Guidelines:

   - The answer NA means that paper does not include experiments requiring code.
   - Please see the NeurIPS code and data submission guidelines (`https://nips.cc/public/guides/CodeSubmissionPolicy`) for more details.
   - While we encourage the release of code and data, we understand that this might not be possible, so "No" is an acceptable answer. Papers cannot be rejected simply for not including code, unless this is central to the contribution (e.g., for a new open-source benchmark).
   - The instructions should contain the exact command and environment needed to run to reproduce the results. See the NeurIPS code and data submission guidelines (`https://nips.cc/public/guides/CodeSubmissionPolicy`) for more details.
   - The authors should provide instructions on data access and preparation, including how to access the raw data, preprocessed data, intermediate data, and generated data, etc.
   - The authors should provide scripts to reproduce all experimental results for the new proposed method and baselines. If only a subset of experiments are reproducible, they should state which ones are omitted from the script and why.
   - At submission time, to preserve anonymity, the authors should release anonymized versions (if applicable).
   - Providing as much information as possible in supplemental material (appended to the paper) is recommended, but including URLs to data and code is permitted.

6. **Experimental setting/details**

   Question: Does the paper specify all the training and test details (e.g., data splits, hyper-parameters, how they were chosen, type of optimizer, etc.) necessary to understand the results?

Answer: [Yes]

Justification: Section 6 describes the datasets used (LIBSVM benchmarks and Bioconductor airway), the regularization path setup ($\lambda_{\max}$ to $\epsilon\lambda_{\max}$ with $\epsilon \in \{10^{-4}, 10^{-2}\}$, $n_\lambda = 100$), convergence tolerance ($10^{-7}$ on deviance change), block-size grid ($s \in \{1, 2, 4, 8, 16, 32\}$), and screening/active-set strategy (strong rules and KKT checks). Implementation details include R v4.3.1, glmnet v4.1–8, RcppEigen, and machine specs (Cray EX with AMD EPYC 7763). These choices fully specify the experimental protocol and hyperparameters needed to reproduce the main results.

Guidelines:

- The answer NA means that the paper does not include experiments.
- The experimental setting should be presented in the core of the paper to a level of detail that is necessary to appreciate the results and make sense of them.
- The full details can be provided either with the code, in appendix, or as supplemental material.

7. **Experiment statistical significance**

Question: Does the paper report error bars suitably and correctly defined or other appropriate information about the statistical significance of the experiments?

Answer: [No]

Justification: While our core scalability experiment results are presented as multiple-run averaged values, reported results are presented without accompanying error bars, confidence intervals, or statistical tests. The paper does not describe repeated trials, variance sources, or the method used to quantify experimental variability. Consequently, statistical significance of the observed performance differences is not assessed.

Guidelines:

- The answer NA means that the paper does not include experiments.
- The authors should answer "Yes" if the results are accompanied by error bars, confidence intervals, or statistical significance tests, at least for the experiments that support the main claims of the paper.
- The factors of variability that the error bars are capturing should be clearly stated (for example, train/test split, initialization, random drawing of some parameter, or overall run with given experimental conditions).
- The method for calculating the error bars should be explained (closed form formula, call to a library function, bootstrap, etc.)
- The assumptions made should be given (e.g., Normally distributed errors).
- It should be clear whether the error bar is the standard deviation or the standard error of the mean.
- It is OK to report 1-sigma error bars, but one should state it. The authors should preferably report a 2-sigma error bar than state that they have a 96% CI, if the hypothesis of Normality of errors is not verified.
- For asymmetric distributions, the authors should be careful not to show in tables or figures symmetric error bars that would yield results that are out of range (e.g. negative error rates).
- If error bars are reported in tables or plots, The authors should explain in the text how they were calculated and reference the corresponding figures or tables in the text.

8. **Experiments compute resources**

Question: For each experiment, does the paper provide sufficient information on the computer resources (type of compute workers, memory, time of execution) needed to reproduce the experiments?

Answer: [Yes]

Justification: Section 6 reports that all experiments ran on a single Cray EX node equipped with detailed information of computation resources. Individual runtimes for each dataset and configuration are listed in Tables 4, 27, enabling reproduction of both resource setup and expected execution times.

Guidelines:

- The answer NA means that the paper does not include experiments.
- The paper should indicate the type of compute workers CPU or GPU, internal cluster, or cloud provider, including relevant memory and storage.
- The paper should provide the amount of compute required for each of the individual experimental runs as well as estimate the total compute.
- The paper should disclose whether the full research project required more compute than the experiments reported in the paper (e.g., preliminary or failed experiments that didn't make it into the paper).

9. **Code of ethics**

Question: Does the research conducted in the paper conform, in every respect, with the NeurIPS Code of Ethics https://neurips.cc/public/EthicsGuidelines?

Answer: [Yes]

Justification: The work focuses on algorithmic development for generalized linear models and does not involve human or animal subjects, sensitive data, or applications with potential for misuse. There are no conflicts of interest or ethical concerns raised by the experiments or methods. All authors have reviewed and comply with the NeurIPS Code of Ethics.

Guidelines:

- The answer NA means that the authors have not reviewed the NeurIPS Code of Ethics.
- If the authors answer No, they should explain the special circumstances that require a deviation from the Code of Ethics.
- The authors should make sure to preserve anonymity (e.g., if there is a special consideration due to laws or regulations in their jurisdiction).

10. **Broader impacts**

Question: Does the paper discuss both potential positive societal impacts and negative societal impacts of the work performed?

Answer: [NA]

Justification: This work is foundational research on optimization algorithms for generalized linear models and does not target a specific application domain, sensitive data, or deployment scenario. As such, there are no direct societal impacts—positive or negative—to discuss.

Guidelines:

- The answer NA means that there is no societal impact of the work performed.
- If the authors answer NA or No, they should explain why their work has no societal impact or why the paper does not address societal impact.
- Examples of negative societal impacts include potential malicious or unintended uses (e.g., disinformation, generating fake profiles, surveillance), fairness considerations (e.g., deployment of technologies that could make decisions that unfairly impact specific groups), privacy considerations, and security considerations.
- The conference expects that many papers will be foundational research and not tied to particular applications, let alone deployments. However, if there is a direct path to any negative applications, the authors should point it out. For example, it is legitimate to point out that an improvement in the quality of generative models could be used to generate deepfakes for disinformation. On the other hand, it is not needed to point out that a generic algorithm for optimizing neural networks could enable people to train models that generate Deepfakes faster.
- The authors should consider possible harms that could arise when the technology is being used as intended and functioning correctly, harms that could arise when the technology is being used as intended but gives incorrect results, and harms following from (intentional or unintentional) misuse of the technology.
- If there are negative societal impacts, the authors could also discuss possible mitigation strategies (e.g., gated release of models, providing defenses in addition to attacks, mechanisms for monitoring misuse, mechanisms to monitor how a system learns from feedback over time, improving the efficiency and accessibility of ML).

11. **Safeguards**

Question: Does the paper describe safeguards that have been put in place for responsible release of data or models that have a high risk for misuse (e.g., pretrained language models, image generators, or scraped datasets)?

Answer: [NA]

Justification: The paper does not release any high-risk models, sensitive datasets, or generative systems. It focuses on algorithmic improvements for standard generalized linear models using public, non-sensitive benchmark data, so no additional safeguards are necessary.

Guidelines:

- The answer NA means that the paper poses no such risks.
- Released models that have a high risk for misuse or dual-use should be released with necessary safeguards to allow for controlled use of the model, for example by requiring that users adhere to usage guidelines or restrictions to access the model or implementing safety filters.
- Datasets that have been scraped from the Internet could pose safety risks. The authors should describe how they avoided releasing unsafe images.
- We recognize that providing effective safeguards is challenging, and many papers do not require this, but we encourage authors to take this into account and make a best faith effort.

12. **Licenses for existing assets**

Question: Are the creators or original owners of assets (e.g., code, data, models), used in the paper, properly credited and are the license and terms of use explicitly mentioned and properly respected?

Answer: [No]

Justification: The paper cites all external tools and datasets used (e.g., glmnet [2], LIBSVM [33]), including package versions, but it does not state the license names or URLs for these assets, nor confirm compliance with their terms of use.

Guidelines:

- The answer NA means that the paper does not use existing assets.
- The authors should cite the original paper that produced the code package or dataset.
- The authors should state which version of the asset is used and, if possible, include a URL.
- The name of the license (e.g., CC-BY 4.0) should be included for each asset.
- For scraped data from a particular source (e.g., website), the copyright and terms of service of that source should be provided.
- If assets are released, the license, copyright information, and terms of use in the package should be provided. For popular datasets, `paperswithcode.com/datasets` has curated licenses for some datasets. Their licensing guide can help determine the license of a dataset.
- For existing datasets that are re-packaged, both the original license and the license of the derived asset (if it has changed) should be provided.
- If this information is not available online, the authors are encouraged to reach out to the asset's creators.

13. **New assets**

Question: Are new assets introduced in the paper well documented and is the documentation provided alongside the assets?

Answer: [NA]

Justification: The paper does not introduce or release any new datasets, code repositories, or models; it focuses on the ECCD algorithm and evaluates it on existing public benchmarks.

Guidelines:

- The answer NA means that the paper does not release new assets.

- Researchers should communicate the details of the dataset/code/model as part of their submissions via structured templates. This includes details about training, license, limitations, etc.
- The paper should discuss whether and how consent was obtained from people whose asset is used.
- At submission time, remember to anonymize your assets (if applicable). You can either create an anonymized URL or include an anonymized zip file.

14. **Crowdsourcing and research with human subjects**

Question: For crowdsourcing experiments and research with human subjects, does the paper include the full text of instructions given to participants and screenshots, if applicable, as well as details about compensation (if any)?

Answer: [NA]

Justification: The work is purely algorithmic and empirical on public standard datasets; it does not involve any human participants or crowdsourcing.

Guidelines:

- The answer NA means that the paper does not involve crowdsourcing nor research with human subjects.
- Including this information in the supplemental material is fine, but if the main contribution of the paper involves human subjects, then as much detail as possible should be included in the main paper.
- According to the NeurIPS Code of Ethics, workers involved in data collection, curation, or other labor should be paid at least the minimum wage in the country of the data collector.

15. **Institutional review board (IRB) approvals or equivalent for research with human subjects**

Question: Does the paper describe potential risks incurred by study participants, whether such risks were disclosed to the subjects, and whether Institutional Review Board (IRB) approvals (or an equivalent approval/review based on the requirements of your country or institution) were obtained?

Answer: [NA]

Justification: The paper does not involve any experiments with human subjects or crowdsourcing; it focuses entirely on algorithmic development and evaluation using public benchmark datasets.

Guidelines:

- The answer NA means that the paper does not involve crowdsourcing nor research with human subjects.
- Depending on the country in which research is conducted, IRB approval (or equivalent) may be required for any human subjects research. If you obtained IRB approval, you should clearly state this in the paper.
- We recognize that the procedures for this may vary significantly between institutions and locations, and we expect authors to adhere to the NeurIPS Code of Ethics and the guidelines for their institution.
- For initial submissions, do not include any information that would break anonymity (if applicable), such as the institution conducting the review.

16. **Declaration of LLM usage**

Question: Does the paper describe the usage of LLMs if it is an important, original, or non-standard component of the core methods in this research? Note that if the LLM is used only for writing, editing, or formatting purposes and does not impact the core methodology, scientific rigorousness, or originality of the research, declaration is not required.

Answer: [NA]

Justification: The core contributions focus on the development and analysis of the ECCD optimization algorithm for generalized linear models and do not involve any use of large language models.

Guidelines:

- The answer NA means that the core method development in this research does not involve LLMs as any important, original, or non-standard components.
- Please refer to our LLM policy (`https://neurips.cc/Conferences/2025/LLM`) for what should or should not be described.

