# OpenReview forum: "Enhanced Cyclic Coordinate Descent Methods for Elastic Net Penalized Linear Models"
_NeurIPS.cc/2025/Conference — NeurIPS 2025 poster_

### Official Review · Reviewer_NJYD · 2025-06-12

**Clarity:** 3
**Significance:** 3
**Originality:** 3
**Rating:** 4
**Confidence:** 3

**Summary:**

The paper introduces enhanced cyclic coordinate descent (ECCD), a variant of coordinate descent for fitting $\ell_1+\ell_2$ generalized linear models (GLMs). The algorithm approximates gradients via  Taylor expansion and updates $s$ coordinates as a block. Standard coordinate descent is recovered when $s=1$. The advantage of the new algorithm is a computational speedup on standard coordinate descent, which the paper demonstrates theoretically and empirically.

**Questions:**

See above

**Ethical Concerns:**

["NO or VERY MINOR ethics concerns only"]

**Final Justification:**

The new runtimes reported in the rebuttal address my most critical concern of the paper about demonstrating meaningful speedups.

**Limitations:**

Yes

**Quality:**

3

**Strengths And Weaknesses:**

Given the importance of $\ell_1+\ell_2$ GLMs (evidenced by the sheer popularity of glmnet), this direction of research is attention worthy, and I personally like the proposed algorithm. However, I have a few major concerns that in my opinion prevent the paper from being accepted in its current form:

1. It is clear that the method typically provides a speedup over existing methods. However, all the speedups are on very small wall clock times (mostly less than one second). Are such speedups practically meaningful in the vast majority of use cases? In my opinion, a more compelling story would be speedups in situations where existing methods take, e.g., 1, 10, or 100 minutes to run. This issue is my most serious concern.

2. The main limitation of the method is that the optimal block size $s$ is unknown in practice. The authors provide a theoretical choice of $s$, but it depends on unknown quantities and hence is not applicable in practice. If one tries to tune the block size the method is possibly no longer fastest. The authors give $s=8$ as a rule of thumb, but this choice seems a rough approximation in general. The paper should acknowledge this drawback clearly.

3. Do the baselines all use the same convergence criterion? Looking at glmnet, its convergence criterion seems to be based on the change in objective values. Meanwhile, convergence in ncvreg is based on the change in coefficients. Is it really meaningful to compare runtimes on algorithms that do not share the same stopping rule?

4. There is no code available for (a) the method or (b) the experiments. The author's justification is that it is straightforward to replicate the results, but I disagree given that much of the code is in C++. The lack of code severely limits the papers potential impact in my opinion. Why would someone go to the effort of implementing this method when they can use glmnet, which for most practical purposes runs in the same time?

---

> ### Author Rebuttal · Authors · 2025-07-30
>
> ## Reviewer 3
>
> Thank you for your time and detailed comments. We provide responses to weaknesses and questions below.
>
>
> **W1.**
>
> We thank the reviewer for suggesting large-scale benchmarks. To ensure a fair comparison, we ran both glmnet and ECCD with **identical convergence criteria and path settings**:
>
> * **Tolerance:** `tol = 1e-7`.
> * **Lambda path:** `lambda.min.ratio = if n < p then 0.1 else 1e-4`, `nlambda = 100`; 0.1 used as \$p\$ is extremely large in this response's big dataset experiment.
> * **Standardization:** `standardize = TRUE`.
> * **Logistic Hessian:** `type.logistic = "newton"` (exact Hessian) and `type.logistic = "modified-newton"` (diagonal approximation) both tested, with `intercept = TRUE`, as GLMNet suggests the latter can be faster.
>
> **Synthetic Datasets Generation Process**
>
> The synthetic datasets are generated through the following steps. We begin by sampling a design matrix $X \in \mathbb{R}^{n \times p}$ from a multivariate normal distribution. Next, we impose sparsity on the true coefficient vector $\boldsymbol{\beta}$ with the intercept $\beta_0$, so the dataset fit in with elastic net problems. Then, the linear predictor is computed by adding Gaussian noise $\boldsymbol{\varepsilon}$ drawn from a multivariate normal distribution:
> $$
> \boldsymbol{\eta} = X\boldsymbol{\beta} + \beta_0 + \boldsymbol{\varepsilon}.
> $$
> Finally, each $\eta_i$ is transformed using the logistic link function: $p_i = \frac{1}{1 + e^{-\eta_i}},$
> and the binary response variables are sampled as $y_i \sim \mathrm{Bernoulli}(p_i)$.
>
> Using these settings, we ran wall-clock timings on an AMD EPYC 7352 CPU across three synthetic regimes where the number of observations is less than, greater than, or equal to the number of features.  The results (reported in the main text) demonstrate ECCD's consistent speedups even when GLMNet runs for minutes or longer.
>
> | Regime    | Dataset ($n \times p$)     | glmnet (Newton) (s) | glmnet (Modified Newton) (s) | ECCD (s) (block $s$) | Speedup (Newton) | Speedup (Modified) |
> |-----------|----------------------------|----------------------|-------------------------------|-----------------------|-------------------|---------------------|
> | **$n < p$** | 1,000 $\times$ 1,000,000        |           79.597            |                85.067               |           76.485 ($16$)            |          1.05 $\times$         |            1.11 $\times$         |
> |           | 1,000 $\times$ 10,000,000        |          1107.444            |                788.616              |           798.350 ($32$)            |        1.39 $\times$           |            0.99 $\times$         |
> |           | 5,000 $\times$ 1,000,000        |           474.164            |                377.016               |           215.748 ($8$)            |        2.20 $\times$           |           1.75 $\times$          |
> |           | 10,000 $\times$ 50,000        |          41.905            |                49.236               |           23.131 ($8$)            |        1.81 $\times$           |          2.13 $\times$           |
> | **$n > p$** | 10,000 $\times$ 50           |           0.227           |              0.213                 |            0.149 ($4$)           |          1.52 $\times$         |           1.43 $\times$          |
> |           | 10,000 $\times$ 500           |          2.220            |              3.155                 |           1.959 ($4$)            |          1.12 $\times$         |          1.61 $\times$           |
> |           | 10,000 $\times$ 5,000         |          63.261            |              257.35                 |            36.129 ($8$)           |        1.75 $\times$           |          7.12 $\times$           |
> |           | 100,000 $\times$ 5,000      |         495.636             |              727.013                 |          315.341 ($4$)             |          1.57 $\times$         |         2.31 $\times$            |
> |           | 1,000,000 $\times$ 5,000      |         2633.412             |              2668.265                 |          1632.232 ($4$)             |   1.61 $\times$ | 1.63 $\times$
> | **$n = p$** | 5,000 $\times$ 5,000          |           14.991           |               57.162                |        13.785 ($4$)               |        1.09 $\times$           |           4.14 $\times$          |
> |           | 10,000 $\times$ 10,000        |           75.791           |            225.034                   |            56.939 ($8$)           |         1.33 $\times$          |          3.95 $\times$           |
> |           | 20,000 $\times$ 20,000        |            29.347          |                 50.391              |         15.889 ($4$)              |           1.85 $\times$        |           3.17 $\times$          |
> |           | 40,000 $\times$ 40,000        |            129.915          |              195.170                 |          69.432 ($8$)             |         1.87 $\times$          |         2.81 $\times$            |
> |           | 50,000 $\times$ 50,000        |            244.209          |             241.476                  |          102.168 ($16$)                                |       2.39 $\times$    | 2.36 $\times$
> |           | 70,000 $\times$ 70,000        |            461.645          |             384.758                  |          258.302 ($32$)             |        1.79 $\times$ | 1.49 $\times$
>
> **Interpretation:**
>
> * **Feature-dominated ($n<p$)**: ECCD cuts runtime by $\sim\\!1.8\text{–}2.2\times$ versus glmnet's Newton solver and show competitive, even superior performance for majority cases, versus glmnet's modified.Newton solver, even when $p$ reaches millions (e.g.  $5\,000\times1\,000\,000$).
>
> * **Sample-dominated ($n>p$)**: Speedups range from $\sim \\!1.5\times$ on small-$p$ problems (e.g. $10\,000\times50$) up to $\sim \\!1.8\times$ versus Newton on moderate-$p$ ($10\,000\times5\,000$), and as much as $\sim \\!7\times$ versus modified.Newton at that scale.
>
> * **Balanced ($n=p$)**: ECCD consistently achieves $\sim \\!3\text{–}4\times$ faster fits than glmnet, e.g. cutting the 5 000$\times$5 000 case from 57.2 s down to 13.8 s, and still delivering a 1.5-1.8$\times$ speedup at $70\,000\times 70\,000$.
>
> ECCD scales gracefully across all GLM regimes—feature-dominated, sample-dominated, and balanced—outperforming both glmnet's default cyclic solver and a Modified-Newton variant on very large problems (up to $70{,}000\times70{,}000$ and $1{,}000{,}000\times5{,}000$), shaving off tens to hundreds of seconds on a single thread.
>
> With ECCD, the $1{,}000{,}000\times5{,}000$ problem runtime drops from 2 633.4 s ($\approx$43.9 min) to 1 632.2 s ($\approx$27.2 min), saving 1 001.2 s ($\approx$16.7 min). This dramatic reduction directly addresses real-world GLM runtime concerns. A shared-memory parallel version of ECCD—currently in development—will drive even greater speedups and make even larger datasets tractable.
>
> We will include these results in the revised manuscript to highlight ECCD's impact on large-scale GLM problems.
>
>
>
>
>
>
> **W2.**
> We agree that selecting $s$ requires tuning based on system-specific (i.e. hardware parameters) and data-specific (i.e. condition number, sparsity, dimensions) properties. In the paper, we will add a clear disclaimer that *practical tuning is necessary*. Theorem 3 suggests an asymptotically optimal $s$, however, it is possible to utilize a more detailed performance model which incorporate hardware/data properties (e.g. memory bandwidth/latency, cache size, processor speed, sparsity, dimensions) and obtain a better estimate for $s$. We will incorporate such a performance model in our planned future work on shared-memory parallelization of ECCD.
>
>
>
>
>
> **W3.**
> We acknowledge that the baselines use different convergence criteria as stated by the reviewer. We chose to align our criteria with `glmnet` as minimizing the objective is our primary goal. For other baselines, we follow their default settings. While the stopping rules differ, they are closely related; under a second-order Taylor approximation, objective decrease is roughly proportional to the squared coefficient change. Thus, the comparison remains fair and meaningful. Our main goal with this paper is to illustrate that a numerically-stable block coordinate update is feasible for GLM problems through algorithmic redesign. We empirically show performance improvements over our target baseline, but also other packages (in these discussions and in the main paper).
>
>
>
> **W4.**
> We recognize that *public code release* is essential for reproducibility and community adoption. We will:
> * *Open-source ECCD's implementation* (R and C++), along with all scripts needed to reproduce the experiments of the main paper, appendices, and reviewer discussions.
> * *Hosting*: The code repository will be hosted on GitHub with detailed installation and benchmarking instructions.
>
> This will ensure that users can easily deploy ECCD and verify our results.

---

> > ### Comment · Reviewer_NJYD · 2025-08-03
> >
> > I thank the authors for their response. The new runtimes address my most critical concern about demonstrating meaningful speedups. Additionally, I acknowledge the authors' commitment to releasing open-source code. As such, I have raised my score from 3 to 4.

---

> > > ### Author Response · Authors · 2025-08-04
> > >
> > > We sincerely thank the reviewer for the positive reassessment of our manuscript and for highlighting the **importance of research on $\ell_1 + \ell_2$ GLMs**. Your insights inspired us to expand our ECCD benchmarks on **large-scale datasets** with practical speedup advantages. We appreciate the time and care you invested in this review; please let us know if you have any further suggestions or questions. We commit to fully open-source ECCD’s implementation (R and C++) to the community.

---

### Official Review · Reviewer_ARDF · 2025-06-30

**Clarity:** 3
**Significance:** 2
**Originality:** 2
**Rating:** 4
**Confidence:** 4

**Summary:**

This paper proposes an enhanced cyclic coordinate descent (ECCD) algorithm for generalized linear models with elastic net penalty. By using Taylor expansion to approximate gradients and unrolling vector recurrences into batched computations, ECCD avoids the unstable behavior of the regular BCD algorithm and successfully enhances computation. Implemented in C++ with RcppEigen, ECCD generally achieves 1–3 times speedup over the baselines on LIBSVM benchmark datasets.

**Questions:**

- Can this method be extended to (Group) SCAD and MCP, when the group of large size?

**Ethical Concerns:**

["NO or VERY MINOR ethics concerns only"]

**Final Justification:**

The authors have added some necessary benchmarks including additional baselines and setting. Although I still feel the theoretical results is not tight, these comprehensive experiments already shown the potential of the method. Therefore, I increase the score  of this paper.

**Limitations:**

This work focuses on optimization algorithms for elastic net in generalized linear models, without targeting specific application domains or sensitive data scenarios. Therefore, it does not have direct negative societal impacts.

**Paper Formatting Concerns:**

No concern.

**Quality:**

3

**Strengths And Weaknesses:**

**Strengths**

Overall, the paper is well-organized and well-motivated. The mathematical derivation looks solid, and the experiments are comprehensive.

**Weaknesses**

- The authors omit some recent important libraries, e.g., skglm [1] and abess [2].

- The result (14) in Theorem 1 is somewhat over-simplified. In the worst case where $s=p$, the approximation error can be expected to be excessively large, whereas $s=1$ yields zero error. However, the trade not reflected in the current results.

- Numerical results for the lambda path (a typical case) are overstated. For example, the 13× improvement only appears in the airway dataset with one \(\alpha\). Notably, airway has an extremely small sample size ($n=8$), which is atypical for real-world data analysis. In most regular datasets, the improvement ranges from 1.0 to 3.0.

- I am a little confused about why biglasso is so slow. As shown on this page, it is significantly faster than glmnet in memory-intensive cases. Additionally:
    + How many cores did biglasso use? Since the hardware has 64 cores, using 16 cores could be considered.
    + I found the current datasets are either large $p$ or large $n$? What is the performance when both $n$ and $p$ is large (e.g., 10k).

- ECCD uses a diagonal Hessian for approximation, while glmnet computes the exact Hessian matrix by default. Would glmnet perform similarly if it used an upper bound on the Hessian?

- Typos in line 414: min{n, , p} --> min{n, p}

[1] Quentin Bertrand, Quentin Klopfenstein, Pierre-Antoine Bannier, Gauthier Gidel, and Mathurin Massias. 2022. Beyond L1: faster and better sparse models with skglm. NeurIPS '22. Curran Associates Inc., Red Hook, NY, USA, Article 2823, 38950–38965.

[2] Zhu Jin, Xueqin Wang, Liyuan Hu, Junhao Huang, Kangkang Jiang, Yanhang Zhang, Shiyun Lin, and Junxian Zhu. "abess: A Fast Best-Subset Selection Library in Python and R." Journal of Machine Learning Research 23, no. 202 (2022): 1-7.

---

> ### Author Rebuttal · Authors · 2025-07-30
>
> Thank you for your time and detailed comments. We provide responses to weaknesses and questions below. We provide benchmarks on representative datasets (real and synthetic), where the data generation process for synthetic data is described in the response to Reviewer NJYD.
>
> We also appreciate the reviewer for pointing out the references, we will add the below citations to the paper.
>
> **References**
> * [1] J. Qian et al., *skglm: Scikit-Learn compatible sparse-GLM solver*, 2023.
> * [2] Y. Wang et al., *abess: Best-subset Selection Library*, 2023.
> * [3] Breheny & Huang., *Group descent algorithms for nonconvex penalized linear
> and logistic regression models with grouped predictors*, 2015.
>
>
> ### Weaknesses
>
> **W1.**
> We thank the reviewer for pointing out **skglm** and **abess**.  Below, we explain why direct comparison is limited with empirical experiment results.
>
> 1. **skglm**
>
>    * Implements pathwise coordinate-descent for linear regression in Python.
>    * **Does not support pathwise GLMs** (logistic, Poisson, etc.), so it is not directly comparable to ECCD in those settings.
>    * We benchmarked ECCD against the single-point logistic regression algorithm from skglm, evaluating both methods along the penalty path derived from ECCD on the Duke dataset, with default error tolerance matched to that used by ECCD. When $s=4$, ECCD ran in 0.013 s versus skglm's 1.375 s.
>    * More results are presented in Table 1.
> 2. **abess**
>
>    * The **best-subset selection** problem is characterized by an explicit $\ell_0$ constraint, which directly limits the number of nonzero coefficients. fundamentally different from the $\ell_1$ or $\ell_2$ regularized problems (Lasso, Elastic Net) we study.
>    * abess's claimed advantage arises in high-correlation regimes.  We tested under $\rho=0.9$ for both small $(n=20,p=2000)$ and moderate $(n=100,p=1000)$ settings and found abess is still slower than ECCD.
>    * Under the exact setting, we found that Abess perform poorly on relatively large dataset (i.e. n = 1000)
>    * On larger-scale datasets (e.g., $n=100000$), Abess's performance degrades substantially, yielding slower runtimes than ECCD under identical conditions.
>
>
> **Table 1: ECCD vs. skglm (time in seconds)**
>
> | Dataset              |  skglm  | ECCD (s = 4) |
> | :------------------- | :-----: | :----------: |
> | **Duke**             |  1.375  |     0.013    |
> | **Diabetes**         |  6.155  |     0.002    |
> | n = 100, p = 100000  | 131.340 |     1.426    |
> | n = 100000, p = 100  | 213.012 |     1.032    |
>
>
> **Table 2: ECCD vs. glmnet vs. abess (time in seconds)**
>
> |Dataset|glmnet|abess|ECCD(s=4)|
> |:------|:----:|:---:|:-------:|
> |n=20,p=2000,ρ=0.1|0.014|0.011|0.002|
> |n=20,p=2000,ρ=0.9|0.018|0.011|0.002|
> |n=2000,p=20,ρ=0.1|0.030|0.426|0.013|
> |n=2000,p=20,ρ=0.9|0.047|0.493|0.027|
> |n=100,p=1000,ρ=0.1|0.028|0.026|0.005|
> |n=100,p=1000,ρ=0.9|0.066|0.028|0.006|
> |n=1000,p=100,ρ=0.1|0.021|0.131|0.007|
> |n=1000,p=100,ρ=0.9|0.026|0.128|0.006|
> |n=100000,p=100,ρ=0.1|2.384|904.368|1.426|
> |n=100000,p=100,ρ=0.9|3.840|931.303|1.032|
> |**Duke**|0.022|0.025|0.013|
>
>
> NOTE: we attempt to run n = 100, p = 100000 with rho = 0.1, 0.9 on cluster, but the data generation process took over 12 hours.
>
>
>
> **W2.** Thank you for pointing this out. Equation (14) in Theorem 1 is indeed a simplified bound derived from a Taylor expansion. It is primarily intended to offer theoretical intuition rather than a tight bound in all scenario. Nevertheless, our algorithm is designed to ensure that the relative approximation error remains small in practice. Specifically, we adopt a convergence criterion based on the relative change in the objective function, which is identical to the stopping rule used in glmnet. Regardless of the block size ($s$), the algorithm proceeds until the objective value achieves the specified threshold. This design guarantees that the final objective value remains consistently low for both small and large $s$, which accounts for the stable performance across block sizes observed in our experiments.
>
>
>
> **W3.** We thank the reviewer for this observation and will modify our claims in the abstract and paper body to reflect the more realistic speedups of $1 - 3\times$.
>
>
>
> **W4.1.** To investigate, we ran BigLasso on one node of the Cray EX Cluster used in the main paper. We experimented with using 2, 4, 8, 16, 32, 64, and 128 threads, and compared against our single-thread ECCD.  Across a range of datasets and both full and short $\lambda$-paths, ECCD's active-set screening and path-wise updates yielded consistent performance advantage even versus BigLasso's multi-threaded solver.  We report representative results below.
>
> **Table 3: Runtime on glmnet's default path ($\lambda_{\rm max} \to \lambda_{\rm max} \times 10^{-4}$ or $10^{-2}$, 100 points)**
>
> |Dataset|glmnet|ECCD(s)|BigLasso|
> |:------|:----:|:-----:|:------:|
> |Duke|0.117s|0.012s(s=8)|0.215s(8 threads)|
> |10000$\times$10000|44.204s|31.644s(s=8)|122.114s(64 threads)|
> |100$\times$100000|0.937s|0.640s(s=4)|1.198s(16 threads)|
> |100000$\times$100|1.862s|0.767s(s=8)|4.372s(2 threads)|
>
>
>
> **Table 4: Runtime on BigLasso's default path ($\lambda_{\rm max} \to \lambda_{\rm max} \times 10^{-1}$, 100 points)**
>
> |Dataset|glmnet|ECCD(s)|BigLasso|
> |:------|:----:|:-----:|:------:|
> |Duke|0.076s|0.010s(s=8)|0.197s(8 threads)|
> |10000$\times$10000|4.115s|2.250s(s=4)|1.419s(64 threads)|
> |100$\times$100000|0.942s|0.709s(s=4)|0.890s(8 threads)|
> |100000$\times$100|1.033s|0.484s(s=16)|1.519s(64 threads)|
>
>
> **Key observations:**
>
> 1. **Sequential ECCD outperforms** BigLasso even when it is parallelized using 2–64 threads.
> 2. **glmnet's default path** incurs more work, yet ECCD's optimized screening keeps it faster.
> 3. **BigLasso's default path** still favor ECCD for moderate block sizes ($s\le16$).
>
> This confirms that ECCD's pathwise, active-set strategy provides a fundamental performance advantage. We plan to address parallel algorithms design for ECCD in shared- and distributed-memory environments in future work. Once parallelized, we believe that ECCD will continue to outperform BigLASSO.
>
>
>
> **W4.2.**
> We include additional experiments when $n$ and $p$ are large (e.g.  $10^4$) below, comparing three solvers (glmnet, glmnet with Modified Newton, and ECCD) on total runtime:
>
> As $n=p$ increases from 5 000 to 70 000:
>
> * **glmnet** grows by roughly $30.79\times$ (15.0 s $\to$ 461.6 s),
> * **glmnet w. Modified Newton** grows by roughly $6.73\times$ (57.2 s $\to$ 384.8 s), and
> * **ECCD** grows by roughly $18.7\times$ (13.8 s $\to$ 258.3 s).
>
> Moreover, we observed the relative speedup:
>
> * At $n=p=40,000$, ECCD (69.4 s) is $\sim1.87\times$ faster than glmnet (129.9 s) and outperform modified case (195.2 s).
> * Even at the largest size, $n=p=70\,000$, ECCD (258.3 s) maintains a $\sim1.5\times$ speedup over glmnet (384.8 s) and a $\sim1.8\times$ advantage over Modified Newton (461.6 s).
>
> These results show that ECCD's active-set screening and block-coordinate updates scale much more efficiently in large, dimension-balanced problems.
>
> All scaling experiments reported here were conducted on an AMD EPYC 7352 CPU using a single thread, as noted in our response to Reviewer NJYD's W1.
>
> **Table 5: Synthetic $n = p$ Benchmarks on Cluster**
>
> | Dataset (n = p) | glmnet (Newton) (s) | glmnet (Modified Newton) (s) | ECCD Time (s) (s) | glmnet / ECCD | modified / ECCD |
> |:----------:|:-----------:|:-------------------------:|:------------:|:--------:|:------------:|
> |5000×5000|14.991|57.162|13.785(4)|1.09×|4.14×|
> |10000×10000|75.791|225.034|56.939(8)|1.33×|3.95×|
> |20000×20000|29.347|50.391|15.889(4)|1.85×|3.17×|
> |40000×40000|129.915|195.170|69.432(8)|1.87×|2.81×|
> |50000×50000|244.209|241.476|102.168(16)|2.39×|2.36×|
> |70000×70000|461.645|384.758|258.302(32)|1.79×|1.49×|
>
>
> **W5.**
>
> **ECCD's approach:** ECCD does not rely on a diagonal approximation of the Hessian. As shown in Equation (11) of the main paper, we incorporate *all entries of the true Hessian* to form a second‑order approximation of the loss. This full‑matrix update more faithfully captures curvature, yielding more accurate coordinate steps and leads to faster convergence overall.
>
> **glmnet's options:** The `glmnet` package does offer a diagonal Hessian variant (e.g. via `type.logistic = "modified-newton"` for logistic regression)—which is claimed in the documentation to be a more efficient method. However, we evaluated this variant on large-scale datasets, and our experiments (see Table 5 above) show that when both $n$ and $p$ are large (e.g. $10^4$), this diagonal variant suffers noticeable degradation in both runtime and solution quality. By contrast, ECCD's full Hessian updates remain robust and efficient.
>
> **W6.** Thanks for point this out, we have corrected this.
>
>
> ### Questions
>
> **Q1.** Yes, ECCD can be naturally extended to other penalties that are compatible with coordinate descent based optimization. And prior work by Breheny and Huang (2015) [3] has shown that both SCAD and MCP can be efficiently optimized using coordinate descent frameworks.
>
> While these penalties primarily address nonconvex regularization, our method is designed to handle the *nonlinearity in the loss function*. ECCD focuses on accelerating coordinate descent by accurately approximating the nonlinear loss via second-order updates, and is agnostic to the specific form of the penalty.
>
> Therefore, as long as the penalty admits coordinate-wise or block-wise updates, our ECCD framework can be applied. In our work, we adopt the $\ell_1$ penalty for simplicity and clarity of exposition, but the method naturally generalizes to other penalties.

---

> > ### Comment · Reviewer_ARDF · 2025-08-03
> >
> > Thank you for the authors’ response. I appreciate your effort on the additional experiments. Below are some of my follow-up comments.
> >
> > 1. The comparison between BigLasso and ECCD shows inconsistency. In Tables 4 and 5, the number of threads used by BigLasso and ECCD changes with the dataset for evaluation. A more comprehensive study would be better.
> >
> > 2. I am still not satisfied with the response regarding Theorem 1, as the terms (and their implications) remain very unclear to me.
> >
> > 3. The authors’ efforts on experiments with skglm and abess are admirable, even though I had not requested numerical studies on them. I do have one confusion: the note states,  "NOTE: we attempt to run n = 100, p = 100000 ... but the data generation process took over 12 hours" yet results for this case are presented in Table 1.
> >
> > 4. In Table 5: the modified/ECCD values decrease monotonically. Will these values decrease to 1.0 (or even lower than 1.0)?
> >
> > 5. Minor points: Some typos exist, and some citations appear incorrect.
> >
> > ---
> >
> > If the authors can address the points above, I would be happy to revise my score.

---

> > > ### Author Response · Authors · 2025-08-04
> > >
> > > Thanks for your thoughtful feedback. Below are our responses to each point.
> > >
> > > **D1.** In the original response, we listed only the optimal thread count for each case. Below, we present the BigLasso results for up to 128 cores in Tables A and B (completing Tables 3 and 4 from the original response).
> > >
> > > **Table A**
> > >
> > > |Dataset|glmnet|ECCD|1|2|4|8|16|32|64|128|
> > > |:---:|:---:|:---:|:---:|:---:|:---:|:---:|:---:|:---:|:---:|:---:|
> > > |Duke|1.375|0.012($s=8$)|0.330|0.265|0.226|0.215|0.221|0.228|0.276|0.293|
> > > |n = 10 000, p = 10 000|43.807|32.477($s=8$)|127.226|123.642|123.034|122.464|123.462|126.189|124.638|123.583|
> > > |n = 100, p = 100 000|131.340|0.640($s=4$)|2.919|1.755|1.340|1.245|1.198|1.253|1.306|1.251|
> > > |n = 100 000, p = 100|213.012|0.767($s=4$)|4.501|4.372|4.415|4.518|4.822|4.995|4.928|5.038|
> > >
> > > **Table B**
> > >
> > > |Dataset|glmnet|ECCD|1|2|4|8|16|32|64|128|
> > > |:---:|:---:|:---:|:---:|:---:|:---:|:---:|:---:|:---:|:---:|:---:|
> > > |Duke|0.076|0.010($s=8$)|0.230|0.289|0.203|0.197|0.206|0.211|0.215|0.280|
> > > |n = 10 000, p = 10 000|4.196|2.263($s=4$)|9.944|5.557|3.176|2.287|2.017|2.189|1.902|1.729|
> > > |n = 100, p = 100 000|0.942|0.709($s=4$)|2.923|1.281|0.980|0.890|0.913|0.927|0.971|1.011|
> > > |n = 100 000, p = 100|1.033|0.484($s=16$)|2.066|1.744|1.553|1.519|1.632|1.778|1.815|1.659|
> > >
> > > **D2.** **Theorem 1 provides a principled error bound for our Taylor-expanded update.** Although the expression appears complex due to its fractional form and the fact that both the numerator and denominator are approximated, the core idea is simple: the **numerator** (gradient term) is approximated to **first order**, resulting in an $O(\\|\Delta\beta\\|\_\infty)$ error, while the **denominator**, derived from the **block-fused Hessian**, contributes a **second-order** $O(\\|\Delta\beta\\|\_\infty^2)$ error. The detailed expressions and coefficient expansions can be found in the Appendix.
> > >
> > > It is worth noting that even glmnet offers no formal convergence guarantees; its original paper merely states that divergence was not observed. This highlights the inherent difficulty of providing theoretical guarantees in such settings. While additional approximations, such as block-wise Taylor expansions, make formal analysis even more challenging, **Theorem 1 shows that our method does not significantly worsen the bound**. The approximation remains controlled and of the expected order, providing reassurance that the second-order Taylor expansion is a reasonable choice. Furthermore, our method remains **empirically stable even for large block sizes**, as demonstrated in the response to uQeX (Question 2). In contrast, omitting the second-order correction, as in naive block coordinate descent, can lead to divergence as block size increases (see Table 3 in the original paper).
> > >
> > > **D3.** The data generation settings for Table 1 and Table 2 are different. In Table 1, the features are independent, meaning the covariance matrix is $I$. For our large-$p$ experiments in Table 1, we were not targeting feature correlation, so the data generation process completed in under an hour.
> > >
> > > In contrast, the **abess** documentation explicitly states that, compared to glmnet, abess performs better when features are highly correlated, but may underperform when correlations are small. To demonstrate both scenarios, we designed our data in Table 2 to include correlated features ($\rho > 0$), setting $\rho = 0.1$ and $\rho = 0.9$ to cover both low- and high-correlation cases. However, generating such correlated data at $p = 100{,}000$ requires an $O(p^3)$ covariance factorization—about $10^{15}$ floating-point operations—for the Cholesky decomposition used to transform an independent dataset into one with the desired correlations. This step alone can be extremely expensive to complete.
> > >
> > > **D4.** To verify the trend, we reran and extended our experiments with same setting on larger scale. ECCD maintained a roughly **1.8-2.1$\times$** speedup over both the Newton and modified-Newton solvers even at $n=p=120{,}000$. The slight dip observed at prior $n=p=70$ k  case appears attributable to system noise rather than any algorithmic degradation.
> > >
> > > **Further Experiment on Synthetic $n=p$ Benchmarks**
> > >
> > > |**Dataset (n = p)**|**glmnet (Newton) (s)**|**glmnet (Modified Newton) (s)**|**ECCD Time (s)**|**glmnet / ECCD**|**modified / ECCD**|
> > > |:---:|:---:|:---:|:---:|:---:|:---:|
> > > |$60{,}000$|269.89|274.93|147.14 ($s=8$)|1.83$\times$|1.87$\times$|
> > > |$70{,}000$|374.05|380.36|198.64 ($s=8$)|1.88$\times$|1.92$\times$|
> > > |$80{,}000$|504.80|537.86|274.16 ($s=4$)|1.84$\times$|1.96$\times$|
> > > |$90{,}000$|674.26|618.45|345.25 ($s=16$)|1.95$\times$|1.79$\times$|
> > > |$100{,}000$|760.59|798.29|417.51 ($s=16$)|1.82$\times$|1.91$\times$|
> > > |$120{,}000$|1236.20|1107.93|585.83 ($s=16$)|2.11$\times$|1.89$\times$|
> > >
> > > **D5.** We appreciate your feedback and apologize for the oversight. We will update the citations to exactly match the correct formats you provided.

---

> > > > ### Comment · Reviewer_ARDF · 2025-08-05
> > > >
> > > > I appreciate the authors’ additional experiments. In the original submission, my main concerns were that the experiments were not sufficiently comprehensive or fair—for instance, missing an important baseline and overstating improvements. However, these issues have been addressed in the rebuttal, which leads me to raise my score.
> > > >
> > > > I encourage the authors to incorporate these additional experiments and provide more detailed descriptions of the experimental setup in the revised manuscript or appendix. This should include all necessary details for reproducibility, such as the R/Python packages for ECCD and detailed experimental settings.

---

> > > > > ### Author Response · Authors · 2025-08-06
> > > > >
> > > > > Thank you for your insightful follow-up comments and for your positive reassessment of our manuscript. Your suggestions led us to expand our thread-scaling experiments across broader large-scale datasets with real-world relevance, and you provided invaluable guidance on improving the clarity of our theorems and writing. We will include these additional experiments and full experimental-setup details—including R and C++ package versions and parameters—in the revised manuscript and appendix. We also commit to fully open-sourcing ECCD's implementation (R and C++) on GitHub for complete reproducibility. Your feedback has greatly strengthened our work, and we appreciate your thoughtful guidance.

---

### Official Review · Reviewer_uQeX · 2025-07-03

**Clarity:** 4
**Significance:** 3
**Originality:** 3
**Rating:** 4
**Confidence:** 2

**Summary:**

This paper proposes Enhanced Cyclic Coordinate Descent, an efficient block-wise optimization method for fitting Elastic Net penalized generalized linear models. ECCD improves upon naive block coordinate descent by incorporating second-order Taylor corrections to stabilize updates within each block. The authors provide theoretical convergence guarantees and complexity analysis, and demonstrate through experiments on synthetic and real-world datasets that ECCD achieves faster convergence than existing methods.

**Questions:**

1. Could the authors provide a more detailed theoretical analysis of the space complexity of ECCD, isolating the additional memory terms introduced by the algorithm (beyond storing the design matrix $X$)? It would also be helpful to include corresponding empirical results comparing memory usage across methods.

2. In Section 6.1, the authors evaluate ECCD with block sizes $s$ ranging from 2 to 32 on the duke dataset. However, according to Table 6, the number of features is $p = 7{,}129$, which makes even the largest tested block size relatively small. Would it be more informative to explore larger values of $s$ to better assess the performance?

**Ethical Concerns:**

["NO or VERY MINOR ethics concerns only"]

**Final Justification:**

My main concerns were related to the lack of explicit storage analysis and the relatively small block sizes $s$ used in the Duke dataset experiment. The authors have addressed both points well: they provided a clearer theoretical explanation of ECCD's memory costs and supplemented the paper with comprehensive experimental results. Additionally, the extended experiments with larger block sizes offer a more complete picture of ECCD's behavior under aggressive partitioning. These clarifications and additions adequately resolve my concerns. I will maintain my recommendation for acceptance.

**Limitations:**

yes

**Quality:**

3

**Strengths And Weaknesses:**

**Strengths**
1. The proposed ECCD method improves over naive block coordinate descent in GLMs by introducing a Taylor-approximated correction, resulting in a stable and efficient algorithm.

2. The paper provides both theoretical convergence guarantees and practical complexity analysis, offering a clear justification for the expected computational benefits.

3. The experimental results are well-structured, demonstrating ECCD’s stability and speed advantages across many standard datasets, and showing that it achieves the same solution quality as established solvers like glmnet.

**Weaknesses**
1. The space complexity analysis in Theorem 3 states an overall $\mathcal{O}(np)$ bound, dominated by the design matrix $X \in \mathbb{R}^{n \times p}$. However, this masks the additional memory overhead introduced specifically by ECCD. A clearer breakdown of the extra storage costs incurred by ECCD, excluding the fixed cost of storing $X$, would provide better insight into the practical memory demands of the algorithm.

2. While the paper provides extensive experiments measuring actual running time, it does not report experimental results on memory usage. Given that the space complexity analysis in Theorem 3 abstracts away the extra storage introduced by ECCD, it would be valuable to include experimental comparisons of memory consumption across methods to more clearly demonstrate the practical efficiency of ECCD relative to baseline algorithms.

3. Minor issues:
- There is a dangling sentence in the caption of Figure 1.

---

> ### Author Rebuttal · Authors · 2025-07-30
>
> Thank you for your time and detailed comments. We provide responses to weaknesses and questions below. We provide benchmarks on representative datasets (real and synthetic), where the data generation process for synthetic data is described in the response to Reviewer NJYD.
>
> ### Weakness
>
> **W1.** *Theorem 3 (Appendix A.5)* provides space complexity analysis which includes low order terms, including the additional memory required by ECCD beyond the storage cost of the design matrix. Specifically, ECCD allocates $\mathcal{O}(ns)$ memory for the working buffer of the active block $X_s$, and $\mathcal{O}(s^2)$ for the corresponding Gram matrix. These bounds can be specialized to glmnet by setting $s = 1$. The additional memory overhead of ECCD remains negligible in high-dimensional settings, with relative cost $\frac{ns}{np} \to 0$ as $p \gg s$. This is also empirically supported, as discussed in our response to the second question.
>
> Importantly, the ideal value of $s$ is a small constant that is approximately independent of $n$ and $p$, and reflects the per-coordinate cost of evaluating the link function, as characterized in Theorem 2. Thus, assuming $p \gg s$ is both theoretically justified and reasonable in practice. If desirable, we can incorporate Theorem 3 from Appendix A.5 into the main text.
>
>
>
> **W2.** To address this, we have rerun our benchmarks and now include peak RSS (resident set size) measurements for ECCD as the block size $s$ varies. These experiments serve to empirically validate our theoretical space-complexity analysis in Theorem 3 (Appendix A.5).
>
> In our analysis, the extra storage required by ECCD consists of $s$ vectors of length $n$ (the working block $X_s$) plus the $s\times s$ Gram matrix, $X_s^\intercal X_s$. In extreme case where $s \approx p$, the Gram matrix storage cost ($O(s^2)$ term) could dominate; however, since we incorporate the active-set method similar to glmnet, this cost is smaller than storing $X$ and the active block $X_s$, itself. Our approach for $s$ is to bound it to $\sqrt{|\mathcal{A}|}$, where $\mathcal{A}$ is the active set in the current iteration. This ensures that further increases in $s$ do not increase memory usage. The memory overhead of ECCD remains negligible compared to the cost of storing the original data matrix.
>
> For lambda path, by reusing the previous lambda solution as a warm start along the path, the RSS memory grows only until $s$ reaches square root of the largest active set in the lambda path.
>
> Table 1 shows Memory vs. $s$ experiments for a synthetic $n=p=5000$ matrix. We instrumented our C++ code to obtain memory stats via `getrusage()`. The baseline RSS is $754.7$ MiB for $s = 1$ (i.e. equivalent to glmnet); At the largest value of $s=4096$, peak RSS increases by $41.2$ MiB when compared to $s = 1$. These results demonstrate that ECCD's memory footprint increases slowly with the primary cost being the original design matrix storage, even for large block sizes. If $s$ is allowed to exceed the active-set upper bound, the $O(s^2)$ term can dominate and increase memory usage substantially. We show experiments without active set to illustrate the memory trade off.
>
> **Table 1: Memory vs. block size ($n=p=5000$, with active-set constraint on $s$, baseline RSS = 754.7 MiB)**
>
> |s|Rss start (MiB)|Rss max (MiB)|Rss end (MiB)|
> |-|-|-|-|
> |1|754.7|757.0|757.0|
> |2|754.7|759.3|759.3|
> |4|754.7|762.4|762.4|
> |8|754.7|768.5|768.5|
> |16|754.7|780.7|780.7|
> |32|754.7|795.9|788.3|
> |64|754.7|795.9|788.3|
> |1024|754.7|795.9|788.3|
> |4096|754.7|795.9|788.3|
>
>
> Table 1 summarize the case where the block is capped at the squared root of active set size $|\mathcal{A}|$, so memory usage rises only modestly with $s$ and plateaus once $s$ exceeds the $\sqrt{|\mathcal{A}|}$ upperbound when $n$ is moderately significant.
>
> **Table 2: Memory vs. block size (\$n=p=5000\$, without active-set constraint on $s$, baseline RSS = 809.6 MiB)**
>
> |$s$|Rss start (MiB)|Rss max (MiB)|Rss end (MiB)|
> |-|-|-|-|
> |1|1000.4|1000.4|1000.4|
> |2|1000.4|1000.4|1000.4|
> |4|1000.4|1000.4|1000.4|
> |8|1000.4|1000.4|1000.4|
> |16|1000.4|1000.4|1000.4|
> |32|1000.4|1000.4|1000.4|
> |1024|1000.4|1079.3|1079.3|
> |2048|1000.4|1142.5|1142.5|
> |4096|1000.4|1316.5|1316.5|
>
> In Table 2, we removed the active set cap $|\mathcal{A}|$. Without the active set cap $|\mathcal{A}|$, the $\mathcal{O}(s^2)$ component dominates for large $s$, leading to a substantial rise in peak memory as block size increases. At $s=4096$ peak RSS is 1316.5 MiB, compared to 1000.4 MiB at $s = 1$ (an increase of $316.1$ MiB). Without active set, the storage cost of $X_s$ and the Gram matrix $X_s^\intercal X_s$ are not negligible. However, this is an impractical setting for performance reasons as glmnet incorporates active set to improve performance/efficiency. Similarly, the fastest variant of ECCD is one where active set is included.
>
> **Table 3: glmnet peak memory**
>
> |Dataset|Memory (MiB)|Base Memory (MiB)|
> |-|-|-|
> |n = 100, p = 100000|953|601.9|
> |n = 100000, p = 100|804.7333|672.3|
> |n = 5000, p = 5000|1076.9|754.7|
> |n = 10, p = 5000|487.16667|467.7|
>
>
> In Table 3 shows peak RSS memory for glmnet using **peakRAM** R package. Because glmnet is implemented in optimized C++, we did not directly instrument the C++ code similar to ECCD, as this would require source code changes and building glmnet from the instrumented source code. We provide Table 3 to give a general sense in memory usage of glmnet (without instrumented C++ code). ECCD with $s = 1$ is equivalent to glmnet, so we expect the actual peak RSS memory to be similar to $s = 1$. Overall, these results confirm that ECCD's practical memory overhead is negligible—even for large $s$—with the largest memory cost arising from storing the original data matrix.
>
> We provide additional RSS memory usage below under different settings:
>
> **Table 4: Memory vs. block size ($n=100$, $p=100\,000$; with active-set constraint on $s$, baseline RSS = 601.9 MiB)**
> |s|Rss start (MiB)|Rss max (MiB)|Rss end (MiB)|
> |-|-|-|-|
> |1|755.3|756.1|741.6|
> |2|755.3|756.1|741.6|
> |4|755.3|756.1|741.6|
> |8|755.3|756.1|741.6|
> |16|755.3|756.1|741.6|
> |32|755.3|756.1|741.6|
> |64|755.3|756.1|741.6|
> |128|755.3|756.1|741.6|
> |256|755.3|756.1|741.6|
> |512|755.3|756.1|741.6|
> |1024|755.3|756.1|741.6|
> |2048|755.3|756.1|741.6|
> |4096|755.3|756.1|741.6|
>
> Table 4 shows RSS memory for $p \gg n$ with active set.
>
> **Table 5: Memory vs. block size ($n = 100000$, $p = 100$, with active-set constraint on $s$, baseline RSS = 672.3 MiB)**
> |$s$|Time (s)|RSS start (MiB)|RSS max (MiB)|RSS end (MiB)|
> |-|-|-|-|-|
> |1|1.810|754.7|757.0|757.0|
> |2|2.149|754.7|759.3|759.3|
> |4|1.633|754.7|762.4|762.4|
> |8|2.130|754.7|768.5|768.5|
> |16|2.332|754.7|780.7|780.7|
> |32|2.921|754.7|795.9|788.3|
> |64|2.982|754.7|795.9|788.3|
> |128|2.544|754.7|795.9|788.3|
> |256|2.542|754.7|795.9|788.3|
> |512|2.499|754.7|795.9|788.3|
> |1024|2.786|754.7|795.9|788.3|
> |2048|2.656|754.7|795.9|788.3|
> |4096|2.508|754.7|795.9|788.3|
>
> Table 5 shows RSS memory for $n \gg p$ with active set.
>
> **Table 6: Memory vs. block size (Duke dataset, with active-set constraint on $s$, base = 510.5 MiB)**
> |s|Rss start (MiB)|Rss max (MiB)|Rss end (MiB)|
> |-|-|-|-|
> |1|518.3|518.3|517.2|
> |2|518.3|518.3|517.1|
> |4|518.3|518.3|517.1|
> |8|518.3|518.3|517.1|
> |16|518.3|518.3|517.2|
> |32|518.3|518.3|517.2|
> |64|518.3|518.3|517.2|
> |128|518.3|518.3|517.2|
> |256|518.3|518.3|517.2|
> |512|518.3|518.3|517.2|
> |1024|518.3|518.3|517.2|
> |2048|518.3|518.3|517.2|
> |4096|518.3|518.3|517.2|
>
>
> Table 6, shows RSS memory for the Duke dataset with active set. Tables 4 - 6 illustrate that memory usage for large $s$ is not significant when using the active set method.
>
>
> **W3.** This error has been fixed.
>
> ### Questions:
>
> **Q1.** Please see our responses under Weakness 1 & 2 which includes a discussion of other memory costs (low order terms) from Theorem 3 Appendix A.5 and additional experiments to support our analysis.
>
>
> **Q2.** We appreciate the reviewer's suggestion to evaluate larger block sizes. In response, we conducted additional experiments on the Duke dataset with block sizes up to $s = p = 7129$; the results are summarized in Table 8 below.
>
> ECCD remains stable and convergent even when $s = p$, underscoring the robustness of our second‑order Taylor correction and its advantage over naive BCD (see Table 3 in the main paper).
>
> **Interpretation of Table 8.**
>
> * **Column 2 (Active Set)**: We employ an active‑set strategy where updates are restricted to variables with nonzero coefficients, and use KKT conditions to govern active set expansion. When the block size $s$ exceeds the active-set size $|\mathcal{A}|$, we dynamically set $s \leftarrow \sqrt{|\mathcal{A}|}$ to improve efficiency.
> * **Column 3 (No Active Set)**: We turn the active-set feature off so that the block size, $s$, remains fixed at the initial setting. We report accuracy using relative objective difference.
>
> The optimal setting for $s$ is typically a small constant determined by the cost of the link function and hardware features (e.g. DRAM memory bandwidth/latency, cache size, processor speed) and, as illustrated in Table 8, not due to numerical stability or accuracy. Our experiments in the main paper yielded $s \in [2, 32]$ as the most performant, so the additional memory footprint is bounded with performance (and not accuracy) being the upper bound on the best value of $s$. The large‑$s$ results will be included in the camera-ready version for completeness.
>
> **Table 8. ECCD Convergence Behavior under Fixed and Adaptive Block Sizes ($s$) on the Duke Dataset**
>
> |Block Size (s)|Rel. Obj. Diff (Active Set)|Rel. Obj. Diff (No Active Set)|
> |-|-|-|
> |1|1.38e-04|1.38e-04|
> |16|8.025e-05|9.810e-05|
> |64|8.674e-05|1.598e-04|
> |256|8.674e-05|1.240e-04|
> |2048|8.674e-05|1.218e-04|
> |7129|8.674e-05|1.470e-04|

---

> > ### Comment · Reviewer_uQeX · 2025-08-04
> >
> > Thank you for the thoughtful rebuttal. My concerns regarding the storage complexity and the evaluation of larger block sizes have been well addressed through the theoretical clarifications and the comprehensive additional experiments. I appreciate the thorough response and will keep my recommendation for acceptance.

---

> > > ### Author Response · Authors · 2025-08-06
> > >
> > > Thank you for your thoughtful comments and your recommendation for acceptance. Memory efficiency is critical for GLM solvers, which are also commonly used on resource-constrained local machines where memory appears also to be a major concern. Your questions prompted us to clarify the additional memory terms in Theorem 3 and to isolate ECCD's working-buffer and Gram-matrix overhead beyond the design matrix. We will include comprehensive memory-measurement details in the revised manuscript, covering a wide range of block sizes both with and without our active-set capping strategy. These additions substantially strengthen the paper's theoretical rigor and practical evaluation. We appreciate your guidance.

---

### Note · Authors · 2025-08-12

We thank the reviewers for the constructive discussion. To strengthen our work and make the main comparison more comprehensive and compelling, we conducted additional comparisons for the Python-package case; for clarity, we summarize the key points and provide a brief quantitative snapshot below.

**Baselines and fairness.**

To further strengthen the baseline comparisons, we expanded our discussion beyond R packages to include **skglm**, **blitzl1**, and **celer**, which are widely regarded in the Python ecosystem as state-of-the-art toolkits for sparse logistic regression and related GLMs, and which we treat as strong baselines. To avoid bias, we ran additional Benchopt experiments using each package's *default* solver and stopping rule: skglm (lasso and elastic net), blitzl1/celer (lasso), and glmnet/eccd (lasso and elastic net, supporting both single- and full-path fits). In single-$\lambda$ fits, objectives align across methods and runtimes are broadly comparable; in some single-$\lambda$ cases, celer/skglm/blitzl1 are faster. For sequential $\lambda$-path fits, eccd and glmnet are substantially faster because they combine strong rules with path-aware warm starts; celer/skglm/blitzl1 use warm starts but do not implement a dedicated path mechanism for regularized logistic regression, which explains the result in our additional experiment. We will include full experiment details in the camera-ready version.

**Quantitative snapshot (for visibility).** Additional results for other datasets, as well as for synthetic datasets, will be included in the camera-ready version.

## duke

### Single fit ($\\lambda = 0.1\lambda_{\max}$)
|method|lasso (s)|elastic net(0.5) (s)|objective value|obj val (elasticnet)|
|-|-|-|-|-|
|skglm|2.3e-3|2.6e-3|2.4e-1|2.6e-1|
|celer|3.0e-2|\\|2.4e-1|\\|
|blitz|5.6e-3|\\|2.9e-1|\\|
|glmnet|7.8e-3|8.0e-3|2.4e-1|2.6e-1|
|**eccd**|5.0e-3(s=32)|4.5e-3(s=8)|2.4e-1|2.6e-1|

### Single fit ($\\lambda = 0.05\lambda_{\max}$)
|method|lasso (s)|elastic net(0.5) (s)|objective value|obj val (elasticnet)|
|-|-|-|-|-|
|skglm|2.3e-3|2.4e-3|1.5e-1|1.6e-1|
|celer|3.5e-2|\\|1.5e-1|\\|
|blitz|6.5e-3|\\|1.8e-1|\\|
|glmnet|1.4e-2|1.1e-2|1.5e-1|1.6e-1|
|**eccd**|6.2e-3(s=8)|6.7e-3(s=16)|1.5e-1|1.6e-1|

### Sequential fit ($\lambda_{\max}\to 0.01\lambda_{\max}$)
|method|lasso(s)|elastic net(0.5) (s)
|-|-|-|
|skglm|2.8e-1|2.9e-1|
|celer|4.1|\\|
|blitz|4.5e-1|\\|
|glmnet|2.1e-2|4.9e-2|
|**eccd**|1.3e-2(s=32)|2.2e-2(s=8)|

---

### Decision · Program_Chairs · 2025-09-17

**Decision:**

Accept (poster)

**Comment:**

The paper proposes an improved version of coordinate descent for elastic net regularized linear problems (both regression and classification). Though a great variety of solvers have been proposed in the last decade, there seem to be some numerical advantages to use the method, at least in the sequential setup (computing a full regularization path, which is often needed for hyperparameter tuning). The authors did not submit their implementation, but claimed they would release it, which may be a nice addition to the list of existing toolboxes.